_Article_

# E3 ligase AREL1 controls perinuclear localization of lysosomes and supports Purkinje cell survival

Luyi Jiang, Jiangfen Tang, Ya-Fen Zhang, Wen-Xuan Zou, Gang Deng, Na Tian, Xiaolu Zhao [ID], Lei Han, Kai Liu, Bao-Liang Song [ID] & Jie Luo [ID] [✉]

## Abstract

**Localization of lysosomes influences their properties, e.g., perinuclear lysosomes are more acidic but less mobile compared with the peripheral ones. Furthermore, the endoplasmic reticulum (ER) can actively regulate the dynamics and functions of lysosomes via membrane contact sites. In this study, we find that ER-resident apoptosis-resistant E3 ubiquitin protein ligase 1 (AREL1) establishes membrane contacts with lysosomes by directly interacting with the $V_o a$ subunit of V-ATPase. AREL1 also catalyzes K33-linked polyubiquitylation of V-ATPase $V_1 B2$ subunit, inducing its binding to UBAC2 localized in the perinuclear ER. Depletion of AREL1 or UBAC2 increases the number of peripheral lysosomes that possess partially assembled V-ATPase, elevated luminal pH, and attenuated degradative capacity. Knockdown of ZRANB1, the deubiquitylating enzyme that antagonizes AREL1-mediated $V_1 B2$ ubiquitylation, promotes perinuclear clustering of lysosomes and increases lysosomal acidity and degradation. Mice lacking _Arel1_ exhibit age-dependent Purkinje cell loss, an ataxic phenotype, and motor impairment. Lipofuscin accumulation in the residual Purkinje cells of $Arel1^{-/-}$ mice indicates lysosomal dysfunction. Orchestration of lysosomal positioning and function by the AREL1–UBAC2–V-ATPase axis underscores the physiological significance of ER-regulated perinuclear lysosomal positioning in neurons.**

**Keywords** AREL1; Lysosomal Positioning; V-ATPase; UBAC2; Purkinje Neurons
**Subject Categories** Membranes & Trafficking; Neuroscience; Post-translational Modifications & Proteolysis

## Introduction

Lysosomes are highly heterogeneous organelles whose spatial distributions and functions are tightly correlated. They tend to cluster in the perinuclear region under normal conditions (Jongsma et al, 2016). Compared with the peripheral ones, perinuclear lysosomes exhibit higher acidity and reduced motility, which

ensure optimal activity of hydrolytic enzymes, enhanced fusion of autophagosomes with lysosomes, and eventually efficient degradation of a broad range of substrates (Johnson et al, 2016; Jongsma et al, 2016; Rayens et al, 2022). When nutrients (amino acids, lipids, and growth factors) are abundant, lysosomes are induced to transport to the cell periphery, where they serve as a signaling platform for mammalian target of rapamycin (mTOR) to sense and transduce nutritional inputs (Jia and Bonifacino, 2019; Korolchuk et al, 2011; Raiborg, 2018).

The endolysosomal transport and positioning are modulated by a complex array of factors including motors and the associated proteins as well as interorganelle contacts (Bonifacino and Neefjes, 2017; Jongsma et al, 2020; Jongsma et al, 2023), particularly ER–endolysosome membrane contact sites (MCSs) (Friedman et al, 2013; Gao et al, 2022; Palomo-Guerrero et al, 2019; Raiborg et al, 2015; Rocha et al, 2009). The ER-embedded UBE2J1/RNF26 ubiquitylation complex has been shown to act cooperatively with USP15, probably USP17 as well, to establish perinuclear positioning of the endolysosomal system (Cremer et al, 2021; Jongsma et al, 2016; Lin et al, 2022). The sorting nexin SNX19 is another ER protein responsible for constraining endolysosomes in the perinuclear region via interacting with phosphatidylinositol 3-phosphate on the endolysosomal surface (Saric et al, 2021). It remains to be determined whether there are other molecular complexes that can actively tether lysosomes to the perinuclear ER, and whether defects in lysosomal perinuclear positioning may impair animal physiology.

The vacuolar-type adenosine triphosphatase (V-ATPase) is a major determinant of luminal acidity of various organelles, including lysosomes. The active holoenzyme is composed of a cytosolic $V_1$ domain for ATP hydrolysis and a membrane-embedded $V_o$ domain for proton pumping (Vasanthakumar and Rubinstein, 2020). These two domains can dissociate to become autoinhibited and reassemble to confer V-ATPase activity in response to various nutrient cues (Collins and Forgac, 2020; McGuire et al, 2017). In mammalian cells, the reassembly of lysosomal V-ATPase is increased by glucose and amino acid starvation (McGuire and Forgac, 2018; Stransky and Forgac, 2015), as well as when mammalian target of rapamycin complex 1 (mTORC1) is inactive (Ratto et al, 2022) or lysosomal phosphatidylinositol 4-phosphate accumulates (Ebner et al, 2023). Intriguingly, these conditions also cause perinuclear clustering of

State Key Laboratory of Metabolism and Regulation in Complex Organisms, Hubei Provincial Research Center for Basic Biological Sciences, College of Life Sciences, Taikang Center for Life and Medical Sciences, Wuhan University, 430072 Wuhan, China. ✉E-mail: jieluo@whu.edu.cn

lysosomes. The results suggest that V-ATPase assembly/disassembly and lysosomal positioning are closely correlated.

In this study, we identify apoptosis-resistant E3 ubiquitin protein ligase 1 (AREL1) as a critical regulator of lysosomal positioning and degradative function. AREL1 acts by two manners: interacting with the $V_o$a subunit of the V-ATPase to establish ER–lysosome MCSs, and catalyzing K33-linked polyubiquitylation of the $V_1$B2 subunit that then binds to ER-resident UBAC2 to confer perinuclear localization of lysosomes. The deubiquitylating enzyme ZRANB1 by counteracting AREL1-mediated $V_1$B2 ubiquitylation reduces ER–lysosome MCSs and releases lysosomes to the cell periphery. Moreover, disruption of the AREL1−UBAC2−V-ATPase axis not only impairs lysosomal perinuclear clustering but also compromises lysosomal pH and degradative capacity. In mice, knockout of *Arel1* induces age-dependent Purkinje cell loss and motor impairments, with lipofuscin granules accumulated in Purkinje cells well before the onset of ataxic phenotype. Our results highlight the importance of ER-mediated lysosomal positioning and degradative function in neuronal health and disease.

# Results

## AREL1 is an ER-localized, phase-separated protein and interacts with the $V_o$a subunit of the V-ATPase

We first sought to identify proteins at ER–lysosome MCSs using the split-TurboID in proximity labeling strategy (Cho et al, 2020), in which the amino- and carboxyl-terminal fragments of TurboID—an engineered *E. coli* biotin ligase—were fused to LAMP1-mCherry and EGFP-Sec61β, respectively (Fig. 1A). When ER and lysosomes were closely apposed, the two inactive fragments reconstituted into a functional enzyme that conjugated nearby proteins with exogenously added biotin. As proof of the strategy, fluorescently labeled streptavidin was detected at the interfaces between the ER and lysosomes in U2OS cells, a human osteosarcoma cell line (Fig. EV1A). We next performed the split-TurboID in proximity labeling experiment coupled to mass spectrometry in HEK293T cells owing to relatively high transfection efficiency (Fig. EV1B). The subcellular fractionation assay confirmed the enrichment of biotinylated proteins in the membrane fractions (Fig. EV1C). These biotinylated membrane proteins were then subjected to streptavidin affinity purification followed by mass spectrometry analysis. Of all the proteins identified (Dataset EV1), 85 harboring transmembrane domain(s) were profoundly enriched in cells transfected with both LAMP1-mCherry-Tb(N) and Tb(C)-EGFP-Sec61β compared with cells transfected with either one. Proteins reported to localize in the membranes other than the ER or lysosomes, or with well-established functions, were further excluded. Among 18 remaining candidates, most were lysosomal or ER proteins (Fig. EV1D and Dataset EV1), including those known to mediate the formation of ER–lysosome MCSs such as VAPA (Rocha et al, 2009), MOSPD2 (Di Mattia et al, 2018), and protrudin (Raiborg et al, 2015). AREL1 is of particular interest given its uncharacterized subcellular localization. It is a HECT-type E3 ligase reported to mediate degradation of mitochondrial proapoptotic proteins and pro-interleukin-1β precursor (Kim et al, 2013; Mishra et al, 2023).

We then examined the effects of each of 18 candidates on ER–lysosome MCSs using shRNA-mediated knockdown (Fig. EV1E) followed by proximity ligation assay (PLA) in U2OS cells. PLA allows the detection of two proteins at a distance of less than 40 nm and provides a quantitative measure of ER–lysosome MCSs (Lim et al, 2019; Saric et al, 2021; Soderberg et al, 2006). Silencing of *AREL1* and several other genes (*ATP6AP1*, *ATP6AP2*, *VAPA*, *VMA21*, *UBE2J1*, *MOSPD2*, *ZFYVE27*, and *TMEM9*) significantly reduced the percentage of lysosomes in contacts with the ER (Fig. EV1F,G). RNF26 is a previously identified ER-resident ubiquitin ligase that tethers the endolysosomal system in the perinuclear region (Cremer et al, 2021; Cremer et al, 2023; Jongsma et al, 2016). Depletion of *RNF26* caused a 24% reduction in PLA puncta in U2OS cells (Fig. EV1F,G).

We also performed the split-Turbo ID in proximity labeling assay using *AREL1* knockout (KO) U2OS cells (Fig. EV1H,I) as an unbiased confirmation of our screening. The total levels of biotinylated proteins were reduced in *AREL1* KO cells (Fig. EV1J). The protein levels of VAPA in the pellets were decreased as well. These results suggest that AREL1 is responsible for establishing ER–lysosome MCSs. Knockout of *AREL1* in U2OS cells and knockdown of *AREL1* in HEK293T cells markedly reduced PLA puncta for LAMP1–calnexin interactions (Fig. EV1K–O).

AREL1 has a transmembrane domain at the very amino terminus, with the rest lying in the cytosol (Fig. 1B). It was mostly present in the ER-enriched membrane fractions in density gradient centrifugation (Fig. EV2A). Ectopically expressed AREL1 showed a perinuclear concentration that corresponded well with the ER marker Sec61β while encompassing LAMP1-positive lysosomes (Fig. 1C). The close apposition of endogenous AREL1 protein to lysosomes (Fig. 1D) was confirmed using an antibody whose specificity had been validated using *AREL1* KO cells (Fig. EV1H,I). The antibody recognizes the carboxyl-terminal HECT domain of AREL1, and therefore failed to detect AREL1 in the purified membrane fractions exposed to trypsin regardless of the presence or absence of detergent (Fig. EV2B). However, deletion of the entire C-terminal HECT domain (AREL1(1–482)) completely dissociated AREL1 from lysosomes (Fig. EV2C–E), whereas inactivation of the ubiquitin ligase activity (AREL1(C790A)) failed to do so (Fig. 1E,F). Using a series of truncations of the HECT domain (Fig. EV2C), we found that the flexible hinge region of six amino acids ($G_{706}$TGDIS), which connects the N-lobe and the C-lobe of the HECT domain (Singh et al, 2019), was responsible for the association of AREL1 with lysosomes (Figs. EV2C–E and 1E,F).

The puncta pattern of AREL1 prompted us to investigate whether AREL1 could undergo phase separation. Indeed, there is an intrinsically disordered region (IDR, amino acids 313–345) followed by a low-complexity region (LCR, amino acids 346–360) between the transmembrane domain and the HECT domain (Fig. 1B). The purified mEGFP-LCR protein, but not mEGFP-IDR protein, formed phase-separated droplets in vitro (Fig. 1G). The sequence of LCR is highly conserved among species (Fig. 1H), and harbors six cationic amino acids (K346, K348, K349, K351, K352, and K360) and two aromatic amino acids (Y354 and Y356). Deletion of lysine residues preceding tyrosine residues (△5 K) or replacement of two tyrosine residues with alanine (Y354A/Y356A) disrupted phase separation of mEGFP-LCR protein, whereas substitutions tyrosine for tryptophan (Y354W/Y356W) or phenylalanine (Y354F/Y356F) had no effects (Fig. 1I), suggesting that the cation-π interaction between lysine and aromatic amino acids

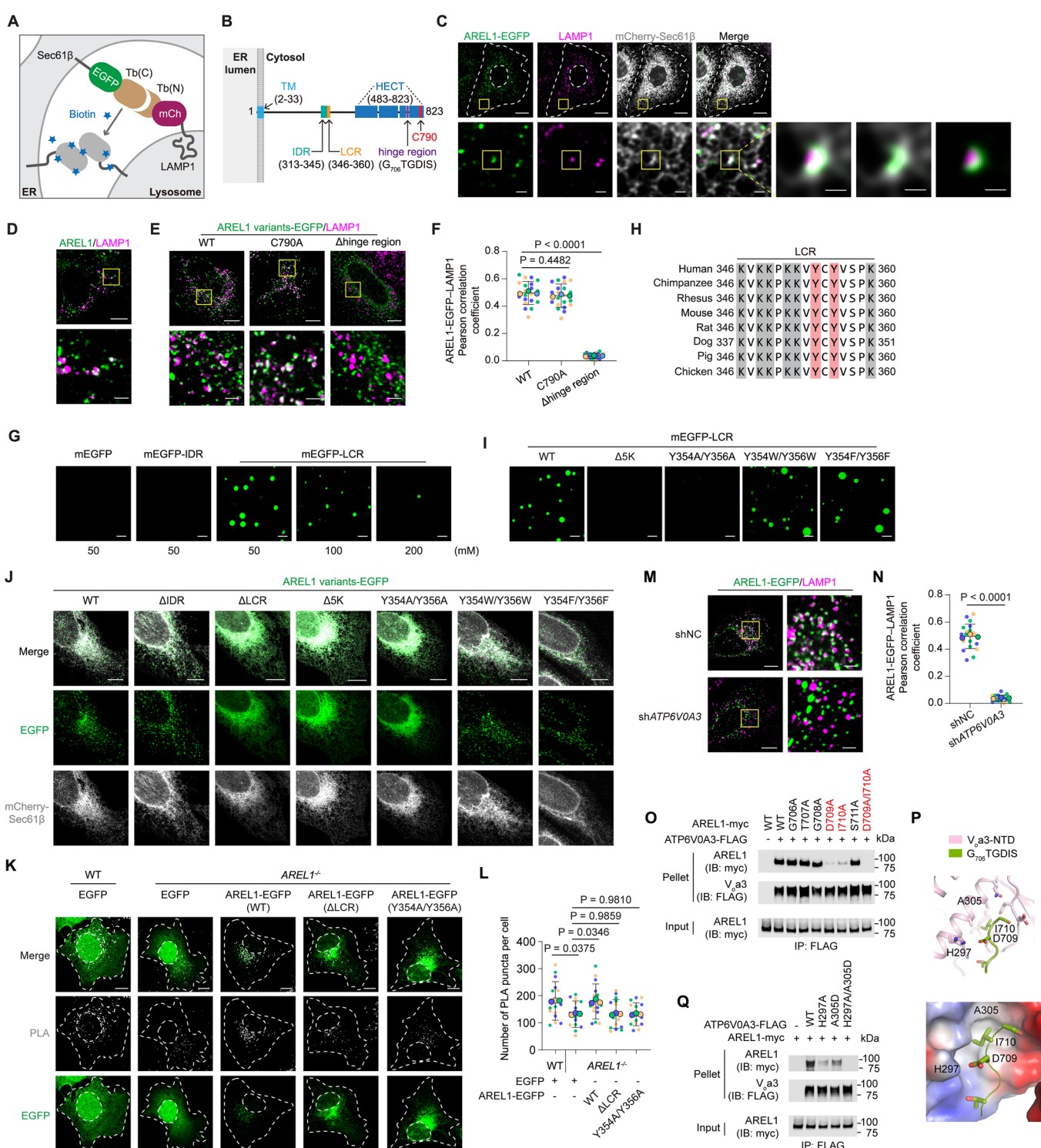

drives phase separation of AREL1. Consistent with the in vitro results, deletion of LCR or lysine residues within LCR as well as substitution of tyrosine residues for alanine resulted in a punctiform-to-meshwork redistribution of AREL1 that coincided with the pattern of the ER marker (Fig. 1J). Notably, re-expression of the mutants defective in phase separation failed to revert the percentage of lysosomes in contact with the ER, as indexed by PLA

puncta, as that of the wild-type (WT) protein did in *AREL1* KO cells (Fig. 1K,L).

To identify the interacting partners of AREL1 on the lysosomal membrane, we employed the split-TurboID in proximity labeling strategy again, with the two enzyme halves fused with AREL1 and LAMP1, respectively (Fig. EV3A). A total of 25 endolysosomal proteins (LAMP1 as the bait was not counted) were significantly

◄

**Figure 1.  AREL1 is an ER-localized, condensed, single transmembrane protein mediating ER–lysosome membrane contact formation through binding to the $V_o$a subunit of V-ATPase.**

(A) Schematic illustration of the split-TurboID-based in proximity labeling assay to identify proteins at ER–lysosome membrane contact sites. Tb(N) TurboID (N terminus), Tb(C) TurboID (C terminus), mCh mCherry. (B) Schematic illustration of the human AREL1 protein. TM transmembrane domain, IDR intrinsically disordered region, LCR low-complexity region, HECT homologous to E6AP C-terminus. C790 is the catalytically active site. (C) Representative confocal images showing AREL1 localization at ER–lysosome membrane contact sites. U2OS cells (a human osteosarcoma cell line) were transduced with lentiviruses expressing AREL1-EGFP and mCherry-Sec61β and then immunostained with anti-LAMP1 antibody. Cell contour and nucleus are outlined using white dashed lines. Boxed areas are enlarged sequentially (first vertically and then horizontally). Scale bars, 10 μm (top row), 2 μm (bottom row), 1 μm (right panel). (D) Representative confocal images showing the apposition of endogenous AREL1 protein to lysosomes. U2OS cells were immunostained with anti-AREL1 and anti-LAMP1 antibodies. The boxed area is enlarged and shown at the bottom. Scale bars, 10 μm (main), 2 μm (inset). (E) Representative confocal images showing the localization of indicated AREL1 variants relative to lysosomes. U2OS cells were transduced with lentiviruses expressing the indicated AREL1 variants tagged with EGFP and immunostained with anti-LAMP1 antibody. Boxed areas are enlarged and shown at the bottom. Scale bars, 10 μm (main), 2 μm (inset). WT wild-type. (F) Superplots showing Pearson's correlation coefficient for AREL1 variants and LAMP1 per cell (small dots) and its mean per independent experiment (large dots). Means and error bars (SD) are shown as black bars. # of cells: AREL1(WT)-EGFP, 21; AREL1(C790A)-EGFP, 21; and AREL1(Δhinge region)-EGFP, 21; from three independent experiments. Unpaired two-tailed Student's *t* test. Pearson's correlation coefficient for LAMP1 and AREL1(WT)-EGFP vs Pearson's correlation coefficient for LAMP1 and AREL1(C790A)-EGFP, $P = 0.4482$; Pearson's correlation coefficient for LAMP1 and AREL1(WT)-EGFP vs Pearson's correlation coefficient for LAMP1 and AREL1(Δhinge region)-EGFP, $P < 0.0001$. (G) Representative images showing droplet formation of purified recombinant proteins in the presence of the indicated concentrations of NaCl. Scale bars, 10 μm. (H) Sequence alignment of the low-complexity region (LCR) of AREL1 from the indicated species. Cationic and aromatic amino acids are in gray and red shadows, respectively. (I) Representative images showing droplet formation of indicated LCR variants in the presence of 50 mM NaCl. Scale bars, 10 μm. Δ5 K, LCR with lysine residues at positions 346, 348, 349, 351, and 352 all deleted. (J) Representative confocal images showing the localization of indicated AREL1 variants relative to the ER. U2OS cells were transduced with lentiviruses expressing the indicated AREL1 variants tagged with EGFP and mCherry-Sec61β. Scale bars, 10 μm. Δ5 K, LCR with lysine residues at positions 346, 348, 349, 351, and 352 all deleted. (K) Representative confocal images showing WT and *AREL1*⁻/⁻ U2OS cells transduced with lentiviruses expressing indicated proteins and immunostained with anti-calnexin and anti-LAMP1 antibodies, followed by proximity ligation assay. Cell contour and nucleus are outlined using white dashed lines. Scale bars, 10 μm. (L) Superplots showing the number of PLA puncta per cell (small dots) and its mean per independent experiment (large dots). Means and error bars (SD) are shown as black bars. # of cells: WT cells expressing EGFP, 15; *AREL1*⁻/⁻ cells expressing EGFP, 15; *AREL1*⁻/⁻ cells expressing AREL1(WT)-EGFP, 15; *AREL1*⁻/⁻ cells expressing AREL1(ΔLCR)-EGFP, 15 and *AREL1*⁻/⁻ cells expressing AREL1(Y354A/Y356A)-EGFP, 15; from three independent experiments. Unpaired two-tailed Student's *t* test. WT cells expressing EGFP vs *AREL1*⁻/⁻ cells expressing EGFP, $P = 0.0375$; *AREL1*⁻/⁻ cells expressing EGFP vs *AREL1*⁻/⁻ cells expressing AREL1(WT)-EGFP, $P = 0.0346$; *AREL1*⁻/⁻ cells expressing EGFP vs *AREL1*⁻/⁻ cells expressing AREL1(ΔLCR)-EGFP, $P = 0.9859$; *AREL1*⁻/⁻ cells expressing EGFP vs *AREL1*⁻/⁻ cells expressing AREL1(Y354A/Y356A)-EGFP, $P = 0.9810$. (M) Representative confocal images showing the localization of AREL1-EGFP relative to lysosomes in U2OS cells transduced with lentiviruses encoding negative control shRNA (shNC) and shRNA against *ATP6V0A3* (shATP6V0A3). Boxed areas are enlarged and shown on the right. Scale bars, 10 μm. (N) Superplots showing Pearson's correlation coefficient for AREL1 variants and LAMP1 per cell (small dots) and its mean per independent experiment (large dots). Means and error bars (SD) are shown as black bars. # of cells: shNC, 21 and shATP6V0A3, 21; from three independent experiments. Unpaired two-tailed Student's *t* test. shNC cells vs shATP6V0A3 cells, $P < 0.0001$. (O) Co-immunoprecipitation (IP) analysis showing the interaction between AREL1 variants and the $V_o$a3 subunit of V-ATPase. HEK293T cells were transfected as indicated and subjected to IP with anti-FLAG beads. (P) AlphaFold 3-predicted interaction of AREL1 hinge region ($G_{706}$TGDIS) (green) and the amino-terminal domain (NTD) of the $V_o$a3 subunit (pink). ipTM = 0.32, pTM = 0.79. (Q) Co-IP analysis showing the interaction between $V_o$a3 variants and AREL1. HEK293T cells were transfected as indicated and subjected to IP with anti-FLAG beads. Source data are available online for this figure.

enriched following biotin treatment, and 7 among them were the subunits of V-ATPase and highly ranked (Fig. EV3B; Dataset EV2). Many ER proteins were enriched as well (Fig. EV3B; Dataset EV2).

The composition of mammalian V-ATPase is shown in Fig. EV3C. Many subunits have multiple isoforms that express in cell-, tissue-, and organelle-specific manners (Toei et al, 2010). All four isoforms of subunit $V_o$a were co-immunoprecipitated with AREL1 (Fig. EV3D). These isoforms share a homologous amino-terminal domain (NTD) (Fig. EV3E). It was this NTD and the hinge region ($G_{706}$TGDIS) that mediated the interaction between $V_o$a and AREL1 (Fig. EV3F,G). The NTDs of all four $V_o$a isoforms were predicted to interact with the hinge region in a similar manner by AlphaFold 3 (Fig. EV3H).

We further sought to map the exact amino acid(s) mediating the interaction between the $V_o$a subunit and the hinge region of AREL1 by alanine-scanning mutagenesis. The $V_o$a3 isoform (encoded by ATP6V0A3) was chosen as a representative because it was highly expressed in U2OS cells (Fig. EV3I), and knockdown of *ATP6V0A3* (Fig. EV3J,K) almost completely disrupted AREL1 association with lysosomes (Fig. 1M,N). The co-immunoprecipitation (IP) of AREL1 by $V_o$a3 was nearly completely eliminated by D709A mutation in the hinge region and profoundly reduced by I710A mutation, with an even greater effect observed for the double-site mutant (Fig. 1O). According to the AlphaFold 3 modeling, the D709 residue of the hinge region forms a hydrogen bond with the H297 residue of $V_o$a3, and the I710 residue inserts into a

hydrophobic groove (Fig. 1P). Consistently, single and double mutations of the H297 and A305 residues of $V_o$a3 markedly abrogated its interaction with AREL1 (Fig. 1Q).

Together, the above results suggest that ER-resident AREL1 establishes ER–lysosome MCSs by interacting with the $V_o$a subunit of V-ATPase.

## AREL1 regulates the perinuclear positioning of lysosomes

The effects of AREL1 on the ER and lysosomes were next evaluated. No apparent alterations in ER morphology or lysosome numbers were detected in *AREL1* KO U2OS cells (Fig. EV4A,B). Compared with WT cells whose lysosomes were mostly clustered in the perinuclear region, *AREL1* KO cells showed a dispersed distribution of lysosomes throughout the cytosol (Fig. 2A), as indicated by increased distances between the nuclear envelope and lysosomal center (Fig. 2B,C). The scattering of lysosomes caused by AREL1 deficiency was completely reverted following re-expression of WT AREL1 but not the catalytically inactive C790A mutant or AREL1 lacking the hinge region, LCR region, or carrying Y354A/Y356A mutations (Fig. 2D,E). Knockdown of AREL1 binding partner *ATP6V0A3* similarly caused lysosome dispersal in U2OS cells (Fig. 2F,G). The spatial distribution of EEA1-positive early endosomes remained unaltered despite the absence of AREL1 (Fig. EV4C,D). We also generated *AREL1* KO HeLa cells (Fig. EV4E,F) and found a redistribution of lysosomes but not

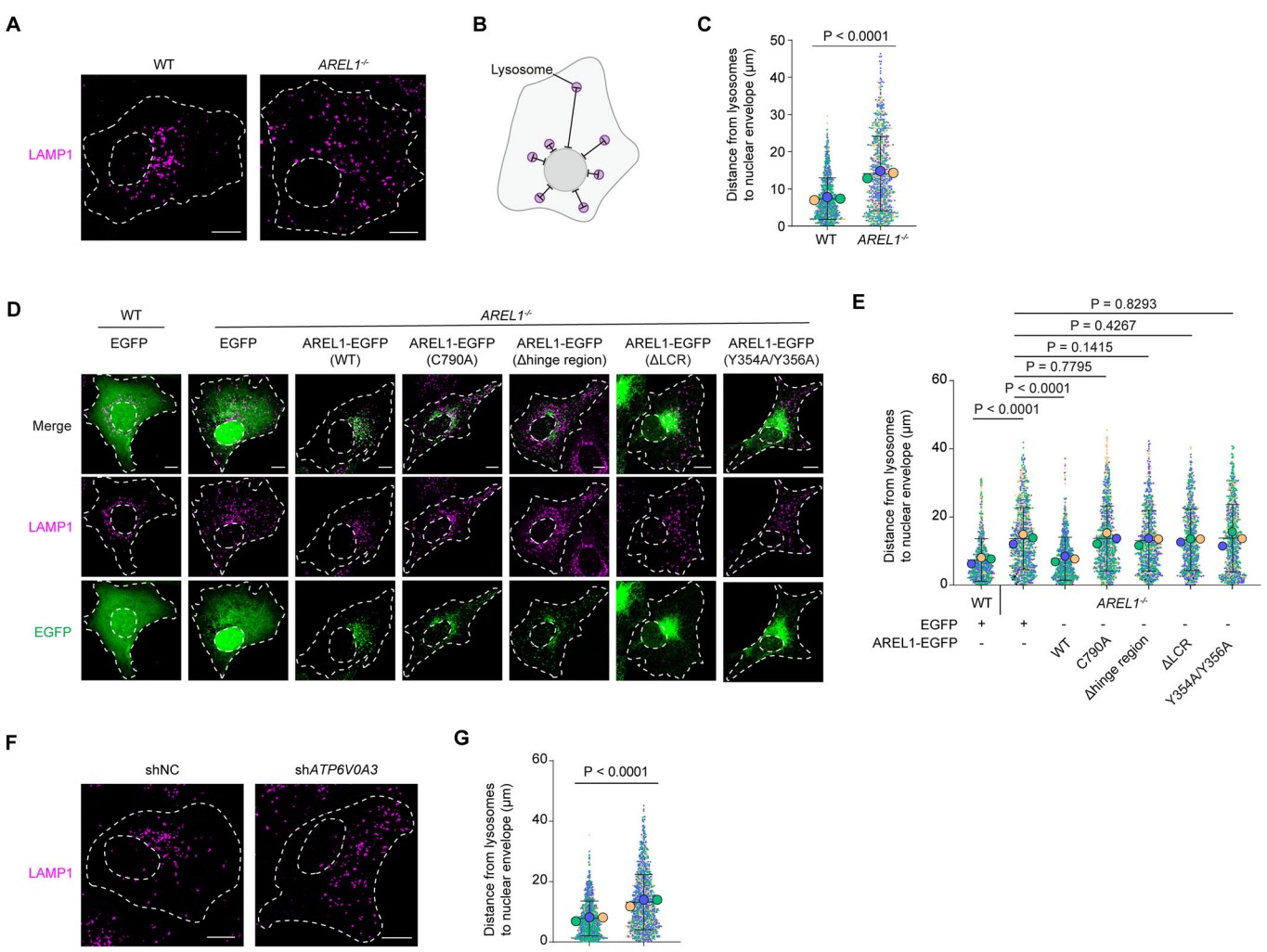

**Figure 2. AREL1 regulates lysosomal positioning.**

(A) Representative confocal images showing the distribution of lysosomes in WT and AREL1⁻/⁻ U2OS cells. Cell contour and nucleus are outlined using white dashed lines. Scale bars, 10 μm. (B) Schematic illustration showing distance measurement between lysosomes and the nuclear envelope. (C) Superplots showing the distance from lysosomes to the nuclear envelope (small dots) and its mean per independent experiment (large dots). Means and error bars (SD) are shown as black bars. # of cells (# of lysosomes): WT, 15 (1076) and AREL1⁻/⁻, 13 (1026); from three independent experiments. Mann–Whitney U test. WT cells vs AREL1⁻/⁻ cells, P < 0.0001. (D) Representative confocal images showing WT and AREL1⁻/⁻ U2OS cells transduced with lentiviruses expressing indicated proteins and immunostained with anti-LAMP1 antibody. Cell contour and nucleus are outlined using white dashed lines. Scale bars, 10 μm. (E) Superplots showing the distance from lysosomes to the nuclear envelope (small dots) and its mean per independent experiment (large dots). Means and error bars (SD) are shown as black bars. # of cells (# of lysosomes): WT cells expressing EGFP, 9 (862); AREL1⁻/⁻ cells expressing EGFP, 9 (950); AREL1⁻/⁻ cells expressing AREL1(WT)-EGFP, 9 (801); AREL1⁻/⁻ cells expressing AREL1(C790A)-EGFP, 9 (948); AREL1⁻/⁻ cells expressing AREL1(Δhinge region)-EGFP, 9 (911); AREL1⁻/⁻ cells expressing AREL1(ΔLCR)-EGFP, 9 (882) and AREL1⁻/⁻ cells expressing AREL1(Y354A/Y356A)-EGFP, 9 (838); from three independent experiments. Mann–Whitney U test. WT cells expressing EGFP vs AREL1⁻/⁻ cells expressing EGFP, P < 0.0001; AREL1⁻/⁻ cells expressing EGFP vs AREL1⁻/⁻ cells expressing AREL1(WT)-EGFP, P < 0.0001; AREL1⁻/⁻ cells expressing EGFP vs AREL1⁻/⁻ cells expressing AREL1(C790A)-EGFP, P = 0.7795; AREL1⁻/⁻ cells expressing EGFP vs AREL1⁻/⁻ cells expressing AREL1(Δhinge region)-EGFP, P = 0.1415; AREL1⁻/⁻ cells expressing EGFP vs AREL1⁻/⁻ cells expressing AREL1(ΔLCR)-EGFP, P = 0.4267; AREL1⁻/⁻ cells expressing EGFP vs AREL1⁻/⁻ cells expressing AREL1(Y354A/Y356A)-EGFP, P = 0.8293. (F) Representative confocal images showing the distribution of lysosomes in U2OS cells transduced with lentiviruses encoding negative control shRNA (shNC) and shRNA against ATP6V0A3 (shATP6V0A3). Cell contour and nucleus are outlined using white dashed lines. Scale bars, 10 μm. (G) Superplots showing the distance from lysosomes to the nuclear envelope (small dots) and its mean per independent experiment (large dots). Means and error bars (SD) are shown as black bars. # of cells (# of lysosomes): shNC, 20 (1357) and shATP6V0A3, 17 (1370); from three independent experiments. Mann–Whitney U test. shNC cells vs shATP6V0A3 cells, P < 0.0001. Source data are available online for this figure.

early endosomes to the cell periphery (Fig. EV4G–J). These results suggest that AREL1 is critical for the perinuclear localization of lysosomes, and that E3 ubiquitin ligase activity, lysosome contacts, and phase separation property of AREL1 are all indispensable for its regulation of lysosomal distribution.

## AREL1 deficiency impairs the acidification and degradative capacity of lysosomes

Perinuclear lysosomes have been found to be more acidic than the peripheral ones (Johnson et al, 2016). By using the Oregon Green

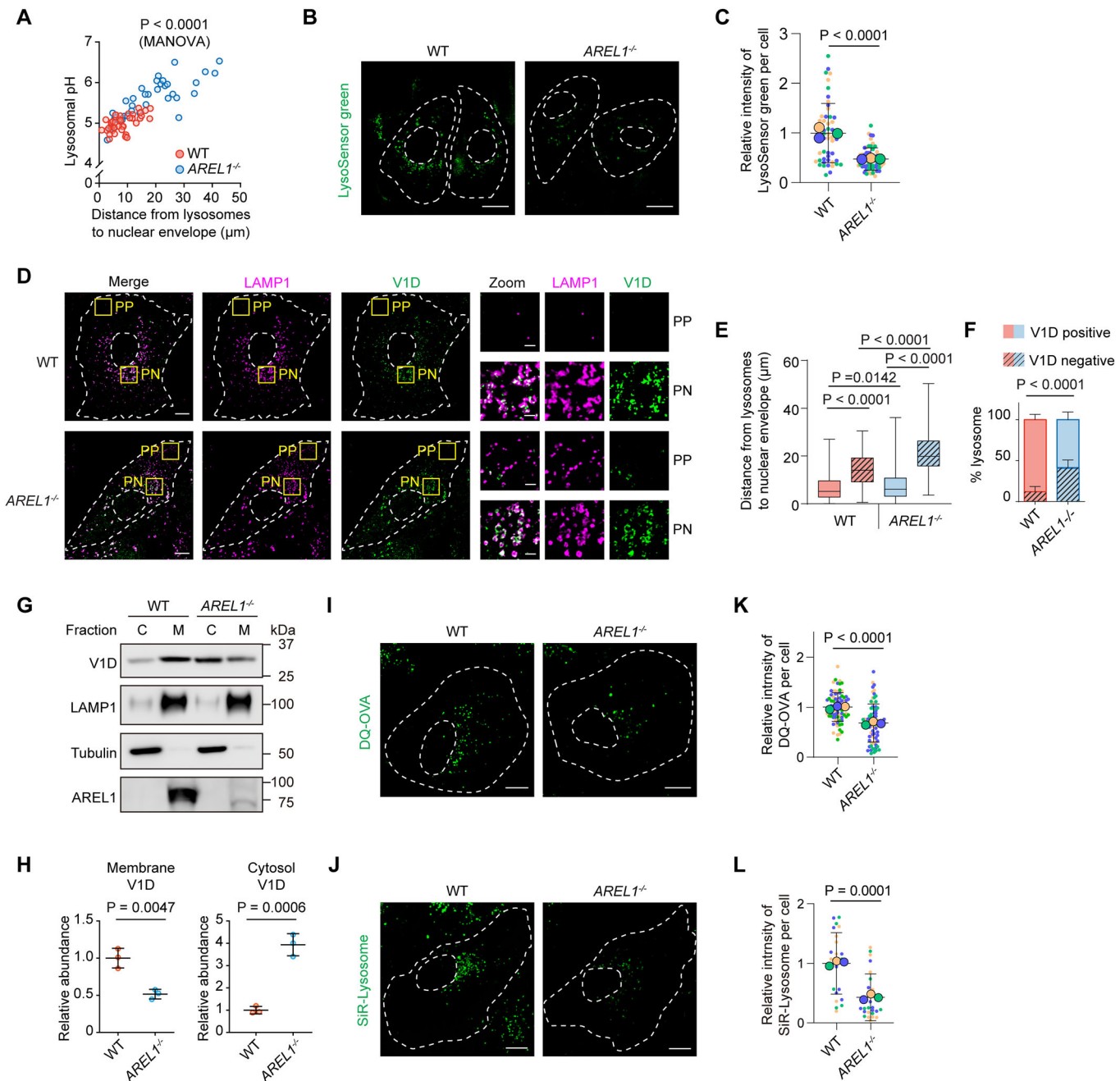

Dextran whose fluorescence emitted upon excitation near 490 nm is exquisitely pH-sensitive whereas that near 440 nm is not (DiCiccio and Steinberg, 2011), we showed that, in both WT and *AREL1* KO cells, lysosomal pH was elevated as lysosomes were further away from the nuclear envelope, and that *AREL1* KO cells had significantly more peripheral lysosomes with higher luminal pH (Fig. 3A). The overall fluorescence intensity of LysoSensor green, which stains the acidic organelles, was drastically reduced in *AREL1* KO cells (Fig. 3B,C).

The luminal pH of lysosomes is determined by the dynamic balance between proton influx through V-ATPase and proton efflux through TMEM175 (Hu et al, 2022), SLC7A11 (Zhou et al,

2025), as well as other transporters or channels. The $V_o$ and $V_1$ domains of V-ATPase are connected by the central stalk of D and F subunits (Fig. EV3C) (Wang et al, 2020). The association of $V_1D$ subunit with the lysosomal membrane protein LAMP1 provides a quantitative measure of intact, active V-ATPases. To better visualize membrane-associated $V_1$ domains, we permeabilized cells with digitonin prior to fixation so that free-floating $V_1$ domains could diffuse out of the cells. In both WT and *AREL1* KO cells, $V_1D$ was perinuclearly distributed and colocalized with most, if not all, perinuclear lysosomes (Fig. 3D). Lysosomes negative for $V_1D$ were more distant from the nucleus than $V_1D$-positive ones (Fig. 3E), suggesting that V-ATPases on the membrane of peripheral

◀ **Figure 3. AREL1 deficiency increases the number of peripheral lysosomes with elevated luminal pH, partially assembled V-ATPase, and reduced degradative potency.**

(A) The correlation of lysosomal pH with its distance to the nuclear envelope in WT and $AREL1^{-/-}$ U2OS cells. The effects of the independent variable (genotype) on two dependent variables (distance and pH) were analyzed using multivariate analysis of variance (MANOVA) in RStudio, and multivariate significance was assessed using Pillai's trace statistic ($n = 40$ lysosomes from WT U2OS cells and 36 lysosomes from $AREL1^{-/-}$ U2OS cells). WT cells vs $AREL1^{-/-}$ cells, $P < 0.0001$. (B) Representative confocal images showing WT and $AREL1^{-/-}$ U2OS cells stained with LysoSensor green. Cell contour and nucleus are outlined using white dashed lines. Scale bars, 20 μm. (C) Superplots showing the relative intensity of LysoSensor green (small dots) and its mean per independent experiment (large dots). Means and error bars (SD) are shown as black bars. # of cells: WT, 52 and $AREL1^{-/-}$, 59; from three independent experiments. Unpaired two-tailed Student's *t* test. WT cells vs $AREL1^{-/-}$ cells, $P < 0.0001$. (D) Representative confocal images showing the localization of $V_1D$ and LAMP1 in WT and $AREL1^{-/-}$ U2OS cells. Boxed areas are enlarged on the right. Cell contour and nucleus are outlined using white dashed lines. PP peripheral, PN perinuclear. Scale bars, 10 μm (main), 2 μm (inset). (E) Box plots showing the distance from $V_1D$-positive and $V_1D$-negative lysosomes to the nuclear envelope in WT and $AREL1^{-/-}$ U2OS cells. Data are presented as median with interquartile range. Each box-and-whisker consists of the 25th quantile (the upper border of box), median (horizontal line inside the box), 75th quantile (the lower border of box), and vertical lines extending to the minimum and maximum values. # of cells (# of lysosomes): WT ($V_1D$-positive), 14 (856); WT ($V_1D$-negative), 14 (112); $AREL1^{-/-}$ ($V_1D$-positive), 14 (770) and $AREL1^{-/-}$ ($V_1D$-negative), 14 (532); from three independent experiments. Mann–Whitney $U$ test. $V_1D$-positive lysosomes vs $V_1D$-negative lysosomes in WT cells, $P < 0.0001$; $V_1D$-positive lysosomes in WT cells vs $V_1D$-positive lysosomes in $AREL1^{-/-}$ cells, $P = 0.0142$; $V_1D$-positive lysosomes vs $V_1D$-negative lysosomes in $AREL1^{-/-}$ cells, $P < 0.0001$; $V_1D$-negative lysosomes in WT cells vs $V_1D$-negative lysosomes in $AREL1^{-/-}$ cells, $P < 0.0001$. (F) Percentages of lysosomes positive and negative for $V_1D$ in WT and $AREL1^{-/-}$ U2OS cells. Data are presented as means ± SD. # of cells: WT, 14 and $AREL1^{-/-}$, 14; from three independent experiments. Unpaired two-tailed Student's *t* test. $V_1D$-negative lysosomes in WT cells vs $V_1D$-negative lysosomes in $AREL1^{-/-}$ cells, $P < 0.0001$. (G) Immunoblotting analysis of cytosolic (C) and membrane (M) fractions from WT and $AREL1^{-/-}$ U2OS cells. (H) Quantification of the relative abundance of $V_1D$ in membrane and cytosolic fractions. Data are presented as means ± SD ($n = $ three independent experiments). Unpaired two-tailed Student's *t* test. $V_1D$ in membrane fractions of WT cells vs $V_1D$ in membrane fractions of $AREL1^{-/-}$ cells, $P = 0.0047$; $V_1D$ in cytosolic fractions of WT cells vs $V_1D$ in cytosolic fractions of $AREL1^{-/-}$ cells, $P = 0.0006$. (I) Representative confocal images showing WT and $AREL1^{-/-}$ U2OS cells incubated with DQ-OVA. Cell contour and nucleus are outlined using white dashed lines. Scale bars, 10 μm. (J) Representative confocal images showing WT and $AREL1^{-/-}$ U2OS cells stained with SiR-Lysosome. Cell contour and nucleus are outlined using white dashed lines. Scale bars, 10 μm. (K) Superplots showing the relative intensity of DQ-OVA (small dots) and its mean per independent experiment (large dots). Means and error bars (SD) are shown as black bars. # of cells: WT, 82 and $AREL1^{-/-}$, 77; from three independent experiments. Unpaired two-tailed Student's *t* test. WT cells vs $AREL1^{-/-}$ cells, $P < 0.0001$. (L) Superplots showing the relative intensity of SiR-Lysosome (small dots) and its mean per independent experiment (large dots). Means and error bars (SD) are shown as black bars. # of cells: WT, 23 and $AREL1^{-/-}$, 24; from three independent experiments. Unpaired two-tailed Student's *t* test. WT cells vs $AREL1^{-/-}$ cells, $P = 0.0001$. Source data are available online for this figure.

lysosomes are incomplete and therefore functionally incompetent. Depletion of *AREL1* significantly increased the percentage of $V_1D$-negative lysosomes as well as their distance to the nuclear envelope (Fig. 3E,F). In support of these results, $V_1D$ was predominantly associated with the membrane fractions isolated from WT cells but shifted to the cytosolic fractions when AREL1 was ablated (Fig. 3G,H).

The degradative potency of lysosomes was evaluated using two kinds of fluorogenic probes. DeQuenched ovalbumin (DQ-OVA) is a BODIPY-labeled substrate for lysosomal proteases that gives off fluorescence upon enzymatic digestion (Albrecht et al, 2020), whereas Silicon rhodamine (SiR)-Lysosome is a fluorophore-tagged pepstatin A that can bind specifically to active cathepsin D, the main acid hydrolase in the lysosome (Lukinavicius et al, 2016). Both probes emitted robust fluorescence in the perinuclear region of WT cells, suggesting perinuclear lysosomes have higher enzymatic activities than those in the peripheral region (Fig. 3I,J). The intensity of fluorescent signals was significantly less in *AREL1* KO cells (Fig. 3K,L).

These findings are consistent with the previous reports that peripheral lysosomes have increased pH (Johnson et al, 2016), reduced proteolytic capacity (Johnson et al, 2016; Korolchuk et al, 2011), and reduced amounts of the $V_1$ subunits of V-ATPase (Tang et al, 2021), underscoring an important role of AREL1 in regulating lysosomal perinuclear positioning and degradative capacity.

## AREL1 ubiquitylates the $V_1B2$ subunit of V-ATPase for binding to UBAC2 in the perinuclear ER

That the C790A mutant despite interacting with lysosomes (Fig. 1E,F) fails to anchor them perinuclearly (Fig. 2D,E) suggests a second pair of proteins, one being ubiquitylated by AREL1, is required for tethering lysosomes in the perinuclear region. We first investigated which subunit(s) of V-ATPase could be ubiquitylated by AREL1. The results showed that AREL1 selectively ubiquitylated the B2 isoform of the $V_1$ domain, among the examined V-ATPase subunits, in a catalytic activity-dependent manner (Figs. 4A and EV5A). Consistent with the findings that AREL1 catalyzes K33-linked polyubiquitylation (Kristariyanto et al, 2015; Michel et al, 2015), the K33R ubiquitin mutant was the only one out of seven K-to-R mutants that failed to confer AREL1-mediated ubiquitylation of $V_1B2$ (Fig. EV5B). The ubiquitin that only contains the lysine residue at position 33 (K33 only) was sufficient to support AREL1-mediated $V_1B2$ ubiquitylation (Fig. EV5C).

*ATP6V1B2* was highly expressed in U2OS cells (Fig. EV6A) and endogenous $V_1B2$ protein was ubiquitylated by AREL1 (Fig. 4B). We further immunoprecipitated endogenous $V_1B2$ protein from membrane and cytosolic fractions, respectively, and found only membrane-associated $V_1B2$ to be ubiquitylated (Fig. 4C). These results suggest that the $V_1B2$ subunit in intact V-ATPase complex, rather than the unassembled cytosolic $V_1B2$, is ubiquitylated by AREL1. In *ATP6V1B2* knockdown cells (Fig. EV6B,C), lysosomes were no longer restrained in the perinuclear region (Fig. 4D,E), and the numbers of calnexin-LAMP1 PLA puncta were reduced (Fig. EV6D,E). Three copies of $V_1B2$ and $V_1A$ constitute the $V_1$ domain that is responsible for ATP binding and hydrolysis (Collins and Forgac, 2020; Vasanthakumar and Rubinstein, 2020). The detrimental effects of *ATP6V1B2* deficiency on lysosomal acidity and degradative function were anticipated and therefore not examined.

ZRANB1 is a K29/K33-specific deubiquitylating enzyme (Licchesi et al, 2011). Overexpression of WT ZRANB1 but not catalytically inactive C443S mutant markedly reduced ubiquitin-positive signals associated with lysosomes (Fig. EV6F,G) and effectively counteracted $V_1B2$ ubiquitylation by AREL1 (Fig. 4F). Lysosomes were dispersed towards the cell periphery when WT

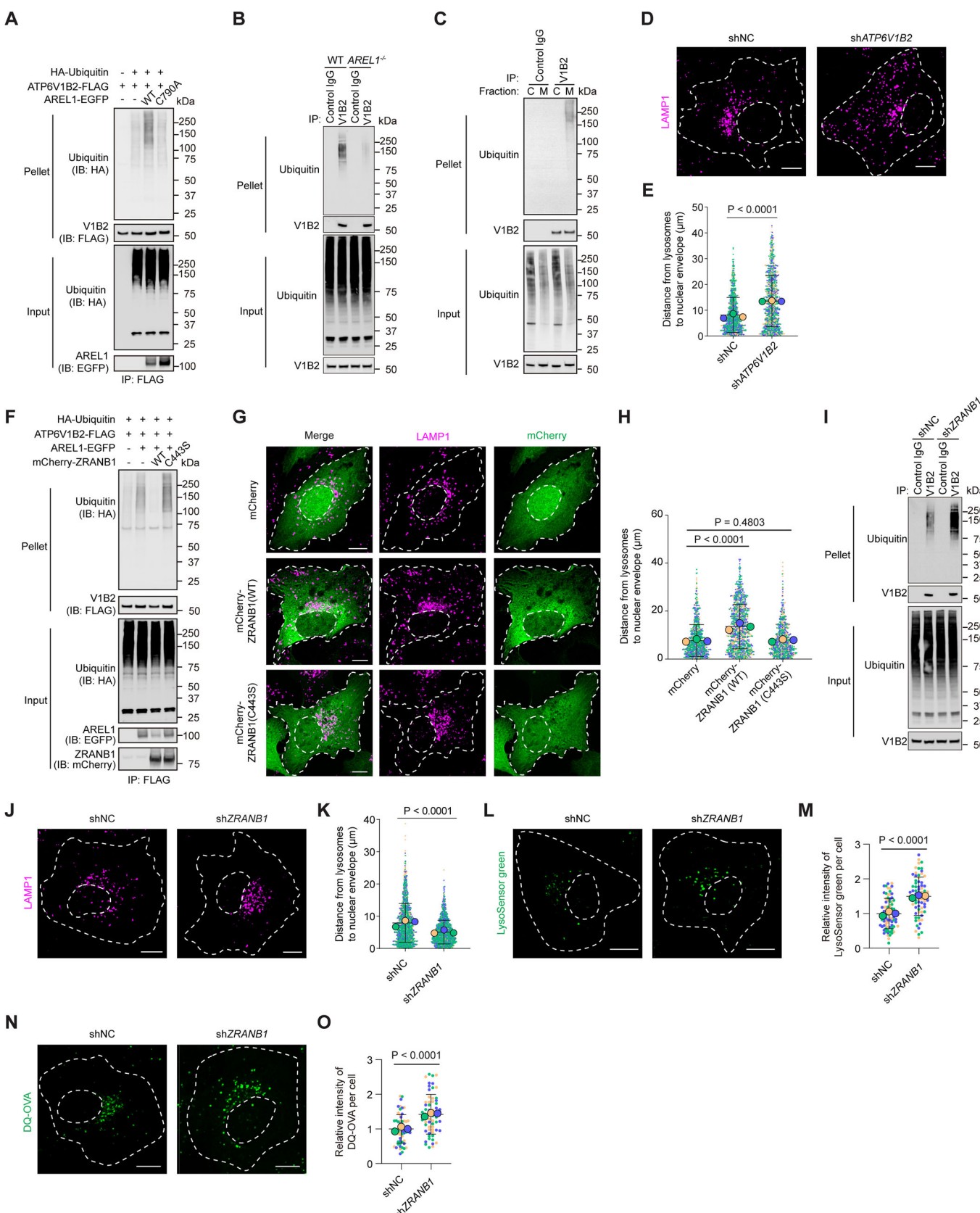

**Figure 4.  AREL1 and ZRANB1 regulate the ubiquitylation of the $V_1B2$ subunit of V-ATPase for lysosomal perinuclear positioning and degradative function.**

(A) HEK293T cells were transfected as indicated and subjected to immunoprecipitation (IP) with anti-FLAG beads followed by immunoblotting to analyze ubiquitylation. (B) WT and *AREL1*[-/-] U2OS cells were harvested and subjected to IP with control IgG beads or anti-$V_1B2$ beads, followed by immunoblotting to analyze ubiquitylation. (C) U2OS cells were harvested and subjected to subcellular fractionation. The cytosolic (C) and membrane (M) fractions were then subjected to IP with control IgG beads or anti-$V_1B2$ beads, followed by immunoblotting to analyze ubiquitylation. (D) Representative confocal images showing the distribution of lysosomes in U2OS cells transduced with lentiviruses encoding negative control shRNA (shNC) and shRNA against *ATP6V1B2* (sh*ATP6V1B2*). Cell contour and nucleus are outlined using white dashed lines. Scale bars, 10 μm. (E) Superplots showing the distance from lysosomes to the nuclear envelope (small dots) and its mean per independent experiment (large dots). Means and error bars (SD) are shown as black bars. # of cells (# of lysosomes): shNC, 12 (1101) and sh*ATP6V1B2*, 12 (1045); from three independent experiments. Mann–Whitney *U* test. shNC cells vs sh*ATP6V1B2* cells, *P* < 0.0001. (F) HEK293T cells were transfected as indicated and subjected to IP with anti-FLAG beads followed by immunoblotting to analyze ubiquitylation. (G) Representative confocal images showing U2OS cells transduced with lentiviruses expressing indicated mCherry-tagged ZRANB1 variants and immunostained with anti-LAMP1 antibody. Cell contour and nucleus are outlined using white dashed lines. Scale bars, 10 μm. (H) Superplots showing the distance from lysosomes to the nuclear envelope (small dots) and its mean per independent experiment (large dots). Means and error bars (SD) are shown as black bars. # of cells (# of lysosomes): mCherry, 12 (974), mCherry-ZRANB1 (WT), 10 (1006), and mCherry-ZRANB1 (C443S), 9 (727); from three independent experiments. Mann–Whitney *U* test. Cells expressing mCherry vs cells expressing mCherry-ZRANB1 (WT), *P* < 0.0001; Cells expressing mCherry vs cells expressing mCherry-ZRANB1 (C443S), *P* = 0.4803. (I) U2OS cells transduced with lentiviruses encoding negative control shRNA (shNC) and shRNA against *ZRANB1* (sh*ZRANB1*) were harvested and subjected to IP with control IgG beads or anti-$V_1B2$ beads, followed by immunoblotting to analyze ubiquitylation. (J) Representative confocal images showing the distribution of lysosomes in U2OS cells transduced with lentiviruses encoding negative control shRNA (shNC) and shRNA against *ZRANB1* (sh*ZRANB1*). Cell contour and nucleus are outlined using white dashed lines. Scale bars, 10 μm. (K) Superplots showing the distance from lysosomes to the nuclear envelope (small dots) and its mean per independent experiment (large dots). Means and error bars (SD) are shown as black bars. # of cells (# of lysosomes): shNC, 22 (1795) and sh*ZRANB1*, 23 (1759); from three independent experiments. Mann–Whitney *U* test. shNC cells vs sh*ZRANB1* cells, *P* < 0.0001. (L) Representative confocal images showing shNC and sh*ZRANB1* U2OS cells stained with LysoSensor green. Cell contour and nucleus are outlined using white dashed lines. Scale bars, 10 μm. (M) Superplots showing the relative intensity of LysoSensor green (small dots) and its mean per independent experiment (large dots). Means and error bars (SD) are shown as black bars. # of cells: shNC, 80 and sh*ZRANB1*, 73; from three independent experiments. Unpaired two-tailed Student's *t* test. shNC cells vs sh*ZRANB1* cells, *P* < 0.0001. (N) Representative confocal images showing shNC and sh*ZRANB1* U2OS cells incubated with DQ-OVA. Cell contour and nucleus are outlined using white dashed lines. Scale bars, 10 μm. (O) Superplots showing the relative intensity of DQ-OVA (small dots) and its mean per independent experiment (large dots). Means and error bars (SD) are shown as black bars. # of cells: shNC, 60 and sh*ZRANB1*, 60; from three independent experiments. Unpaired two-tailed Student's *t* test. shNC cells vs sh*ZRANB1* cells, *P* < 0.0001. Source data are available online for this figure.

ZRANB1 was overexpressed but remained perinuclearly following C443S mutant overexpression (Fig. 4G,H). To examine whether ZRANB1 could trim ubiquitin chains from $V_1B2$ protein in the endogenous context, we generated *ZRANB1* knockdown U2OS cells (Fig. EV6H) and indeed found substantially increased ubiquitylation of endogenous $V_1B2$ protein compared with control cells (Fig. 4I). The average distances between lysosomes and the nuclear envelope were reduced while the PLA signals of ER–lysosome MCSs were increased in *ZRANB1* knockdown cells (Figs. 4J,K and EV6I,J). Silencing of *ZRANB1* markedly increased the fluorescence intensities of LysoSensor green and DQ-OVA (Fig. 4L–O).

We hypothesized that the interaction between ubiquitylated $V_1B2$ and membrane-anchored ubiquitin-binding protein(s) might account for perinuclear localization of lysosomes. Hundreds of the proteins harboring ubiquitin-binding domains were analyzed for the presence of transmembrane domains. Ancient ubiquitous protein 1 (AUP1), rhomboid domain-containing protein 3 (RHBDD3), and ubiquitin-associated domain-containing protein 2 (UBAC2) were the three that fit the criteria (Fig. EV7A). Whereas AUP1 and RHBDD3 appeared as the cytoplasmic foci, UBAC2 was colocalized with the ER marker Sec61β and concentrated around the nucleus (Fig. EV7B). Knockdown of *UBAC2* dispersed lysosomes to the cell periphery, while that of *AUP1* or *RHBDD3* did not affect lysosomal positioning (Fig. EV7C–E).

UBAC2 has three transmembrane segments and a cytosolic ubiquitin-associated (UBA) domain for interacting with ubiquitylated proteins (Fig. 5A). The WT form of UBAC2 was co-immunoprecipitated with $V_1B2$, whereas the UBA-deleted mutant failed to do so (Fig. 5B). Knockout of *AREL1* completely abolished the interaction between UBAC2 and $V_1B2$, and complementing the WT form but not the catalytically inactive C790A mutant rescued UBAC2–$V_1B2$ interaction in *AREL1* KO cells (Fig. 5C). Moreover, overexpression of WT ZRANB1 instead of the inactive C443S

mutant abolished the interaction between UBAC2 and $V_1B2$ (Fig. 5D).

*UBAC2* depletion induced the peripheral distribution of lysosomes, and the phenotype was completely rescued by re-expression of WT UBAC2 but not the UBA domain-deleted, $V_1B2$-binding incompetent mutant (Fig. 5E,F). As seen in *AREL1* KO cells (Fig. 3), LysoSensor green signal was greatly reduced in *UBAC2* knockdown cells (Fig. 5G,H), together with significant increases in the percentage of $V_1D$-negative lysosomes and their distance from the nucleus (Fig. 5I–K). Depletion of *UBAC2* markedly attenuated the fluorescence of DQ-OVA and SiR-Lysosome (Fig. 5L,M). *UBAC2* knockdown cells displayed reduced numbers of calnexin-LAMP1 PLA puncta (Fig. EV7F,G).

Deficiency of *AREL1* did not affect the subcellular distribution of UBAC2 (Fig. EV7H). However, in *UBAC2* knockdown cells where lysosomes were peripherally dispersed (Fig. EV7D), WT AREL1 and the C790A mutant were redistributed to the cell periphery, whereas the hinge region-deleted AREL1 still stayed perinuclearly (Fig. EV7I). Since the hinge region-deleted AREL1 failed to interact with lysosomes (Fig. 1E,F), these results suggest that the peripheral distribution of AREL1 is actually conferred by that of lysosomes in *UBAC2* knockdown cells. The findings that AREL1 as an integral membrane protein can co-travel with lysosomes are not totally unexpected, since lysosomes can actively regulate ER structure and distribution (Lu et al, 2020).

To reconcile our model with rapid assembly/disassembly kinetics of V-ATPase, we performed fluorescence recovery after photobleaching (FRAP) experiments as described previously (Bodzeta et al, 2017; Sava et al, 2024). $V_1E1$ tagged with mCherry exhibited a robust punctate staining pattern that was colocalized with LAMP1-EGFP (Fig. EV8A), and was chosen as a measure for the recruitment of $V_1$ domain to the relatively immobile perinuclear lysosomes. The perinuclear regions of WT and *ZRANB1*

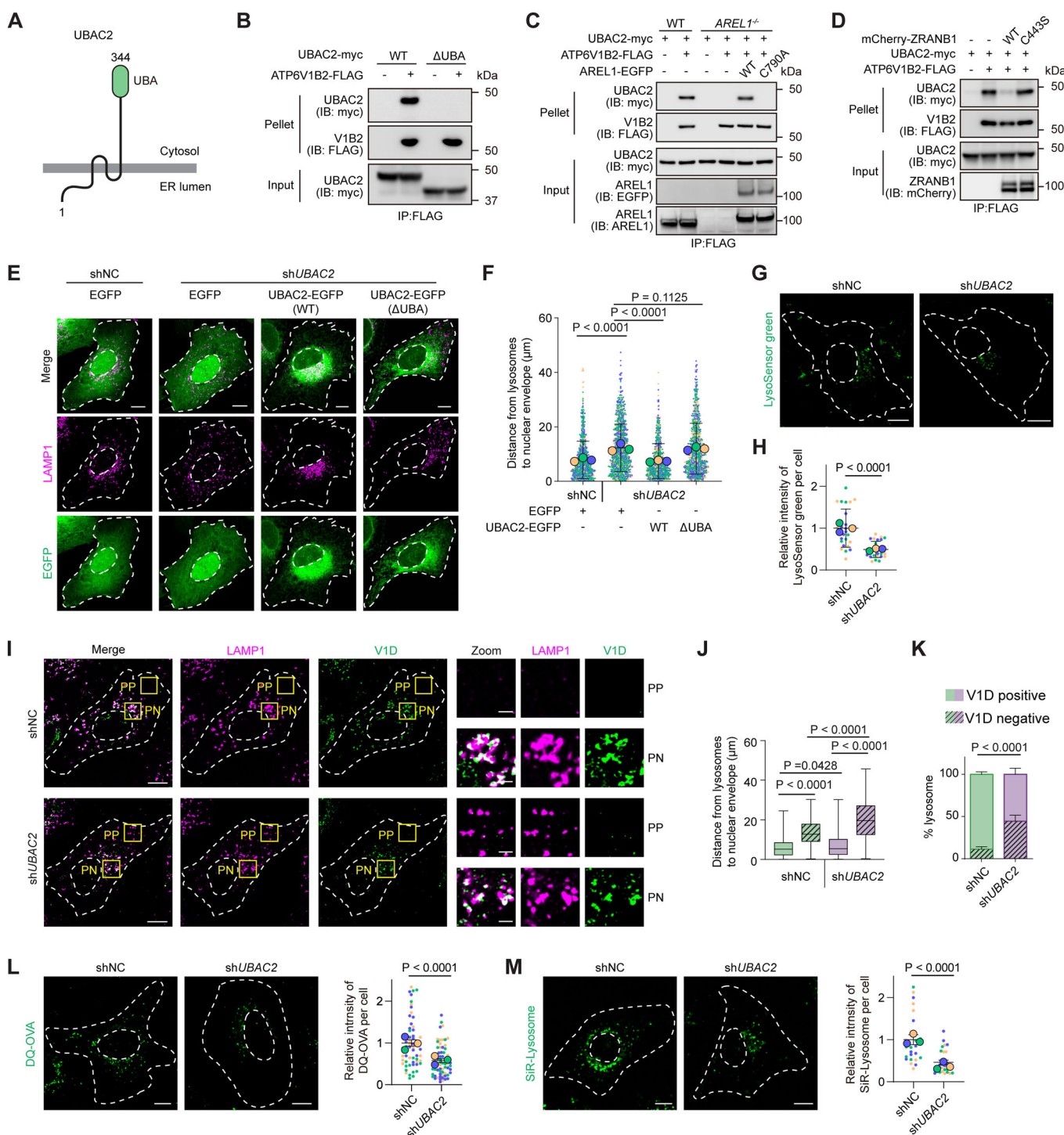

knockdown cells were photobleached, and the recovery of fluorescence was measured. The half-time of fluorescence recovery (τ1/2) was 24.46 s in WT cells and 44.06 s in *ZRANB1* knockdown cells (Fig. EV8B,C), indicating a delayed exchange of V₁E1 between the cytosolic pool and that bound to the lysosomal surface when *ZRANB1* was depleted. These results suggest that increased ubiquitylation of V₁B2 can help stabilize the V-ATPase holoenzyme and facilitate UBAC2-mediated perinuclear localization of

lysosomes. It should also be emphasized that the recovery time in our FRAP experiments was similar to that reported by the abovementioned previous studies (Bodzeta et al, 2017; Sava et al, 2024), suggesting that V-ATPase assembly is indeed rapid but still subjected to regulation by ubiquitylation/deubiquitylation.

Together, these results suggest that AREL1 functions in concert with UBAC2 to anchor lysosomes in the perinuclear region for optimal degradative capacity.

◀ **Figure 5. UBAC2 interacts with the ubiquitylated V₁B2 subunit and regulates lysosomal positioning and degradative function.**

(A) Schematic illustration of the human UBAC2 protein. UBA, ubiquitin-associated. (B) Co-immunoprecipitation (IP) analysis of HEK293T cells transfected as indicated. (C) Co-IP of WT and *AREL1$^{-/-}$* U2OS cells transfected as indicated. (D) Co-IP analysis of HEK293T cells transfected as indicated. (E) Representative confocal images showing the distribution of lysosomes in U2OS cells transduced with lentiviruses encoding negative control shRNA (shNC) and shRNA against *UBAC2* (sh*UBAC2*), transfected with the plasmids expressing indicated proteins, and immunostained with anti-LAMP1 antibody. Cell contour and nucleus are outlined using white dashed lines. Scale bars, 10 μm. (F) Superplots showing the distance from lysosomes to the nuclear envelope (small dots) and its mean per independent experiment (large dots). Means and error bars (SD) are shown as black bars. # of cells (# of lysosomes): shNC cells expressing EGFP, 11 (930), sh*UBAC2* cells expressing EGFP, 12 (1112), sh*UBAC2* cells expressing UBAC2(WT)-EGFP, 9 (808), and sh*UBAC2* cells expressing UBAC2(ΔUBA)-EGFP, 9 (974); from 3 independent experiments. Mann–Whitney $U$ test. shNC cells expressing EGFP vs sh*UBAC2* cells expressing EGFP, $P < 0.0001$; sh*UBAC2* cells expressing EGFP vs sh*UBAC2* cells expressing UBAC2(WT)-EGFP, $P < 0.0001$; sh*UBAC2* cells expressing EGFP vs sh*UBAC2* cells expressing UBAC2(ΔUBA)-EGFP, $P = 0.1125$. (G) Representative confocal images showing LysoSensor green in control and *UBAC2* knockdown U2OS cells. Cell contour and nucleus are outlined using white dashed lines. Scale bars, 20 μm. (H) Superplots showing the relative intensity of LysoSensor green (small dots) and its mean per independent experiment (large dots). Means and error bars (SD) are shown as black bars. # of cells: shNC, 26 and sh*UBAC2*, 27; from three independent experiments. Unpaired two-tailed Student's $t$ test. shNC cells vs sh*UBAC2* cells, $P < 0.0001$. (I) Representative confocal images showing the localization of V₁D and LAMP1 in U2OS cells transduced with lentiviruses encoding negative control shRNA (shNC) and shRNA against *UBAC2* (sh*UBAC2*). Cell contour and nucleus are outlined using white dashed lines. Boxed areas are enlarged on the right. Scale bars, 10 μm (main), 2 μm (inset). (J) Box plots showing the distance from V₁D-positive and V₁D-negative lysosomes to the nuclear envelope in shNC and sh*UBAC2* U2OS cells. Data are presented as median with interquartile range. Each box-and-whisker consists of the 25th quantile (the upper border of box), median (horizontal line inside the box), 75th quantile (the lower border of box), and vertical lines extending to the minimum and maximum values. # of cells (# of lysosomes): shNC (V₁D-positive), 11 (713); shNC (V₁D-negative), 11 (91); sh*UBAC2* (V₁D-positive), 11 (581) and sh*UBAC2* (V₁D-negative), 11 (454); from three independent experiments. Mann–Whitney $U$ test. V₁D-positive lysosomes vs V₁D-negative lysosomes in shNC cells, $P < 0.0001$; V₁D-positive lysosomes in shNC cells vs V₁D-positive lysosomes in sh*UBAC2* cells, $P = 0.0428$; V₁D-positive lysosomes vs V₁D-negative lysosomes in sh*UBAC2* cells, $P < 0.0001$; V₁D-negative lysosomes in shNC cells vs V₁D-negative lysosomes in sh*UBAC2* cells, $P < 0.0001$. (K) Percentages of lysosomes positive and negative for V₁D in shNC and sh*UBAC2* U2OS cells. Data are presented as mean±SD. # of cells: shNC, 11 and sh*UBAC2*, 11; from three independent experiments. Unpaired two-tailed Student's $t$ test. V₁D-negative lysosomes in shNC cells vs V₁D-negative lysosomes in sh*UBAC2* cells, $P < 0.0001$. (L) Representative confocal images showing DQ-OVA in shNC and sh*UBAC2* U2OS cells. Cell contour and nucleus are outlined using white dashed lines. Scale bars, 10 μm. The right is superplots showing the relative intensity of DQ-OVA (small dots) and its mean per independent experiment (large dots). Means and error bars (SD) are shown as black bars. # of cells: shNC, 59 and sh*UBAC2*, 66; from three independent experiments. Unpaired two-tailed Student's $t$ test. shNC cells vs sh*UBAC2* cells, $P < 0.0001$. (M) Representative confocal images showing SiR-Lysosome in shNC and sh*UBAC2* U2OS cells. Cell contour and nucleus are outlined using white dashed lines. Scale bars, 10 μm. Right is superplots showing the relative intensity of SiR-Lysosome (small dots) and its mean per independent experiment (large dots). Means and error bars (SD) are shown as black bars. # of cells: shNC, 24 and sh*UBAC2*, 27; from three independent experiments. Unpaired two-tailed Student's $t$ test. shNC cells vs sh*UBAC2* cells, $P < 0.0001$. Source data are available online for this figure.

## AREL1 deficiency causes age-associated neurodegeneration in mice

To investigate the role of AREL1 in animal physiology, we generated whole-body *Arel1* knockout mice (Fig. EV9A). *Arel1* knockout homozygotes (*Arel1$^{-/-}$*) were born at a normal Mendelian ratio and exhibited no gross abnormalities after birth. Surprisingly, *Arel1$^{-/-}$* mice at 12 months of age started to lose balance and showed circling behavior and head tilt (Fig. 6A; Movies EV1 and EV2). When lifted up by the tails, *Arel1$^{-/-}$* mice displayed a hindlimb clasping phenotype (Fig. 6B,C; Movies EV3 and EV4). To systematically assess the locomotor functions of *Arel1* mice, we subjected 12-month-old males and females for footprint analysis (Fig. 6D). Compared with age-matched controls, both male and female *Arel1$^{-/-}$* mice displayed aberrant gait patterns characterized by altered stride length, sway and stance (Fig. 6E). In the rotarod test, *Arel1$^{-/-}$* males and females spent significantly less time on the rod (Fig. 6F). It is interesting that the rotarod performance was worse as compared to footprint one in males, whereas footprint parameters were a bit more altered in females. These results suggest that *Arel1* deficiency causes late-onset motor impairment in mice.

According to several tissue expression databases [BIOGPS, Expression Atlas, TissueEnrich (Jain and Tuteja, 2019), JensenLab (Palasca et al, 2018), and Human Protein Atlas], AREL1 is widely expressed across mouse tissues, with a relatively higher level in the adult brain of mice and humans. Quantitative real-time PCR analysis confirmed *AREL1* expression in the mouse brain (Fig. EV9B). Using *Arel1$^{-/-}$* mice as negative controls for the antibody, we found the cerebellum was one of the brain regions expressing AREL1 protein (Fig. 6G). Further examination of cerebellar sections by immunohistochemistry showed that AREL1

was concentrated in Purkinje cells (Fig. 6H). In support of our findings, the in situ hybridization data retrieved from Allen Mouse Brain Atlas also reveal the high expression of *Arel1* that corresponds to calbindin-positive Purkinje cells (Fig. EV9C). In fact, by analyzing the previously published single-nucleus transcriptomics of mouse cerebellar cortex (Kozareva et al, 2021), we found that *Arel1*, *Zranb1*, *V-ATPase*, and *Ubac2* were highly expressed in Purkinje neurons (Fig. EV9D).

We detected no apparent abnormalities in the cerebral cortex or hippocampus of 12-month-old *Arel1$^{-/-}$* mice (Fig. EV9E). However, *Arel1$^{-/-}$* cerebellum had significantly less Purkinje cells compared with WT controls (Fig. 6I,J). Lipofuscin is composed of undigested remnants, including proteins, lipids, carbohydrates, and other cell materials due to lysosomal dysfunction over time (Heinsen, 1979; Sobaniec-Lotowska, 2001). We observed many more lipofuscin-positive puncta in the residual Purkinje cells of *Arel1$^{-/-}$* mice (Fig. 6K,L).

Under the electron microscope, lysosomes appeared as spherical organelles with uniformly dense matrices (Fig. 6M, Box A and yellow dots in Fig. EV9F), and lipofuscin granules were highly osmiophilic structures irregular in shape and size and contained one or multiple electron-lucent droplets (Fig. 6M, Box B and red patches in Fig. EV9F). In *Arel1$^{-/-}$* Purkinje cells, lysosomes were significantly larger and more distant from the nucleus (Fig. 6N,O). Lipofuscin granules were occasionally observed in WT Purkinje cells, probably as a result of aging; however, their numbers and sizes were markedly increased in *Arel1$^{-/-}$* Purkinje cells (Fig. 6P,Q). Periodic acid-Schiff staining also revealed the accumulation of lipofuscin granules in *Arel1$^{-/-}$* Purkinje cells (Fig. 6R). These results indicate lysosomal positioning and degradative function are impaired in Purkinje cells of *Arel1$^{-/-}$* mice.

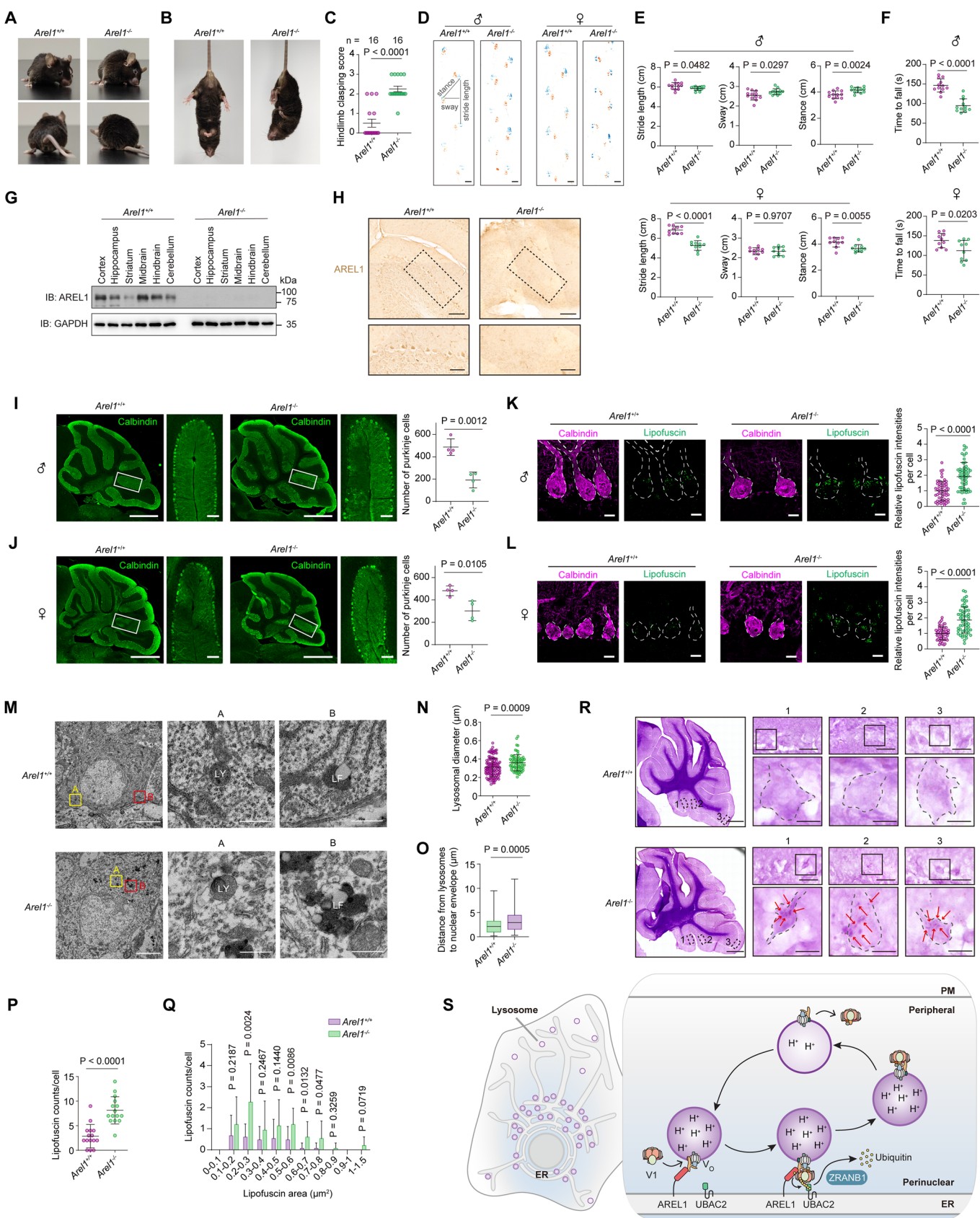

**Figure 6. *Arel1*$^{-/-}$ mice show age-dependent cerebellar ataxia, Purkinje cell loss, and lipofuscin accumulation.**

(A) Representative images showing the walking behavior of 12-month-old *Arel1*$^{+/+}$ and *Arel1*$^{-/-}$ male mice. (B) Representative images showing the postures of 12-month-old *Arel1*$^{+/+}$ and *Arel1*$^{-/-}$ male mice in the tail suspension test. (C) Hindlimb clasping score of 12-month-old *Arel1*$^{+/+}$ and *Arel1*$^{-/-}$ male mice. Data are presented as mean ± SD ($n = 16$ mice per group). Unpaired two-tailed Student's $t$ test. *Arel1*$^{+/+}$ mice vs *Arel1*$^{-/-}$ mice, $P < 0.0001$. (D) Representative footprints of 12-month-old *Arel1*$^{+/+}$ and *Arel1*$^{-/-}$ male and female mice. Scale bars, 2 cm. (E) Stride length, sway, and stance of 12-month-old *Arel1*$^{+/+}$ and *Arel1*$^{-/-}$ mice. Data are presented as mean ± SD ($n = 12$ for male mice per genotype, and $n = 10$ for female mice per genotype). Unpaired two-tailed Student's $t$ test. Stride length in *Arel1*$^{+/+}$ male mice vs stride length in *Arel1*$^{-/-}$ male mice, $P = 0.0482$; sway in *Arel1*$^{+/+}$ male mice vs sway in *Arel1*$^{-/-}$ male mice, $P = 0.0297$; stance in *Arel1*$^{+/+}$ male mice vs stance in *Arel1*$^{-/-}$ male mice, $P = 0.0024$; stride length in *Arel1*$^{+/+}$ female mice vs stride length in *Arel1*$^{-/-}$ female mice, $P < 0.0001$; sway in *Arel1*$^{+/+}$ female mice vs sway in *Arel1*$^{-/-}$ female mice, $P = 0.9707$; stance in *Arel1*$^{+/+}$ female mice vs stance in *Arel1*$^{-/-}$ female mice, $P = 0.0055$. (F) Time to fall off the rods of 12-month-old *Arel1*$^{+/+}$ and *Arel1*$^{-/-}$ mice. Data are presented as mean ± SD ($n = 12$ for male mice per genotype, and $n = 10$ for female mice per genotype). Unpaired two-tailed Student's $t$ test. *Arel1*$^{+/+}$ male mice vs *Arel1*$^{-/-}$ male mice, $P < 0.0001$; *Arel1*$^{+/+}$ female mice vs *Arel1*$^{-/-}$ female mice, $P = 0.0203$. (G) Immunoblotting analysis showing AREL1 protein expression in the indicated brain regions from 12-month-old *Arel1*$^{+/+}$ and *Arel1*$^{-/-}$ male mice. (H) Representative immunohistochemical staining images showing AREL1 expression in the cerebellum of 12-month-old *Arel1*$^{+/+}$ and *Arel1*$^{-/-}$ male mice. Boxed areas are enlarged and shown at the bottom. Scale bars, 100 μm (left), 50 μm (right). (I, J) Representative confocal images showing calbindin expression in the cerebellum of 12-month-old *Arel1*$^{+/+}$ and *Arel1*$^{-/-}$ male (I) and female (J) mice. Boxed areas are enlarged and shown on the right. Scale bars, 1 mm (main), and 50 μm (inset). Quantification of calbindin-positive Purkinje cells is presented as mean ±SD ($n = 4$ mice per genotype). Unpaired two-tailed Student's $t$ test. *Arel1*$^{+/+}$ male mice vs *Arel1*$^{-/-}$ male mice, $P = 0.0012$; *Arel1*$^{+/+}$ female mice vs *Arel1*$^{-/-}$ female mice, $P = 0.0105$. (K, L) Representative confocal images showing calbindin staining (magenta) and lipofuscin autofluorescence (green) in Purkinje cells of 12-month-old *Arel1*$^{+/+}$ and *Arel1*$^{-/-}$ male (K) and female (L) mice. Scale bars, 10 μm (main). Quantification of lipofuscin autofluorescence in calbindin-positive Purkinje cells is presented as mean ± SD ($n = 55$ and 52 cells for *Arel1*$^{+/+}$ and *Arel1*$^{-/-}$ males, respectively, and $n = 57$ and 56 cells for *Arel1*$^{+/+}$ and *Arel1*$^{-/-}$ females, respectively). Unpaired two-tailed Student's $t$ test. *Arel1*$^{+/+}$ male mice vs *Arel1*$^{-/-}$ male mice, $P < 0.0001$; *Arel1*$^{+/+}$ female mice vs *Arel1*$^{-/-}$ female mice, $P < 0.0001$. (M) Representative electron micrographs showing Purkinje cells in 12-month-old *Arel1*$^{+/+}$ and *Arel1*$^{-/-}$ male mice. Lysosomes (LY) and lipofuscin (LF) granules are enlarged in A and B, respectively. Scale bars, 5 μm (main), 0.5 μm (Box A), 0.5 μm (Box B). (N) Lysosome diameters measured using electron micrographs. Data are presented as mean ± SD ($n = 115$ and 70 lysosomes for 12-month-old *Arel1*$^{+/+}$ and *Arel1*$^{-/-}$ male mice, respectively). Unpaired two-tailed Student's $t$ test. *Arel1*$^{+/+}$ mice vs *Arel1*$^{-/-}$ mice, $P = 0.0009$. (O) Box plots showing the distance from lysosomes to the nuclear envelope measured using electron micrographs. Data are presented as median with interquartile range ($n = 115$ and 70 lysosomes for 12-month-old *Arel1*$^{+/+}$ and *Arel1*$^{-/-}$ male mice, respectively). Mann–Whitney $U$ test. *Arel1*$^{+/+}$ mice vs *Arel1*$^{-/-}$ mice, $P = 0.0005$. (P) Lipofuscin numbers per cell. Data are presented as mean ± SD ($n = 15$ cells for both 12-month-old *Arel1*$^{+/+}$ and *Arel1*$^{-/-}$ male mice). Unpaired two-tailed Student's $t$ test. *Arel1*$^{+/+}$ mice vs *Arel1*$^{-/-}$ mice, $P < 0.0001$. (Q) Size distributions of lipofuscins. Data are presented as means ± SD ($n = 43$ and 122 for 12-month-old *Arel1*$^{+/+}$ and *Arel1*$^{-/-}$ male mice, respectively). Unpaired two-tailed Student's $t$ test. $P$ values from left to right: 0.2187, 0.0024, 0.2467, 0.1440, 0.0086, 0.0132, 0.0477, 0.3259, 0.0719. (R) Representative periodic acid-Schiff staining showing lipofuscin granules in the cerebellum of 12-month-old *Arel1*$^{+/+}$ and *Arel1*$^{-/-}$ male mice. Boxed areas are enlarged as numbered. Cell contour is outlined using black dashed lines. Red arrows indicate aggregates. Scale bars, 500 μm (main), 30 μm (upper), 15 μm (lower). (S) Working model of AREL1-mediated lysosome positioning and function. Source data are available online for this figure.

To investigate whether ataxia, Purkinje cell loss, and lipofuscin accumulation seen in mid-aged *Arel1*$^{-/-}$ mice occurred concomitantly or sequentially, we examined male mice at 6 months of age, when *Arel1*$^{-/-}$ ones showed no deficits in footprint analysis (Fig. EV9G,H) or the rotarod test (Fig. EV9I). The numbers of Purkinje cells in *Arel1*$^{-/-}$ mice were similar to those in WT mice (Fig. EV9J). However, significantly more lipofuscins were detected in *Arel1*$^{-/-}$ Purkinje cells (Fig. EV9K). These results suggest that lysosomal dysfunction may cause late-onset Purkinje cell loss and ataxia. Whether *Arel1* deficiency may result in other neurological impairments such as central and peripheral vestibular problems merits further investigation.

## Discussion

In the current study, we demonstrate that lysosomes are anchored perinuclearly by the ER via a dual-tethering mechanism that involves two ER-embedded proteins (AREL1 and UBAC2) and two V-ATPase subunits ($V_o$a and $V_1$B2). AREL1 interacts with $V_o$a and further catalyzes K33-linked ubiquitylation of $V_1$B2. The ubiquitylated $V_1$B2 then binds to perinuclearly localized UBAC2 to anchor lysosomes in the perinuclear region (Fig. 6S). The deubiquitylating enzyme ZRANB1 can hydrolyze K33-linked ubiquitin chains from the $V_1$B2 subunit, thereby releasing lysosomes to the cell periphery. Via such ubiquitylation/deubiquitylation cycles, lysosomes dynamically shuttle between perinuclear and peripheral regions.

The positioning of lysosomes closely correlates with their luminal pH and functions. Perinuclear lysosomes are known to be more acidic than the peripheral ones, partly because the latter

have reduced proton-pumping activity and increased rates of proton leakage (Johnson et al, 2016). The increased density of V-ATPase holoenzyme in perinuclear lysosomes can be another contributing factor as well. In fact, both $V_1$A subunit, as indexed by the SidK probe, and $V_1$C1 have been found to be perinuclearly concentrated (Maxson et al, 2022; Tang et al, 2021). The $V_1$ subunits are, as revealed by proteomics, highly enriched on lysosomes isolated from OSW-1-treated cells and strongly decreased from amino acid-restimulated cells, where lysosomes are induced to cluster in the perinuclear and peripheral region, respectively (Ebner et al, 2023; Ratto et al, 2022). However, the mechanisms by which $V_1$ subunits are perinuclearly present are poorly understood. In addition to microtubule acetylation that accounts for perinuclear $V_1$C1 localization (Tang et al, 2021), we hereby show that $V_1$B2 ubiquitylation by AREL1 is required for its interaction with UBAC2 in the perinuclear ER, which can help secure lysosomes for complete digestion of cargos. Without AREL1, $V_1$B2 is not ubiquitylated, and lysosomes are dissociated from the perinuclear ER. It is also possible that AREL1 deficiency impairs the attachment or retention of $V_1$B2 on the lysosomal surface, which also leads to peripherally localized lysosomes with less acidity and compromised degradative capacity. These findings provide new mechanistic explanations for perinuclear lysosomes being more acidic and less mobile compared with the peripheral ones (Johnson et al, 2016). Since Rabconnectin-3 is essential for the assembly and proper function of V-ATPases (Eaton et al, 2024; Einhorn et al, 2012; Jaskolka et al, 2021; Yan et al, 2009), it will be interesting to examine whether Rabconnectin-3 is involved in AREL1-regulated V-ATPase assembly.

We show that UBAC2 via regulating lysosomal positioning determines the subcellular localization of AREL1 (Fig. EV7I).

However, why UBAC2 specifically resides in the perinuclear region is unclear. It has been shown that the ER-embedded proteins such as CLIMP63, p180 and KTN1, and cytoskeletons such as microtubules with different modifications and vimentin-containing intermediate filaments help define the perinuclear morphology of the ER and restrain proteins in the perinuclear ER (Cremer et al, 2023; Zheng et al, 2022). Whether these ER-shaping factors define the perinuclear localization of UBAC2 needs to be explored further.

Regarding the physiological consequences of proper lysosomal positioning, it has been shown, in cultured cells, that lysosomes can change their intracellular positioning in response to nutrient availability, with the perinuclear ones being responsible for substrate degradation (Jerabkova-Roda et al, 2024; Korolchuk et al, 2011). The RNF26-mediated perinuclear localization of lysosomes has been shown to promote lysosomal trafficking of activated EGFR and termination of EGF-induced AKT signaling (Cremer et al, 2021), as well as ER reorganization in response to proteotoxic stress (Cremer et al, 2023). In our study, we take a step further and evaluate the functional significance of AREL1-mediated lysosomal positioning at the animal level. $Arel1^{-/-}$ mice display more peripherally localized lysosomes and, strikingly, increased numbers and sizes of lipofuscin granules—suggestive of lysosomal dysfunction and senescence—in Purkinje cells (Figs. 6K–R and EV9F). They show age-dependent loss of Purkinje cells (Fig. 6I–L), where *ARE1L1* is highly expressed (Figs. 6H and EV9C,D), and progressive ataxic phenotype (Fig. 6A–F; Movies EV1–4). Consistent with these phenotypes, genes involved in the AREL1−UBAC2−V-ATPase−ZRANB1 axis were highly expressed in Purkinje cells (Fig. EV9D). These results highlight the importance of AREL1-regulated lysosomal distribution and function in highly polarized, postmitotic neurons.

Lysosomal dysfunction has been closely implicated in normal aging and neurodegenerative diseases, including Alzheimer's disease and Parkinson's disease, and many of the identified mutations affect the activities of lysosomal enzymes, V-ATPase, and transporters (Lie and Nixon, 2019; Tan and Finkel, 2023; Udayar et al, 2022). Our work identifies an upstream regulator of lysosomal function and suggests that disrupted ER regulation of lysosomal positioning is a previously uncharacterized driving factor for Purkinje cell loss. It will be worth investigating whether aberrant lysosomal positioning underlies neurodegeneration in various pathological conditions.

# Methods

### Reagents and tools table

| Reagent/resource | Reference or source | Identifier or catalog number |
|---|---|---|
| **Experimental models** | | |
| *Arel1*^floxed/floxed *(M. musculus)* | Nanjing Biomedical Research Institute of Nanjing University | N/A |
| CMV-Cre *(M. musculus)* | Shulaibao (Wuhan) Biotechnology Co., Ltd. | YDG21032203 |
| HEK293T cells *(H. sapiens)* | ATCC | CRL-3216 |

| Reagent/resource | Reference or source | Identifier or catalog number |
|---|---|---|
| HeLa cells *(H. sapiens)* | ATCC | CRM-CCL-2 |
| U2OS cells *(H. sapiens)* | ATCC | HTB-96 |
| **Recombinant DNA** | | |
| pLVX-LAMP1-mCherry-TurboID(N terminus, 1-73)-IRES-Puro | This study | N/A |
| pLVX-TurboID(C terminus, 74-320)-EGFP-Sec61β-IRES-Puro | This study | N/A |
| pLVX-TurboID(C terminus, 74-320)-miRFP670-Sec61β-IRES-Puro | This study | N/A |
| pLVX-EGFP-Sec61β-IRES-Puro | This study | N/A |
| pLVX-mCherry-Sec61β-IRES-Puro | This study | N/A |
| pLVX-AREL1(WT)-EGFP-IRES-Puro | This study | N/A |
| pLVX-AREL1(ΔIDR, 313-348)-EGFP-IRES-Puro | This study | N/A |
| pLVX-AREL1(ΔLCR, 346–360)-EGFP-IRES-Puro | This study | N/A |
| pLVX-AREL1(Δ5K)-EGFP-IRES-Puro | This study | N/A |
| pLVX-AREL1(Y354A/Y356A)-EGFP-IRES-Puro | This study | N/A |
| pLVX-AREL1(Y354F/Y356F)-EGFP-IRES-Puro | This study | N/A |
| pLVX-AREL1(Y354W/Y356W)-EGFP-IRES-Puro | This study | N/A |
| pLVX-AREL1(1–482)-EGFP-IRES-Puro | This study | N/A |
| pLVX-AREL1(C790A)-EGFP-IRES-Puro | This study | N/A |
| pLVX-AREL1(1-712)-EGFP-IRES-Puro | This study | N/A |
| pLVX-AREL1(1-705)-EGFP-IRES-Puro | This study | N/A |
| pLVX-AREL1(1-668)-EGFP-IRES-Puro | This study | N/A |
| pLVX-AREL1(1-589)-EGFP-IRES-Puro | This study | N/A |
| pLVX-AREL1(Δ706-711)-EGFP-IRES-Puro | This study | N/A |
| pLVX-mCherry-ZRANB1(WT)-IRES-Puro | This study | N/A |
| pLVX-mCherry-ZRANB1(C443S)-IRES-Puro | This study | N/A |
| pLVX-UBAC2-mCherry-IRES-Puro | This study | N/A |
| pLVX-UBAC2(WT)-EGFP-IRES-Puro | This study | N/A |
| pLVX-UBAC2(ΔUBA, 304-344)-EGFP-IRES-Puro | This study | N/A |
| pLVX-AUP1-EGFP-IRES-Puro | This study | N/A |
| pLVX-RHBDD3-EGFP-IRES-Puro | This study | N/A |

| Reagent/resource | Reference or source | Identifier or catalog number |
|---|---|---|
| p3×FLAG-CMV-14-ATP6AP1 | This study | N/A |
| p3×FLAG-CMV-14-ATP6AP2 | This study | N/A |
| p3×FLAG-CMV-14-ATP6V0A1 | This study | N/A |
| p3×FLAG-CMV-14-ATP6V0B | This study | N/A |
| p3×FLAG-CMV-14-ATP6V0C | This study | N/A |
| p3×FLAG-CMV-14-ATP6V0D1 | This study | N/A |
| p3×FLAG-CMV-14-ATP6V1A | This study | N/A |
| p3×FLAG-CMV-14-ATP6V1B1 | This study | N/A |
| p3×FLAG-CMV-14-ATP6V1B2 | This study | N/A |
| p3×FLAG-CMV-14-ATP6V1C1 | This study | N/A |
| p3×FLAG-CMV-14-ATP6V1D | This study | N/A |
| p3×FLAG-CMV-14-ATP6V1E1 | This study | N/A |
| p3×FLAG-CMV-14-ATP6V1G1 | This study | N/A |
| p3×FLAG-CMV-14-ATP6V1H | This study | N/A |
| p3×FLAG-CMV-14-ATP6V0A1(NTD, 1-388) | This study | N/A |
| p3×FLAG-CMV-14-ATP6V0A1(CTD, 389-837) | This study | N/A |
| p3×FLAG-CMV-14-ATP6V0A2(NTD, 1-393) | This study | N/A |
| p3×FLAG-CMV-14-ATP6V0A2(CTD, 394-856) | This study | N/A |
| p3×FLAG-CMV-14-ATP6V0A3(NTD, 1-385) | This study | N/A |
| p3×FLAG-CMV-14-ATP6V0A3(CTD, 386-830) | This study | N/A |
| p3×FLAG-CMV-14-ATP6V0A4(NTD, 1-390) | This study | N/A |
| p3×FLAG-CMV-14-ATP6V0A4(CTD, 391-840) | This study | N/A |
| pcdna3.0-AREL1(WT)-myc | This study | N/A |
| pcdna3.0-AREL1(Δ706-711)-myc | This study | N/A |
| pcdna3.0-UBAC2(WT)-myc | This study | N/A |
| pcDNA3.0-UBAC2(ΔUBA)-myc | This study | N/A |
| pET-28a(+)-mEGFP | This study | N/A |
| pET-28a(+)-mEGFP-IDR | This study | N/A |
| pET-28a(+)-mEGFP-LCR | This study | N/A |
| pET-28a(+)-mEGFP-LCR(Δ5K) | This study | N/A |
| pET-28a(+)-mEGFP-LCR(Y354A/Y356A) | This study | N/A |
| pET-28a(+)-mEGFP-LCR(Y354W/Y356W) | This study | N/A |
| pET-28a(+)-mEGFP-LCR(Y354F/Y356F) | This study | N/A |
| pEF-HA-ubiquitin(WT, K6R, K11R, K27R, K29R, K33R, K48R, K63R, K29, K33) | This study | N/A |
| pX330-U6-Chimeric_BB-CBh-hSpCas9 | Addgene | 42230 |

| Reagent/resource | Reference or source | Identifier or catalog number |
|---|---|---|
| pLKO.1 puro | Addgene | 8453 |
| pMD2.G | Addgene | 12259 |
| psPAX2 | Addgene | 12260 |
| **Antibodies** | | |
| Anti-alpha Tubulin | Proteintech | 11224-1-AP |
| Anti-ATP6V0A3 | Proteintech | 83351-6-RR |
| Anti-ATP6V1B2 | Proteintech | 15097-1-AP |
| Anti-ATP6V1B2 | Proteintech | 68441-1-Ig |
| Anti-ATP6V1D | Proteintech | 14920-1-AP |
| Anti-calbindin | Cell Signaling Technology | 13176 |
| Anti-calnexin | Proteintech | 10427-2-AP |
| Anti-calnexin | Merck | MAB3126 |
| Anti-DYKDDDDK tag | Proteintech | 20543-1-AP |
| Anti-EEA1 | BD Biosciences | 610457 |
| Anti-HA tag | Proteintech | 51064-2-AP |
| Anti-LAMP1 | Developmental Studies Hybridoma Bank | clone H4A3 |
| Anti-LAMP1 | Cell Signaling Technology | 9091 |
| Anti-mCherry | Proteintech | 26765-1-AP |
| Anti-myc tag | Proteintech | 16286-1-AP |
| control mouse IgG | Proteintech | B900620 |
| Anti-AREL1 | DIA-AN (Wuhan, China) | N/A |
| Anti-EGFP | Homemade | N/A |
| peroxidase AffiniPure goat anti-mouse IgG secondary antibody | Jackson ImmunoResearch | 115-035-003 |
| peroxidase AffiniPure goat anti-rabbit IgG secondary antibody | Jackson ImmunoResearch | 111-035-144 |
| HRP-conjugated goat anti-rabbit secondary antibody | Proteintech | PR30009 |
| Alexa Fluor 488 goat anti-mouse IgG | Thermo Fisher Scientific | A11001 |
| Alexa Fluor 488 goat anti-rabbit IgG | Thermo Fisher Scientific | A11008 |
| Alexa Fluor 555 donkey anti-mouse IgG | Thermo Fisher Scientific | A31570 |
| Alexa Fluor 647 goat anti-rabbit IgG | Thermo Fisher Scientific | A21244 |
| **Oligonucleotides and other sequence-based reagents** | | |
| sgRNA targeting sequences human *AREL1*: ACTATTTATGACTACGTGCG, CTCTCATGTCGAGCTAGCAG, CACGTGCGGCAAGCTCAAAG | This study | N/A |
| shRNA targeting sequences negative control: CCTAAGGTTAAGTCGCCCTCG | This study | N/A |
| shRNA targeting sequence human *ANKLE2*: ATGTCAGCCAGGATCGCTAAA | This study | N/A |

| Reagent/resource | Reference or source | Identifier or catalog number |
|---|---|---|
| shRNA targeting sequence human *AREL1*: CCGGGAATGGTTTGAGCTAAT | This study | N/A |
| shRNA targeting sequence human *ATP6AP1*: GCATTGAGGATTTCACAGCAT | This study | N/A |
| shRNA targeting sequence human *ATP6AP2*: AGTCTTGACAGTGTTGCAAAT | This study | N/A |
| shRNA targeting sequence human *ATP6V0A3*: CAACTCCTTCAAGATGAAGAT | This study | N/A |
| shRNA targeting sequence human *ATP6V1B2*: GCTGAATTTCTGGCGTACCAA | This study | N/A |
| shRNA targeting sequence human *AUP1*: GCAGAGCACATGAAGCGACAA | This study | N/A |
| shRNA targeting sequence human *ESYT1*: GCGTCTCACCACAGTCTTAAA | This study | N/A |
| shRNA targeting sequence human *ESYT2*: GCTCGCAGAGAAACAAGCTTA | This study | N/A |
| shRNA targeting sequence human *MOSPD2*: CCCAGATGGTTATTGGAAATT | This study | N/A |
| shRNA targeting sequence human *MXRA7*: GAGAAGGCTTCTCCTTCAAAT | This study | N/A |
| shRNA targeting sequence human *RNF26*: CCCTTGGAAATTGCTGAAGGA | This study | N/A |
| shRNA targeting sequence human *RHBDD3*: GCCGTGTCACTGTTGGTTGGA | This study | N/A |
| shRNA targeting sequence human *STX18*: GACATAAGAGAGGCCATTAAA | This study | N/A |
| shRNA targeting sequence human *TMEM9*: GCATCTGTCCACCTTATAGAA | This study | N/A |
| shRNA targeting sequence human *TMEM214*: GAAGGTACAGAAGTCTTTGCA | This study | N/A |
| shRNA targeting sequence human *TMEM106B*: CCATTATTGGTCCACTTGATA | This study | N/A |
| shRNA targeting sequence human *TMX1*: GCTGAAAGTAAAGAAGGAACA | This study | N/A |
| shRNA targeting sequence human *UBAC2*: GCCATTACATTAGCATGTATT | This study | N/A |
| shRNA targeting sequence human *UBE2J1*: GATGATATACCTACAACATTC | This study | N/A |

| Reagent/resource | Reference or source | Identifier or catalog number |
|---|---|---|
| shRNA targeting sequence human *USE1*: GACGTAAGGAAGAGAACTGGA | This study | N/A |
| shRNA targeting sequence human *VAPA*: GCGAAATCCATCGGATAGAAA | This study | N/A |
| shRNA targeting sequence human *VMA21*: GCTCCTGTTCTTCACAGCTTT | This study | N/A |
| shRNA targeting sequence human *ZFYVE27*: GTGTAACCAGACCTTGAGCAA | This study | N/A |
| shRNA targeting sequence human *ZRANB1*: CACGCTGGAAAGATTGGGAAT | This study | N/A |
| **Chemicals, enzymes, and other reagents** | | |
| DAB staining kit | Proteintech | PR30010 |
| DAPI | Thermo Fisher Scientific | D1306 |
| D-biotin | Sangon Biotech | A600078 |
| DQ-OVA | Thermo Fisher Scientific | D12053 |
| DTT | Sangon Biotech | A620058 |
| DuoLink In Situ Orange Starter Kit | Merck | DUO92102 |
| ECL Plus western blotting substrate | Thermo Fisher Scientific | 32132 |
| FLAG beads | Merck | A2220 |
| FluorSave mounting medium | Millipore | 345789 |
| FuGENE HD | Promega | E2311 |
| LysoSensor Green DND-189 | YEASEN Biotech | 40767ES50 |
| NeutrAvidin agarose resin | Thermo Fisher Scientific | 29204 |
| Nigericin | Targetmol | T3092 |
| Ni-NTA Agarose | QIAGEN | 30210 |
| Oregon Green 488 conjugated dextran-10 kD | Molecular Probes | D7170 |
| PAS staining kit | Solarbio | G1281 |
| Polyethyleneimine | Polysciences | 23966 |
| Protease inhibitor cocktail | Millipore | P8340-5mL |
| Protein A/G agarose beads | Smart-Lifesciences | SA032005 |
| Puromycin | Biofroxx | 1299MG025 |
| SiR-Lysosome | Cytoskeleton | CY-SC012 |
| Trypsin | Sangon Biotech | A003702 |
| **Software** | | |
| DigitalMicrograph | Gatan | N/A |
| Fiji | ImageJ | N/A |
| DigitalMicrograph | Gatan | N/A |
| GraphPad Prism | Prism | N/A |
| Perseus | Perseus Software Ltd. | N/A |

| Reagent/resource | Reference or source | Identifier or catalog number |
| --- | --- | --- |
| Rstudio | Posit | N/A |
| ZEN (black edition) | ZEISS | N/A |
| Other | | |
| Standard chow | Research Diets | D10001 |

## Ethical statement

Mice were maintained and used in accordance with the guidelines of the Institutional Animal Care and Use Committee of Wuhan University under the protocol number WDSKY0201408.

## Mice

$Arel1^{floxed/floxed}$ mice were generated by Nanjing Biomedical Research Institute of Nanjing University using the CRISPR-cas9 system. Cas9 mRNA, single guide RNAs (sgRNAs), and donor were co-injected into zygotes. SgRNAs directed cas9 endonuclease cleavage in intron 3 and intron 8, resulting in loxP sites inserted into intron 3 and intron 8, respectively, by homologous recombination. $Arel1^{floxed/floxed}$ mice were first crossed with CMV-Cre transgenic mice. Heterozygous offsprings were intercrossed to generate *Arel1* homozygous knockout mice ($Arel1^{-/-}$) and wild-type littermate controls.

Mice were housed in a specific pathogen-free environment under a 12-h light/12-h dark cycle, with the temperature at 21–23 °C and relative humidity at 50–60%. Mice had ad libitum access to water and standard chow (Research Diets, D10001). For behavioral experiments, mice were subjected to habituation in the behavioral test room for 1 h. Age and gender of mice used for the experiments were stated in the relevant figure legends.

## Plasmids

The plasmids were generated using standard molecular cloning techniques. Those expressing various forms of ubiquitin were generated as described previously (Wang et al, 2017).

## Cell culture

HEK293T (a human embryonic kidney cell line), HeLa (a human cervical cancer cell line), and U2OS (a human osteosarcoma cell line) cells were obtained from ATCC and grown as a monolayer at 37 °C in 5% $CO_2$. No further authentication of the cell lines was performed before use. No test for mycoplasma contamination was performed.

Cells were maintained in DMEM containing 100 units/ml penicillin and 100 µg/ml streptomycin sulfate supplemented with 10% fetal bovine serum.

## Generation of *AREL1* knockout cells using CRISPR/Cas9

U2OS and HeLa cells were transfected with three pX330 vectors containing gRNAs targeting human *AREL1* exons. Cells were plated into 96-well plates in single colonies. Knockout of *AREL1* was verified by genomic sequencing and immunoblotting analysis.

## Transfection

Transient transfection of HEK293T and U2OS cells was performed using linear polyethyleneimine (Polysciences, 23966) and FuGENE HD (Promega, E2311), respectively, following the manufacturers' instructions.

## Short hairpin RNA (shRNA)-mediated knockdown

Lentiviruses were packaged with pMD2.G and psPAX2 in HEK293T cells, and transduced into U2OS cells. After 48 h, cells were subjected to 2 µg/ml puromycin selection for 5 days to generate *AREL1* knockdown stable cell lines.

## Subcellular fractionation

U2OS cells were harvested and washed once with ice-cold PBS. Then, cells were lysed with a Dounce homogenizer in ice-cold hypotonic buffer (10 mM HEPES, pH 7.2, 10 mM KCl, 1.5 mM $MgCl_2$, 0.1 mM EGTA, and protease inhibitors). The homogenates were centrifuged at $3000\times g$ at 4 °C for 5 min to pellet unbroken cells and nuclei. The supernatants were subjected to centrifugation at $30,000\times g$ at 4 °C for 10 min. The supernatants were cytosolic fractions, and the pellets were membrane fractions. Membrane fractions were washed three times with hypotonic buffer before sample preparation.

For the $Na_2CO_3$ treatment, cell homogenates were treated with 0.1 M $Na_2CO_3$ at 37 °C for 30 min and then subjected to centrifugation at $30,000\times g$ at 4 °C for 10 min to isolate cytosolic and membrane fractions.

For the trypsin treatment, cell homogenates were treated with trypsin (Sangon Biotech, A003702) at 37 °C for 30 min and then subjected to sample preparation.

## Proximity labeling using split-TurboID

HEK293T cells and *AREL1* knockout U2OS cells were transfected with split-TurboID plasmids using LPEI and FuGENE HD, respectively. After 48 h, D-biotin (Sangon Biotech, A600078) was added to the culture medium to a final concentration of 100 µM in HEK293T or indicated concentrations in *AREL1* knockout U2OS cells. After 4-h incubation, cells were washed once with ice-cold PBS and harvested. Then cells were subjected to membrane fractionation as described above.

To pulldown biotinylated membrane proteins, membrane fractions were homogenized in ice-cold lysis buffer (50 mM HEPES, 2.5 mM $MgCl_2$, 200 mM KCl, 5% glycerol, 1% Triton X-100 plus protease inhibitors), and centrifuged at $12,000\times g$ for 10 min to remove the non-dissolved parts. The supernatants were incubated with 100 µl high-capacity NeutrAvidin agarose resin (Thermo Scientific, 29204) at 4 °C for 4 h. After washing with the lysis buffer for three times, the biotinylated proteins were eluted by incubating beads with 2× loading buffer plus 2 mM D-biotin and 20 mM DTT for 10 min at 95 °C. The eluents were subjected to SDS–PAGE and mass spectrometry analysis.

## Mass spectrometry

Enriched biotinylated proteins were separated by SDS–PAGE and digested by in-gel tryptic digestion. The gel slices were treated with

10 mM DTT and 55 mM iodoacetamide to reduce the disulfide bond and alkylate the resulting thiol group. Trypsin was added at a final concentration of 10 ng/µl for overnight digestion at 37 °C. The phosphopeptides were enriched by using a homemade TiO$_2$ microcolumn. In brief, the digested peptides were loaded in 80% acetonitrile (ACN), 5% trifluoroacetic acid (TFA), 1 M glycolic acid, washed with 80% ACN, 1% TFA, and 10% ACN, 0.1% TFA, eluted with 2 M NH$_3$·H$_2$O, and desalted in a R3 microcolumn. LC–MS/MS was performed using EASY-nLC 1000 system interfaced to Q Exactive HF (Thermo Fisher Scientific).

To identify lysosomal proteins that interact with AREL1, the database search was performed using Maxquant software, and the quantified protein list was further analyzed using Perseus software for statistical analysis. The endogenously biotinylated proteins, namely pyruvate carboxylase (PC), propionyl-CoA carboxylase subunit a (PCCA), propionyl-CoA carboxylase subunit b (PCCB), methylcrotonoyl-CoA carboxylase subunit a (MCCC1), and methylcrotonoyl-CoA carboxylase beta chain (MCCC2), were served as the internal references for normalization. A two-sample *t* test was conducted to calculate *P* values of changed proteins using the implemented function in Perseus.

### Immunostaining and confocal microscopy

Cells were grown on glass coverslips for 48 h, washed with PBS, and fixed with 4% paraformaldehyde (PFA) for 10 min at room temperature (RT). Cells were then permeabilized with liquid nitrogen and incubated with primary and secondary antibodies diluted in PBS containing 1% bovine serum albumin for 1 h at RT. After washing 3 times with PBS, coverslips were mounted on glass slides with FluorSave mounting medium (Millipore, 345789).

For endogenous AREL1 staining, cells were first fixed with PFA and then incubated with citrate buffer (pH 6.0) at 95 °C for 30 min followed by permeabilization with liquid nitrogen. Then, cells were incubated with the anti-AREL1 (20 µg/ml) overnight, followed by secondary antibody staining for 1 h at RT.

For endogenous ATP6V1D staining, cells were incubated in 25 µg/ml digitonin for 10 min on ice to remove cytosolic proteins before fixation. Then, fixed cells were permeabilized with liquid nitrogen and incubated with the anti-ATP6V1D overnight, followed by secondary antibody staining for 1 h at RT. Images were taken using a Leica SP8 LIGHTNING confocal system equipped with a ×63 objective lens.

### Proximity ligation assay (PLA) and quantification analysis

PLA was conducted using a mouse primary antibody against endogenous ER protein calnexin and rabbit primary antibody against endogenous lysosomal protein LAMP1, followed by mouse and rabbit PLA probes from the DuoLink In Situ Orange Starter Kit Mouse/Rabbit (Sigma-Aldrich, DUO92102), following the manufacturer's instructions. Images were taken using a Leica SP8 LIGHTNING confocal microscope with the 405 nm (for DAPI) and 561 nm (for PLA signal) channels.

PLA signal quantification was performed using Fiji as previously described (Saric et al, 2021). The 561 nm-channel images were imported to Fiji and converted to binary images, and a threshold was applied to eliminate background using the *Image>Adjust>Threshold* command. The Watershed feature was applied to separate any signal dots apparently in touch using the *Process>Binary>Watershed* command. Dots that were at least 5 pixels in size were counted using the *Analyze>Analyze Particles* command, and the total number of dots per cell (counterstained by DAPI) was measured and recorded.

### Immunoblotting

Equal amounts of total proteins were resolved by SDS–PAGE and transferred to PVDF membranes. Membranes were blocked with 5% skim milk in tris-buffered saline (TBS) containing 0.075% Tween-20 (TBST), probed with indicated primary antibodies overnight at 4 °C. After washing in TBST three times, membranes were incubated with horseradish peroxidase-conjugated secondary antibodies diluted in TBST supplemented with 5% skim milk for 1 h at RT, followed by at least three washes with TBST. Chemiluminescent signals were detected using Pierce ECL Plus western blotting substrate (Thermo Scientific, 32132).

For detecting biotinylated proteins, PVDF membranes were incubated with streptavidin-HRP diluted in TBST supplemented 3% bovine serum albumin at RT for 1 h. Membranes were then washed three times with TBST, followed by chemiluminescent detection.

### Protein expression and purification

pET-28a(+) plasmids encoding mEGFP fusion proteins were transformed into the *Escherichia coli* strain BL21(DE3), and expression was induced by 0.5 mM IPTG in LB at 16 °C overnight. Cells were harvested for sonication in PBS with protease inhibitors, followed by centrifugation at 5000× *g* for 5 min. The supernatants were incubated with Ni-NTA Agarose (QIAGEN, 30210) at for 4 h 4 °C, washed three times with binding buffer (20 mM sodium phosphate, 0.5 M NaCl, 40 mM imidazole, pH 7.4), and eluted with 500 mM imidazole (20 mM sodium phosphate, 0.5 M NaCl, pH 7.4). The eluents were concentrated by ultra-centrifugation and stored at −30 °C.

### In vitro phase separation

Purified recombinant proteins were diluted in the buffer composed of 50 mM HEPES (pH 7.4) and 50 mM NaCl at RT. Mixed reactions were pipetted to 96-well microscopy plates and imaged by a Leica SP8 LIGHTNING confocal system.

### Immunoprecipitation (IP)

Cells were collected and lysed in IP buffer (50 mM Tris-HCl, pH 7.6, 150 mM NaCl, 5 mM EDTA, 0.5% NP40, and protease inhibitors). After centrifugation at 12,000× *g* for 10 min, supernatants were collected and immunoprecipitated with 20 µl anti-FLAG beads (Sigma-Aldrich, A2220) at 4 °C for 4 h. Beads were centrifuged at 1000× *g* for 5 min and washed three times with 0.5% NP40 IP buffer. Beads were then incubated with the loading buffer to elute pulled-down proteins. Aliquots of eluents were subjected to immunoblotting analysis.

## Ubiquitylation

For ubiquitylation assay in transfected HEK293T cells, cells were harvested 48 h post transfection and lysed in the ubiquitylation buffer (1% Nonidet P-40, 1% deoxycholate, 5 mM EDTA, 5 mM EGTA, protease inhibitors, and 10 mM N-ethylmaleimide in PBS). Lysates were immunoprecipitated with 20 µl anti-FLAG beads at 4 °C for 4 h. Then, the beads were washed three times with the ubiquitylation buffer, and incubated with the loading buffer to elute pulled down proteins. Aliquots were subjected to immunoblotting analysis.

For ubiquitylation assay of endogenous $V_1B2$ in whole-cell lysate, U2OS cells (ten 15-cm dishes per group) were harvested and lysed in the ubiquitylation buffer (1% Nonidet P-40, 1% deoxycholate, 5 mM EDTA, 5 mM EGTA, protease inhibitors, 10 mM N-ethylmaleimide, 10 µM PR-619 in PBS). For ubiquitylation assay of endogenous $V_1B2$ in cytosol and membrane fractions, U2OS cells (ten 15-cm dishes per group) were harvested and broken with Dounce homogenizer in ice-cold hypotonic buffer (10 mM HEPES, pH 7.2, 10 mM KCl, 1.5 mM $MgCl_2$, 0.1 mM EGTA, protease inhibitors, 10 mM N-ethylmaleimide, and 10 µM PR-619 in PBS), followed by subcellular fractionation as described above. Isolated membrane fractions were then lysed with RIPA buffer containing protease inhibitors and deubiquitinase inhibitors.

Lysates (1 mg total protein), cytosolic fractions (750 µg total proteins), or lysed membrane fractions (250 µg total proteins) were incubated with 2 µg anti-ATP6V1B2 antibody or control mouse IgG and 100 µl Protein A/G agarose beads (Smart-Lifesciences, SA032005) at 4 °C for 12 h. Then, beads were washed 3 times with the corresponding buffer, and incubated with the loading buffer to elute pulled down proteins. Aliquots were subjected to immunoblotting analysis.

## Measurement of lysosomal pH

Lysosomal pH measurement with a ratiometric pH-sensitive dye Oregon Green 488 conjugated dextran-10 kD (Molecular Probes, D7170) was performed as described previously (Hu et al, 2022). In brief, cells grown on culture plates with glass-bottom were loaded with 150 µg/mL Oregon Green 488 Dextran overnight and chased in medium without dye for 3 h before imaging. Cells were washed with Ringer's buffer (155 mM NaCl, 5 mM KCl, 2 mM $CaCl_2$, 1 mM $MgCl_2$, 2 mM $NaH_2PO_4$, 10 mM HEPES, and 10 mM glucose, pH 7.4) and imaged using a Zeiss LSM 880 confocal microscope equipped with a ×60 objective lens and a temperature-controlled stage. The fluorescence emission (530 ± 20 nm) excited at 440-nm and 490-nm wavelengths were acquired with a cooled digital CCD camera.

To generate the pH calibration curve, cells were incubated with pH standard buffers (pH 3.5, 4.5, 5.5, 6.5, 7.5) supplemented with 10 µM nigericin (Targetmol, T3092). The fluorescence intensities excited at 490 nm and 440 nm in response to varied pH were quantified by Fiji. The ratios of fluorescence intensities excited at 490 nm and 440 nm in response to varied pH were fitted to a sigmoidal equation. The lysosomal pH values were then calculated with the resulting intensity ratios (490 nm/440 nm) of individual lysosomes based on the calibration curve.

## LysoSensor green staining

U2OS cells were cultured in a 24-well glass-bottom dish. After 24 h, cells were incubated with 1 µM LysoSensor Green DND-189 (YEASEN Biotech, 40767ES50) diluted in culture medium at 37 °C for 30 min. Live cells were then washed twice and kept in culture medium without phenol red for imaging. Images were taken using a Leica SP8 LIGHTNING confocal system equipped with a 63× objective lens and a temperature-controlled stage. The fluorescence intensities of LysoSensor green were quantified by Fiji.

## SiR-Lysosome staining

U2OS cells were cultured in a 24-well glass-bottom dish. After 24 h, live cells were incubated with 0.5 µM SiR-Lysosome (Cytoskeleton, CY-SC012) diluted in culture medium at 37 °C for 1 h. Live cells were then washed twice and kept in culture medium without phenol red for imaging. Images were taken using a Leica SP8 LIGHTNING confocal system equipped with a 63× objective lens and a temperature-controlled stage. The fluorescence intensities of SiR-Lysosome were quantified by Fiji.

## Endocytosis of DQ-OVA

U2OS cells were grown on glass coverslips. After 24 h, cells were incubated with 10 µg/ml DQ-OVA (Invitrogen, D12053) at 37 °C for 3 h. Cells were washed three times with PBS, fixed with 4% PFA, and mounted for imaging. Images were taken using a Leica SP8 LIGHTNING confocal system equipped with a ×63 objective lens. The fluorescence intensities of DQ-OVA were quantified by Fiji.

## FRAP

Indicated cells grown on glass-bottom dishes were transfected with the plasmid expressing $V_1E1$-mCherry. After 48 h, live cells were washed twice, kept in culture medium without phenol red at a temperature-controlled stage (37 °C), and imaged using a Zeiss LSM 880 confocal microscope equipped with a ×60 objective lens.

Perinuclear lysosomes due to relatively immobility were selected for region of interest (ROI) imaging. Photobleaching of the mCherry in ROI was achieved using 20 iterations of the 561 nm laser line at 100% intensity, and post-bleached images were collected for 180 s with 5-s intervals to determine recovery of mCherry fluorescence in the bleached region. Quantitation of the time-lapse fluorescence intensity of ROI and the calculation of half-time of fluorescence recovery ($\tau1/2$) were performed using Zen software (Zen Black 2.3 SP1).

## Footprint analysis

The footprint analysis was performed as previously described (Wertman et al, 2019). In brief, mice were acclimated to the behavior rooms for 1 h prior to testing. The fore and hind paws were painted with orange and blue non-toxic water-based paints, respectively. Mice were allowed to walk in a straight line in a narrow tunnel (60 cm × 10 cm × 10 cm) on white paper, with a darkened cage used as the bait at the end of the tunnel. Mice were repeatedly tested three times. In each test, three hind paw steps

from the middle portion were measured for stride length, stride width, and stance.

## Hindlimb clasping

Mice were lifted clear of all surrounding objects by the tail. The positions of their hindlimbs were observed for 1 min. Hindlimb positions were scored manually as described previously (Guyenet et al, 2010; Petkovic et al, 2020): 0, the hindlimbs were consistently splayed outward, away from the abdomen; (1) one hindlimb retracted toward the abdomen for more than 50% of the time suspended; (2) both hindlimbs were partially retracted toward the abdomen for more than 50% of the time suspended; (3) both hindlimbs were entirely retracted and touching the abdomen for more than 50% of the time suspended.

## Rotarod test

The rotarod test was performed as previously described (de Haas et al, 2016; Hayashi-Takagi et al, 2015; Kim et al, 2019; Shiotsuki et al, 2010). In brief, the mice were trained 3 days before test, mice were trained at 8 rpm for three times per day with a 5 min resting period between each trial. On day 4, mice were placed on an accelerating rotarod cylinder, and the latency time of the animals was measured. The speed was slowly increased from 4 to 40 rpm within 5 min. A trial ended if the animal fell off the rods.

## Immunostaining of cerebellar sections

Mice were anesthetized and transcardially perfused with PBS followed by 4% PFA in PBS. Brains were post-fixed with 4% PFA for 24 h and then incubated in 30% sucrose/PBS till sinking. Brains were embedded in OCT compound and snap frozen with liquid nitrogen-cooled isopentane. OCT blocks were then sectioned in 30 µm-thick sections using a Leica Cryostat and kept at −80 °C.

For cryoimmunolabeling, cryosections were thawed at RT and then washed three times with PBS for 2 min each. Sections were permeabilized with PBS containing 0.1% Triton X-100 for 10 min, followed by three times washing with PBS for 2 min. Sections were blocked in 10% normal goat serum in PBS for 1 h at RT, and then incubated at 4 °C with the anti-calbindin antibody. After overnight incubation, sections were washed three times with PBS for 5 min and incubated with goat anti-rabbit secondary antibody at RT for 1 h. After three times washing with PBS for 5 min, sections were mounted with FluorSave mounting medium.

For lipofuscin autofluorescence detection, cryosections were first immunostained with anti-calbindin primary antibody and Alexa Fluor 555 secondary antibody.

For immunohistochemical staining, cryosections were thawed at RT and then washed 3 times with PBS for 2 min. Sections were quenched with 3% hydrogen peroxide in PBS for 15 min at RT. Then antigen retrieval was done by incubating sections with citrate buffer (pH 6.0) for 20 min at 95 °C. Then, slides were incubated with anti-AREL1 antibody overnight at 4 °C. After washing, the HRP-conjugated secondary antibody was added and incubated for 30 min at RT. After washing, the DAB staining was conducted with a DAB staining kit (Proteintech, PR30010). After washing three times with PBS, sections were dehydrated by increasing concentrations of ethanol and xylene, and then mounted with neutral balsam mounting medium.

## Transmission electron microscopy

Mice were anesthetized and transcardially perfused with 4% PFA and 2% glutaraldehyde in PBS. After dissection, brains were post-fixed in the same fixative overnight. Cerebellar samples were stained with 1% osmium tetroxide, dehydrated in increasing concentrations of acetone from 30% to 100% and infiltrated and embedded in SPI-PON812 resin (SPI-CHEM). Samples were then sectioned at 2-µm thickness and stained with 1% toluidine blue for light microscopic assessment. The 70-nm ultrathin sections were cut using an ultra-microtome (Leica, UC7), followed by staining with uranyl acetate and lead citrate. Sections were mounted on copper grids and viewed with a transmission electron microscope (JEOL, JEM-1400Plus). Electron micrographs were captured by a Gatan digital camera (Gatan, Rio9) and its application software (Gatan Digital Micrograph 3.0 software).

## Periodic acid Schiff (PAS) staining

PAS staining was conducted following the manufacturer's instructions of the PAS staining kit (Solarbio, G1281). The brain sections were dehydrated by increasing concentrations of ethanol and xylene, and then mounted with neutral balsam mounting medium.

## Quantification of confocal images

Cells were selected and analyzed as previously described (Saric et al, 2021; Williamson et al, 2022). Those with flat and round shapes were selected, and the narrow ones were excluded. In the cases of overexpression, cells expressing medium and similar levels of indicated proteins were chosen for analysis, and those showing high expression levels were excluded to avoid artifacts.

For vesicle distribution measurement, a total of 10 z-stacks were taken per cell from the top to the bottom and then merged. Images were exported with scale bars using Leica LAS X software. Images were then imported to Fiji, and scale bars were used to set scale using the *Analyze>Set Scale* command. Straight lines were manually drawn from vesicles to the nuclear envelope, and the distances of straight lines were calculated using the *Analyze>Measure* command and recorded.

For fluorescence intensity quantification of LysoSensor green, SiR-lysosome, and DQ-OVA, cell borders were manually outlined using the *Freehand selections* tool in Fiji, and total fluorescence intensities within the borders were quantified using the *Analyze>Measure* command and recorded. Relative fluorescence intensities were calculated by comparing experimental groups with control groups.

For colocalization quantification, images were exported in separate color channels using Leica LAS X software. Images were then imported to Fiji. Cell borders were manually outlined using the *Freehand selections* tool in one channel, and the borders were applied to the other channel using the *Edit>Selection>Restore Selection* command. The Pearson's correlation coefficient of the two channels per cell was quantified by the Coloc 2 plugin and recorded.

For lipofuscin autofluorescence quantification, a total of 10 z-stacks were taken per cell from the top to the bottom and then merged. The calbindin signals (under the 555-nm channel) were used to draw the borders of somas of Purkinje cells, and the of lipofuscin autofluorescence (under the 488-nm channel) was quantified by Fiji.

### Statistical analysis

All statistical analyses were performed using GraphPad Prism 10. Data were first analyzed for normality using Shapiro–Wilk test. The normally distributed data are presented as mean ± SD, and unpaired two-tailed Student's *t* test was used for comparison between two groups. The non-normally distributed data are presented as mean ± SD (in superplots) or median with inter-quartile range (in box plots), with Mann–Whitney *U* test used for comparison between two groups.

The effects of the independent variable (genotype) on two dependent variables (distance and pH) were analyzed using multivariate analysis of variance (MANOVA). The analysis was performed in Rstudio (R version 4.4.1), and multivariate significance was assessed using Pillai's trace statistic.

Sample sizes are stated in the relevant figure legends, and the exact *P* values (unless <0.0001) are shown in the figures.

## Data availability

This study includes no data deposited in external repositories.

The source data of this paper are collected in the following database record: biostudies:S-SCDT-10_1038-S44318-025-00654-3.

## Peer review information

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

## Acknowledgements

We thank Dr. Anbing Shi (Huazhong University of Science and Technology) for generously sharing the PLA kit. We thank Drs. Di Wu (Wuhan University) and Ming-Liang Tang (Wuhan University) for assisting with mass spectrometry analysis and FRAP analysis, respectively. This work was supported by grants from the National Natural Science Foundation of China (32293203), National Key R&D Program of China (2024YFA1802801), National Natural Science Foundation of China (32021003, 92354301, 32400564), and Natural Science Foundation of Hubei Province (2025AFA001, 2023AFA050) and the 111 Project of Ministry of Education of China (B16036).

## Author contributions

**Luyi Jiang**: Data curation; Formal analysis; Validation; Investigation; Visualization. **Jiangfen Tang**: Formal analysis; Validation; Investigation; Visualization. **Ya-Fen Zhang**: Investigation. **Wen-Xuan Zou**: Investigation. **Gang Deng**: Investigation. **Na Tian**: Investigation. **Xiaolu Zhao**: Resources. **Lei Han**: Software; Formal analysis; Validation. **Kai Liu**: Data curation; Formal analysis; Validation; Methodology. **Bao-Liang Song**: Conceptualization; Supervision; Funding acquisition; Project administration; Writing—review and editing. **Jie Luo**: Conceptualization; Supervision; Funding acquisition; Visualization; Writing—original draft; Project administration; Writing—review and editing.

Source data underlying figure panels in this paper may have individual authorship assigned. Where available, figure panel/source data authorship is listed in the following database record: biostudies:S-SCDT-10_1038-S44318-025-00654-3.

## Disclosure and competing interests statement

The authors declare no competing interests.

# Expanded View Figures

**Figure EV1.   Identification of proteins at ER–lysosome MCSs using the split-TurboID in proximity labeling assay, related to Fig. 1.**

(**A**) Representative confocal images showing U2OS cells transduced with lentiviruses expressing TurboID (C terminus)-EGFP-Sec61β (white) and LAMP1-mCherry-TurboID (N terminus) (green), treated with 100 μM biotin for 4 h and then stained with streptavidin-647 (magenta). Boxed areas are enlarged as numbered and shown on the right. Scale bars, 5 μm (main), 1 μm (inset). (**B**) Workflow of streptavidin affinity purification coupled to mass spectrometry (MS) performed in HEK293T cells. (**C**) Immunoblotting analysis showing biotinylation in post nuclear supernatants (PNS), supernatants (S) and pellets (P) from HEK293T cells transfected as indicated and treated with 100 μM biotin for 4 h. (**D**) Top 18 hits identified at ER–lysosome MCSs. (**E**) Verification of knockdown efficiency of candidates in U2OS cells analyzed by quantitative real-time PCR. Data are presented as mean ± SD ($n = 3$ independent experiments). Unpaired two-tailed Student's $t$ test. $P$ values from left to right: 0.0002, <0.0001, <0.0001, 0.0002, <0.0001, 0.0001, <0.0001, 0.0009, <0.0001, 0.0001, 0.0002, <0.0001, <0.0001, <0.0001, 0.0003, 0.0004, 0.0002, <0.0001, 0.0003. (**F**) Representative confocal images showing U2OS cells stably expressing shRNAs targeting indicated genes, fixed and immunostained with anti-calnexin and anti-LAMP1 antibodies followed by proximity ligation assay. Scale bars, 10 μm. (**G**) Superplots showing the number of PLA puncta per cell (small dots) and its mean per independent experiment (large dots). Means and error bars (SD) are shown as black bars. # of cells: 21 per condition; from three independent experiments. Unpaired two-tailed Student's $t$ test. $P$ values from left to right: 0.0057, 0.8821, 0.0119, 0.0006, 0.0038, 0.9349, 0.7540, 0.5507, 0.8070, 0.7957, 0.5766, 0.0211, 0.0125, 0.5876, 0.0177, 0.0071, 0.6837, 0.0155, 0.0146. (**H**) SnapGene images showing that the exon 4 of *AREL1* genome was targeted by sgRNA (top) and Sanger sequencing of *AREL1* knockout (KO) U2OS cells, in which an extra T insertion resulted in premature termination of protein translation (bottom). (**I**) Verification of *AREL1* knockout in U2OS cells by immunoblotting. (**J**) Immunoblotting analysis showing biotinylation in WT and *AREL1*$^{-/-}$ U2OS cells were transfected as indicated and treated with indicated concentrations of biotin for 4 h. (**K**) Representative confocal images showing PLA signals in WT and *AREL1*$^{-/-}$ U2OS cells. Cells were fixed and immunostained with anti-calnexin and anti-LAMP1 antibodies followed by proximity ligation assay. Scale bars, 10 μm. (**L**) Superplots showing the number of PLA puncta per cell (small dots) and its mean per independent experiment (large dots). Means and error bars (SD) are shown as black bars. # of cells: WT, 82 and *AREL1*$^{-/-}$, 76; from 3 independent experiments. Unpaired two-tailed Student's $t$ test. WT cells vs *AREL1*$^{-/-}$ cells, $P < 0.0001$. (**M**) Knockdown efficiency of *AREL1* in HEK293T cells analyzed by quantitative real-time PCR. Data are presented as mean ± SD ($n = 3$ independent experiments). Unpaired two-tailed Student's $t$ test. shNC cells vs sh*AREL1* cells, $P = 0.0015$. (**N**) Representative confocal images showing PLA signals in shNC and sh*AREL1* HEK293T cells. Cells were fixed and immunostained with anti-calnexin and anti-LAMP1 antibodies followed by proximity ligation assay. Scale bars, 10 μm. (**O**) Superplots showing the number of PLA puncta per cell (small dots) and its mean per independent experiment (large dots). Means and error bars (SD) are shown as black bars. # of cells: shNC, 60 and sh*AREL1*, 60; from 3 independent experiments. Unpaired two-tailed Student's $t$ test. shNC cells vs sh*AREL1* cells, $P < 0.0001$ Source data are available online for this figure.

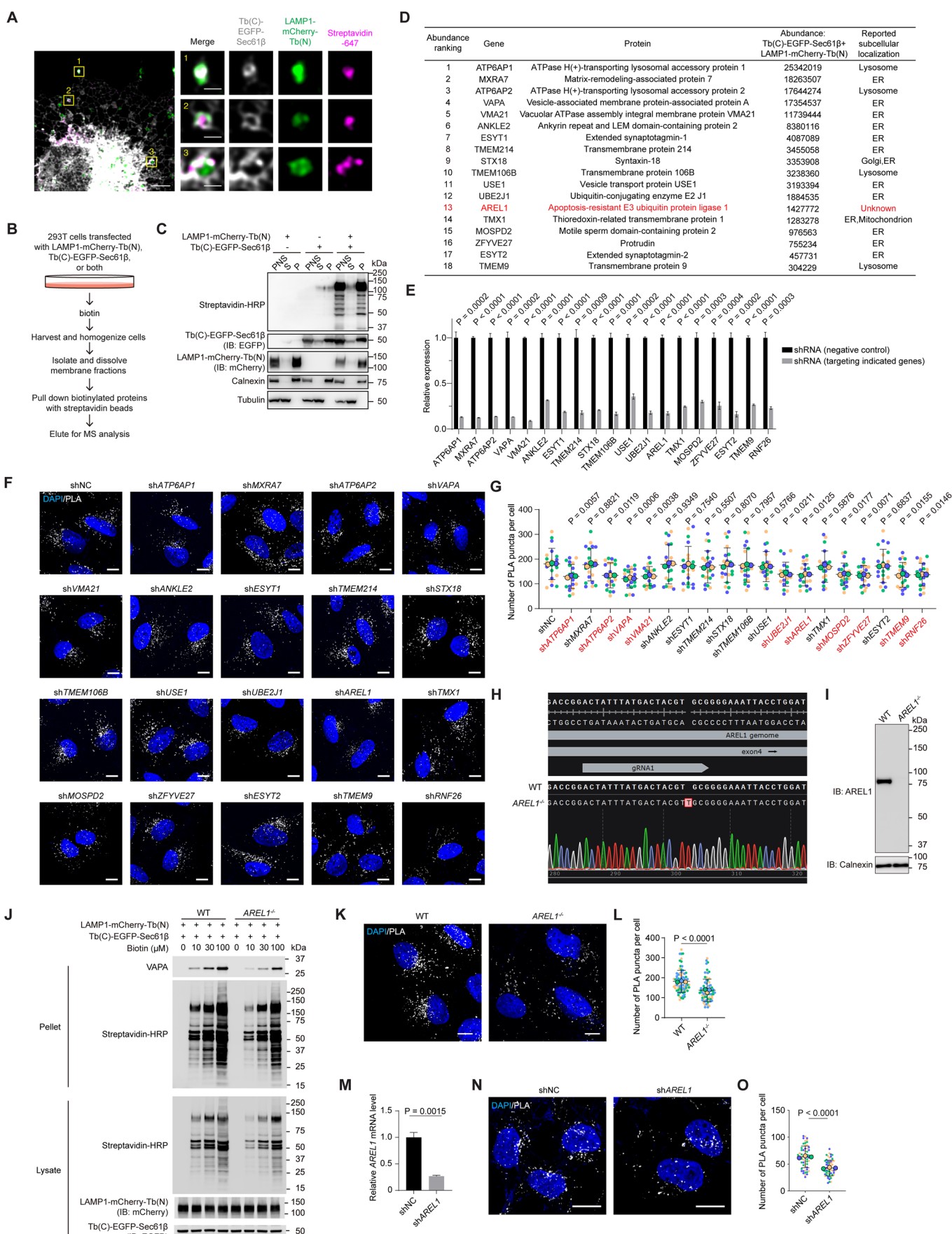

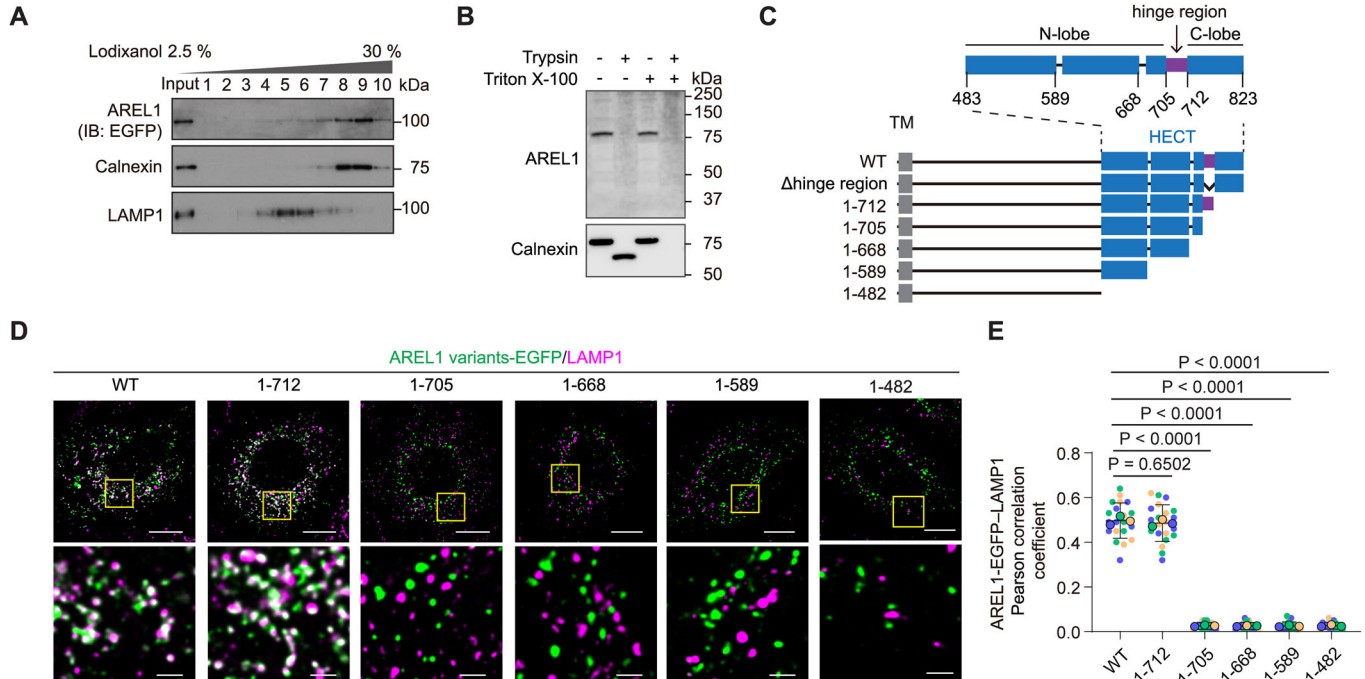

**Figure EV2. AREL1 is an ER protein and interacts with lysosomes via the hinge region, related to Fig. 1.**

(A) Immunoblotting analysis of membrane fractions isolated from U2OS cells transfected with the plasmid expressing AREL1-EGFP and subjected to iodixanol density gradient centrifugation. (B) Immunoblotting analysis of U2OS cell lysates treated with (+) or without (−) 0.1 μg/ μl trypsin or 0.1% Triton X-100. (C) Schematic illustration of AREL1 variants. (D) Representative confocal images showing the localization of AREL1 variants relative to lysosomes. U2OS cells were transduced with lentiviruses expressing indicated AREL1 variants tagged with EGFP and immunostained with anti-LAMP1 antibody. Boxed areas are enlarged and shown on the bottom. Scale bars, 10 μm (main), 2 μm (inset). (E) Superplots showing Pearson's correlation coefficient for AREL1 variants and LAMP1 per cell (small dots) and its mean per independent experiment (large dots). Means and error bars (SD) are shown as black bars. # of cells: 21 for cells expressing AREL1(WT)-EGFP, AREL1(1-712)-EGFP, AREL1(1-705)-EGFP, AREL1(1-668)-EGFP, AREL1(1-589)-EGFP and AREL1(1–482)-EGFP, respectively; from 3 independent experiments. Unpaired two-tailed Student's *t* test. AREL1(WT)-EGFP vs AREL1(1-712)-EGFP, $P = 0.6502$; AREL1(WT)-EGFP vs AREL1(1-705)-EGFP, $P < 0.0001$; AREL1(WT)-EGFP vs AREL1(1-668)-EGFP, $P < 0.0001$; AREL1(WT)-EGFP vs AREL1(1-589)-EGFP, $P < 0.0001$; AREL1(WT)-EGFP vs AREL1(1–482)-EGFP, $P < 0.0001$ Source data are available online for this figure.

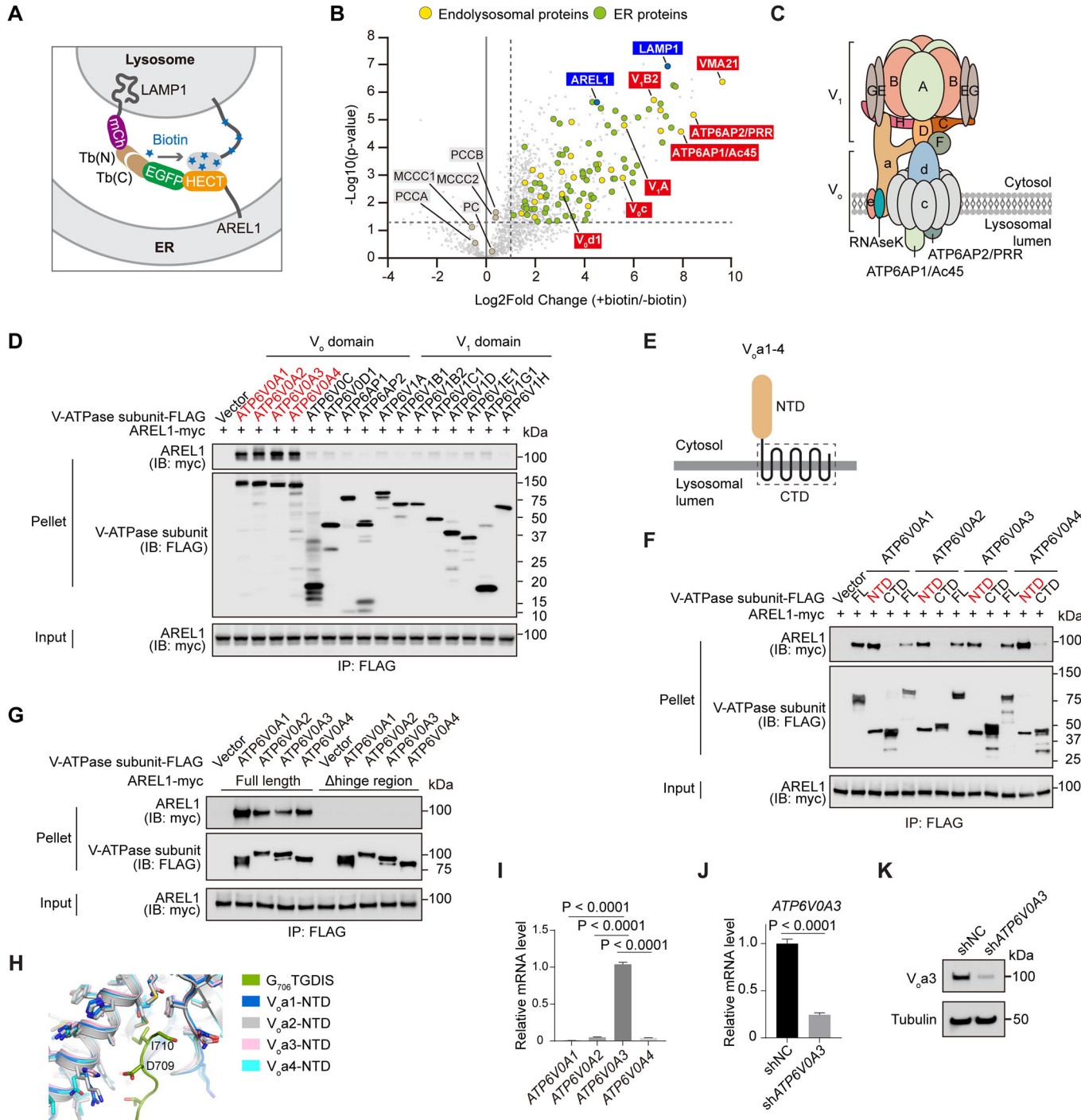

◄  **Figure EV3.   The hinge region of AREL1 interacts with the amino-terminal domain of $V_o$a subunit of V-ATPase, related to Fig. 1.**

(**A**) Schematic illustration of the split-TurboID-based proximity labeling assay to identify AREL1 binding partners. Tb(N), TurboID (N terminus); Tb(C), TurboID (C terminus); mCh, mCherry. (**B**) Volcano plot showing proteins enriched at AREL1-mediated ER–lysosome membrane contact sites. U2OS cells expressing LAMP1-mCherry-TurboID (N terminus) and TurboID (C terminus)-EGFP-AREL1 were treated with ($+$) or without ($-$) 100 μM biotin for 4 h. Biotinylated membrane proteins were enriched and subjected to mass spectrometry analysis. PC (Pyruvate carboxylase), PCCA (Propionyl-CoA carboxylase subunit a), PCCB (Propionyl-CoA carboxylase subunit b), MCCC1 (Methylcrotonoyl-CoA carboxylase subunit a) and MCCC2 (Methylcrotonoyl-CoA carboxylase beta chain) were served as the internal references for normalization. Four independent experiments were performed. A two-sample $t$ test was conducted to calculate $p$-values using the implemented function in Perseus. Significantly enriched ($Log_2$(Fold Change) > 1 and $-Log_{10}$($P$ value) > 1.3) endolysosomal proteins and ER proteins are in yellow and green dots, respectively, with each specific V-ATPase subunit highlighted. AREL1 and LAMP1 as the baits are displayed as blue dots. (**C**) Schematic illustration of the mammalian V-ATPase. (**D**) Co-immunoprecipitation (IP) analysis of HEK293T cells transfected as indicated. (**E**) Schematic illustration of human $V_o$a isoforms 1–4. NTD, N-terminal domain; CTD, C-terminal domain. (**F, G**) Co-IP analysis of HEK293T cells transfected as indicated. (**H**) AlphaFold 3-predicted interactions between the hinge region of AREL1 ($G_{706}$TGDIS) and the NTDs of four $V_o$a isoforms. The NTDs of $V_o$a1-4 are in marine, gray, lightpink, cyan, respectively, and $G_{706}$TGDIS is in splitpea. (**I**) Relative expression levels of indicated *ATP6V0A* in U2OS cells analyzed by quantitative real-time PCR. Data are presented as mean ± SD ($n = 6$ independent experiments). Unpaired two-tailed Student's $t$ test. *ATP6V0A1* vs *ATP6V0A3*, $P < 0.0001$; *ATP6V0A2* vs *ATP6V0A3*, $P < 0.0001$; *ATP6V0A4* vs *ATP6V0A3*, $P < 0.0001$. (**J**) Verification of knockdown efficiency of *ATP6V0A3* in U2OS cells analyzed by quantitative real-time PCR. Data are presented as mean ± SD ($n = 3$ independent experiments). Unpaired two-tailed Student's $t$ test. shNC cells vs sh*ATP6V0A3* cells, $P < 0.0001$. (**K**) Verification of *ATP6V0A3* knockdown in U2OS cells by immunoblotting Source data are available online for this figure.

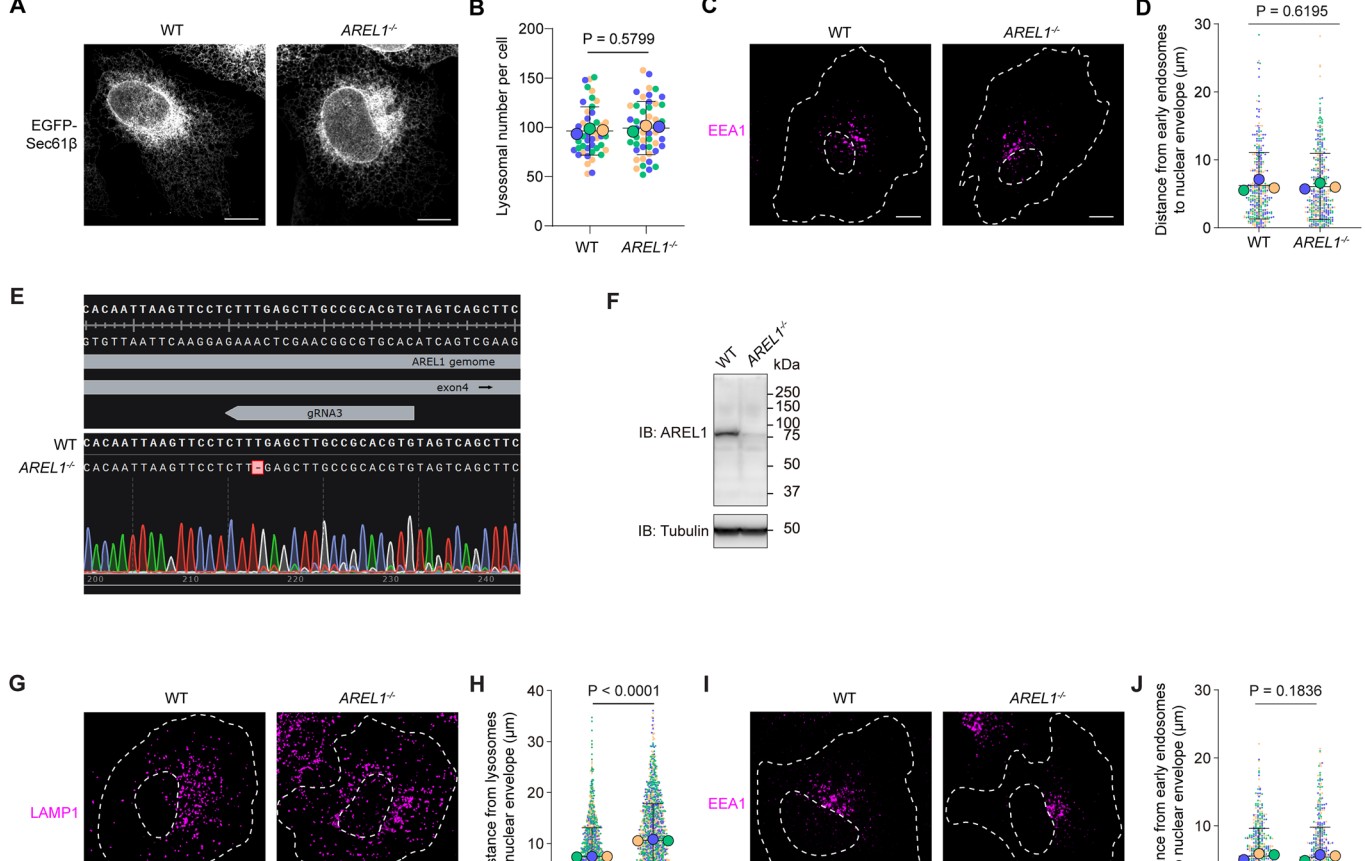

**Figure EV4.    The effects of AREL1 deficiency on ER morphology, lysosomal numbers, as well as the distribution of lysosomes and early endosomes, related to Fig. 2.**

(**A**) Representative confocal images showing ER distribution in WT and *AREL1*−/− U2OS cells. Scale bars, 10 μm. (**B**) Superplots showing the number of lysosomes per cell (small dots) and its mean per independent experiment (large dots). Means and error bars (SD) are shown as black bars. # of cells: WT, 60 and *AREL1*−/−, 60; from 3 independent experiments. Unpaired two-tailed Student's *t* test. WT cells vs *AREL1*−/− cells, *P* = 0.5799. (**C**) Representative confocal images showing WT and *AREL1*−/− U2OS cells immunostained with anti-EEA1 antibody. Cell contour and nucleus are outlined using white dashed lines. Scale bars, 10 μm. (**D**) Superplots showing the distance from early endosomes to the nuclear envelope (small dots) and its mean per independent experiment (large dots). Means and error bars (SD) are shown as black bars. # of cells (# of early endosomes): WT, 9 (369) and *AREL1*−/−, 9 (410); from 3 independent experiments. Mann–Whitney *U* test. WT cells vs *AREL1*−/− cells, *P* = 0.6195. (**E**) SnapGene images showing that the exon 4 of *AREL1* genome was targeted by sgRNA (top) and Sanger sequencing of *AREL1* knockout (KO) HeLa cells, in which a T deletion resulted in premature termination of protein translation (bottom). (**F**) Verification of *AREL1* knockout in HeLa cells by immunoblotting. (**G**) Representative confocal images showing WT and *AREL1*−/− HeLa cells immunostained with anti-LAMP1 antibody. Cell contour and nucleus are outlined using white dashed lines. Scale bars, 10 μm. (**H**) Superplots showing the distance from lysosomes to the nuclear envelope (small dots) and its mean per independent experiment (large dots). Means and error bars (SD) are shown as black bars. # of cells (# of lysosomes): WT, 11 (1353) and *AREL1*−/−, 11 (1566); from 3 independent experiments. Mann–Whitney *U* test. WT cells vs *AREL1*−/− cells, *P* < 0.0001. (**I**) Representative confocal images showing WT and *AREL1*−/− HeLa cells immunostained with anti-EEA1 antibody. Cell contour and nucleus are outlined using white dashed lines. Scale bars, 10 μm. (**J**) Superplots showing the distance from early endosomes to the nuclear envelope (small dots) and its mean per independent experiment (large dots). Means and error bars (SD) are shown as black bars. # of cells (# of early endosomes): WT, 9 (531) and *AREL1*−/−, 9 (528); from 3 independent experiments. Mann–Whitney *U* test. WT cells vs *AREL1*−/− cells, *P* = 0.1836 Source data are available online for this figure.

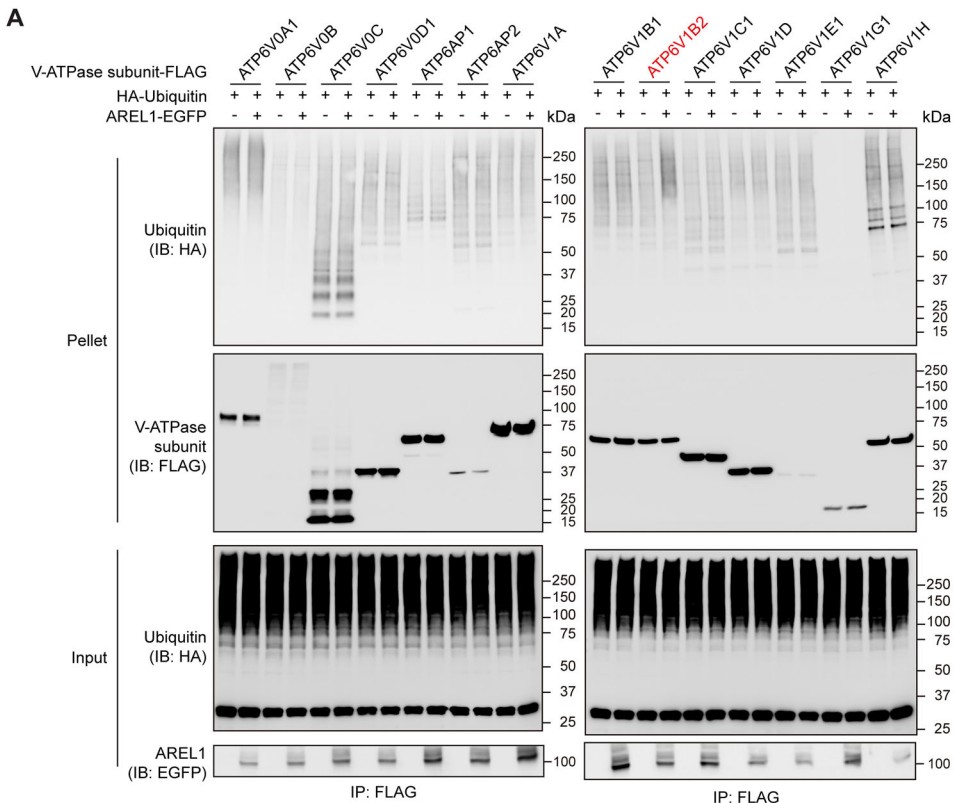

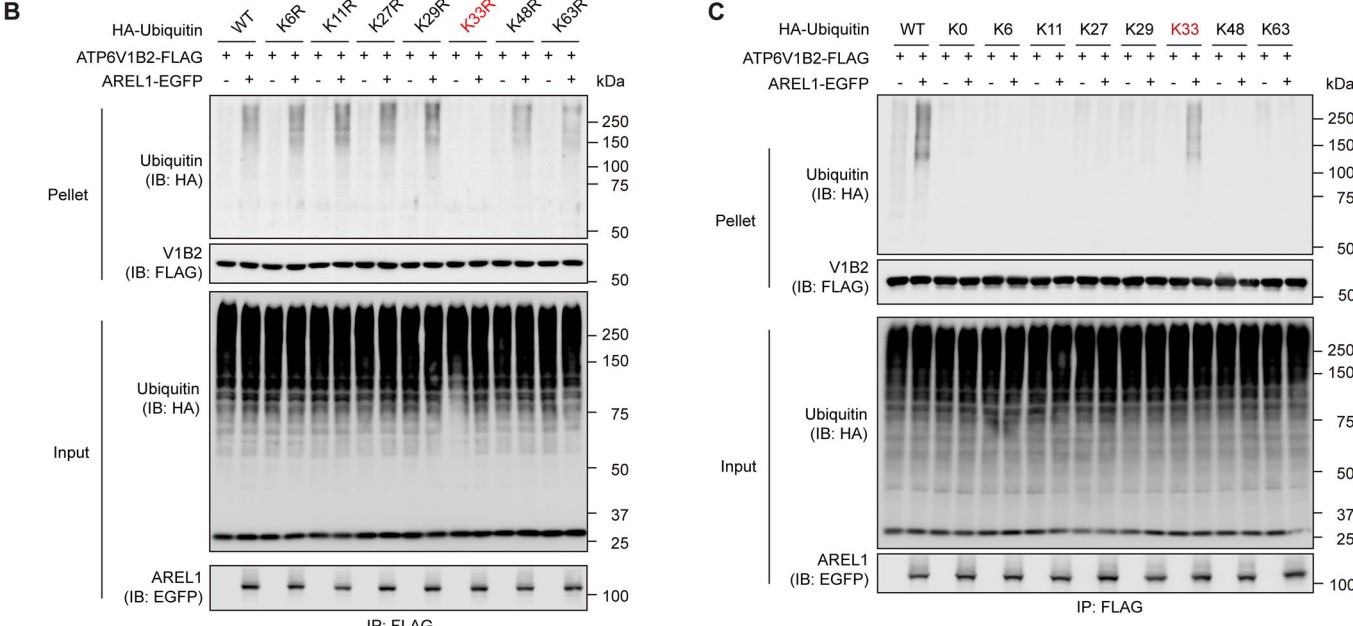

**Figure EV5. AREL1 selectively catalyzes K33-linked ubiquitylation of the V₁B2 subunit of the V-ATPase, related to Fig. 4.**

(A–C) HEK293T cells were transfected as indicated and subjected to IP with anti-FLAG beads followed by immunoblotting to analyze ubiquitylation Source data are available online for this figure.

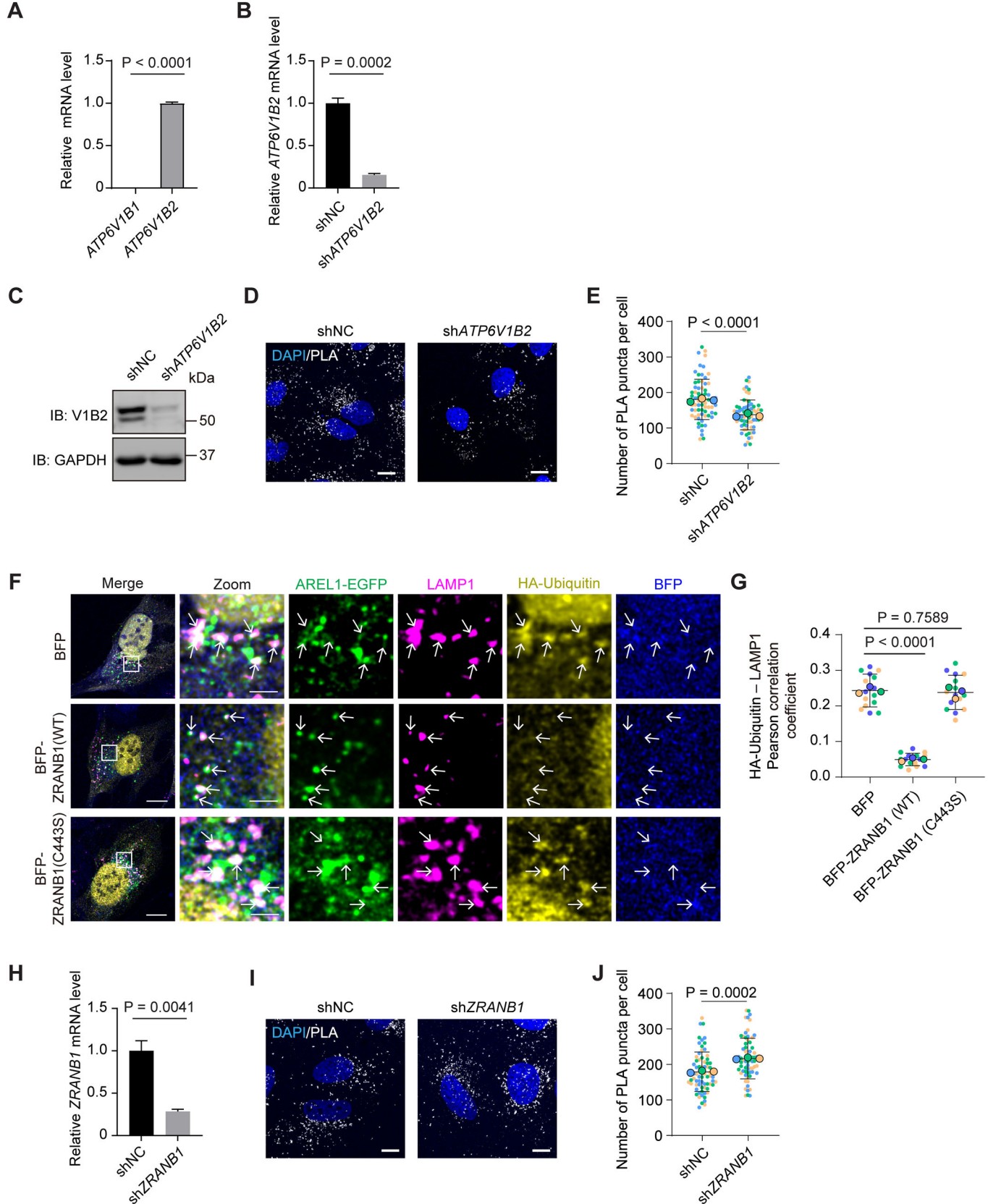

◄ **Figure EV6.   Characterization of the effects of *ATP6V1B2* and *ZRANB1* deficiency on lysosomal contacts with the ER and those of ZRANB1 overexpression on lysosomal ubiquitin signals, related to Fig. 4.**

(**A**) Relative expression levels of *ATP6V1B1* and *ATP6V1B2* in U2OS cells analyzed by quantitative real-time PCR. Data are presented as mean ± SD ($n = 3$ independent experiments). Unpaired two-tailed Student's $t$ test. *ATP6V1B1* vs *ATP6V1B2*, $P < 0.0001$. (**B**) Verification of knockdown efficiency of *ATP6V1B2* in U2OS cells analyzed by quantitative real-time PCR. Data are presented as mean ± SD ($n = 3$ independent experiments). Unpaired two-tailed Student's $t$ test. shNC cells vs sh*ATP6V1B2* cells, $P = 0.0002$. (**C**) Verification of *ATP6V1B2* knockdown in U2OS cells by immunoblotting. (**D**) Representative confocal images showing PLA signals in shNC and sh*ATP6V1B2* U2OS cells. Cells were fixed and immunostained with anti-calnexin and anti-LAMP1 antibodies followed by proximity ligation assay. Scale bars, 10 μm. (**E**) Superplots showing the number of PLA puncta per cell (small dots) and its mean per independent experiment (large dots). Means and error bars (SD) are shown as black bars. # of cells: shNC, 71 and sh*ATP6V1B2*, 69; from 3 independent experiments. Unpaired two-tailed Student's $t$ test. shNC cells vs sh*ATP6V1B2* cells, $P < 0.0001$. (**F**) Representative confocal images showing U2OS cells transduced with lentiviruses expressing AREL1-EGFP, HA-tagged ubiquitin, together with those expressing blue fluorescent protein (BFP) alone or with BFP-tagged ZRANB1 variants, followed by immunostaining with anti-LAMP1 and anti-HA tag antibodies. Boxed areas are enlarged on the right. White arrows indicate contacts between AREL1-EGFP and LAMP1. Scale bars, 10 μm. (**G**) Superplots showing Pearson's correlation coefficient for HA-ubiquitin and LAMP1 per cell (small dots) and its mean per independent experiment (large dots). Means and error bars (SD) are shown as black bars. # of cells: 15 for U2OS cells expressing BFP, BFP-ZRANB1(WT), and BFP-ZRANB1(C443S), respectively; from 3 independent experiments. Unpaired two-tailed Student's $t$ test. Cells expressing BFP vs cells expressing BFP-ZRANB1(WT), $P < 0.0001$; Cells expressing BFP vs cells expressing BFP-ZRANB1(C443S), $P = 0.7589$. (**H**) Verification of knockdown efficiency of *ZRANB1* in U2OS cells analyzed by quantitative real-time PCR. Data are presented as mean ± SD ($n = 3$ independent experiments). Unpaired two-tailed Student's $t$ test. shNC cells vs sh*ZRANB1* cells, $P = 0.0041$. (**I**) Representative confocal images showing PLA signals in shNC and sh*ZRANB1* U2OS cells. Cells were fixed and immunostained with anti-calnexin and anti-LAMP1 antibodies followed by proximity ligation assay. Scale bars, 10 μm. (**J**) Superplots showing the number of PLA puncta per cell (small dots) and its mean per independent experiment (large dots). Means and error bars (SD) are shown as black bars. # of cells: shNC, 73 and sh*ZRANB1*, 74; from 3 independent experiments. Unpaired two-tailed Student's $t$ test. shNC cells vs sh*ZRANB1* cells, $P = 0.0002$ Source data are available online for this figure.

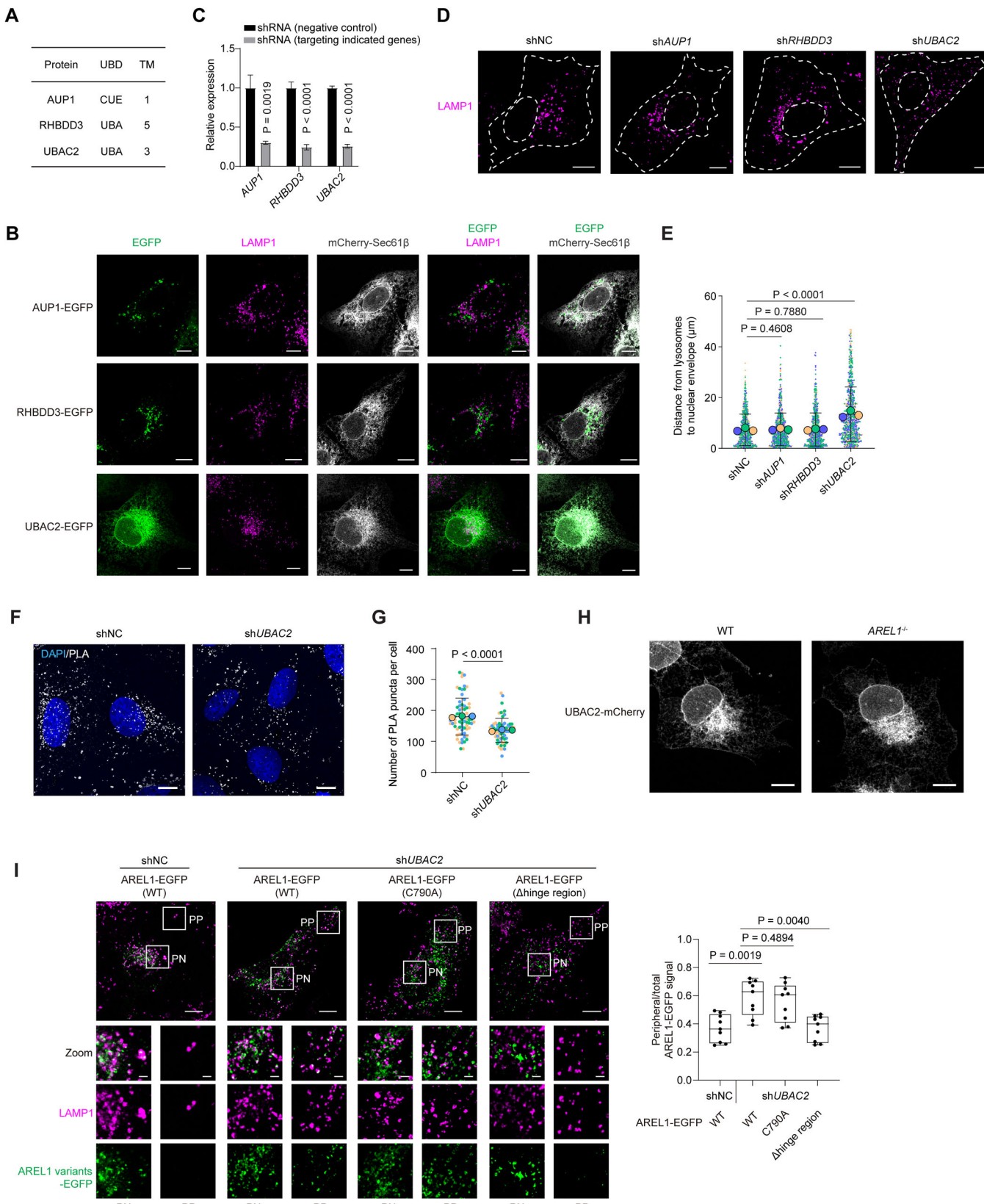

◀ **Figure EV7. The effects of UBAC2 deficiency on lysosomal positioning, ER–lysosome MCSs, and AREL1 subcellular distribution, related to Fig. 5.**

(A) List of ubiquitin-binding proteins with transmembrane domains. CUE coupling of ubiquitin conjugation to ER degradation, UBA Ubiquitin-associated, UBD ubiquitin-binding domain, TM transmembrane. (B) Representative confocal images showing the localization of AUP1, RHBDD3 and UBAC2. U2OS cells were transduced with lentiviruses expressing mCherry-Sec61β together with AUP1-EGFP, RHBDD3-EGFP, or UBAC2-EGFP and then immunostained with anti-LAMP1 antibody. Scale bars, 10 μm. (C) Verification of knockdown efficiency of *AUP1, RHBDD3 and UBAC2* in U2OS cells analyzed by quantitative real-time PCR. *P* values from left to right: 0.0019, <0.0001, <0.0001. (D) Representative confocal images showing the distribution of lysosomes in U2OS cells transduced with lentiviruses encoding negative control shRNA (shNC) and shRNA against indicated genes. Cell contour and nucleus are outlined using white dashed lines. Scale bars, 10 μm. (E) Superplots showing the distance from lysosomes to the nuclear envelope (small dots) and its mean per independent experiment (large dots). Means and error bars (SD) are shown as black bars. # of cells (# of lysosomes): shNC, 9 (678); sh*AUP1*, 9 (732); sh*RHBDD3*, 9 (677) and sh*UBAC2*, 9 (770); from 3 independent experiments. Mann–Whitney *U* test. shNC cells vs sh*AUP1* cells, *P* = 0.4608; shNC cells vs sh*RHBDD3* cells, *P* = 0.7880; shNC cells vs sh*UBAC2* cells, *P* < 0.0001. (F) Representative images showing PLA signals in shNC and sh*UBAC2* U2OS cells. Cells were fixed and immunostained with anti-calnexin and anti-LAMP1 antibodies followed by proximity ligation assay. Scale bars, 10 μm. (G) Superplots showing the number of PLA puncta per cell (small dots) and its mean per independent experiment (large dots). Means and error bars (SD) are shown as black bars. # of cells: shNC, 69 and sh*UBAC2*, 67; from 3 independent experiments. Unpaired two-tailed Student's *t* test. shNC cells vs sh*UBAC2* cells, *P* < 0.0001. (H) Representative confocal images showing the distribution of UBAC2-mCherry in WT and *AREL1*−/− U2OS cells. Scale bars, 10 μm. (I) Representative confocal images showing the localization of AREL1-EGFP variants and lysosomes in shNC and sh*UBAC2* U2OS cells. Boxed areas are enlarged on the bottom. Scale bars, 10 μm (main), 2 μm (inset). Right is box plots showing relative AREL1-EGFP signal in the peripheral region (3 outermost shells over 5 total shells per cell). Data are presented as median with interquartile range. Each box-and-whisker consists of the 25th quantile (the upper border of box), median (horizontal line inside the box), 75th quantile (the lower border of box), and vertical lines extending to the minimum and maximum values. # of cells: 9 for shNC cells expressing AREL1(WT)-EGFP, sh*UBAC2* cells expressing AREL1(WT)-EGFP, sh*UBAC2* cells expressing AREL1(C790A)-EGFP and sh*UBAC2* cells expressing AREL1(hinge region)-EGFP, respectively; from 3 independent experiments. Mann–Whitney *U* test. shNC cells expressing AREL1(WT)-EGFP vs sh*UBAC2* cells expressing AREL1(WT)-EGFP, *P* = 0.0019; sh*UBAC2* cells expressing AREL1(WT)-EGFP vs sh*UBAC2* cells expressing AREL1(C790A)-EGFP, *P* = 0.4894; sh*UBAC2* cells expressing AREL1(WT)-EGFP vs sh*UBAC2* cells expressing AREL1(hinge region)-EGFP, *P* = 0.0040 Source data are available online for this figure.

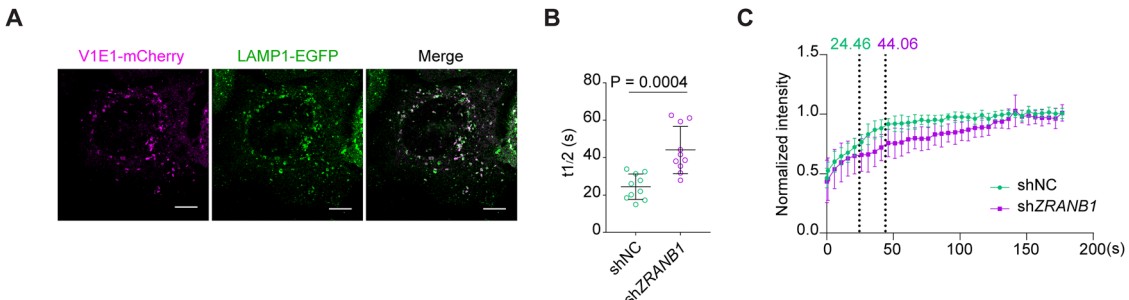

**Figure EV8. The effects of ZRANB1 deficiency on V-ATPase assembly.**

(A) Representative confocal images showing the localization of $V_1E1$-mCherry in U2OS cell. Scale bar, 10 μm. (B) Quantification of half times of fluorescence recovery ($\tau_{1/2}$) of $V_1E1$-mCherry. Data are presented as mean ±SD. # of cells: shNC, 9 and sh*ZRANB1*, 9; from 2 independent experiments. Unpaired two-tailed Student's *t* test. shNC cells vs sh*ZRANB1* cells, $P = 0.0004$. (C) Recovery curves of $V_1E1$-mCherry normalized to the fluorescence signal after 180 s. Data are presented as mean ± SD. # of cells: shNC, 10 and sh*ZRANB1*, 10; from 2 independent experiments Source data are available online for this figure.

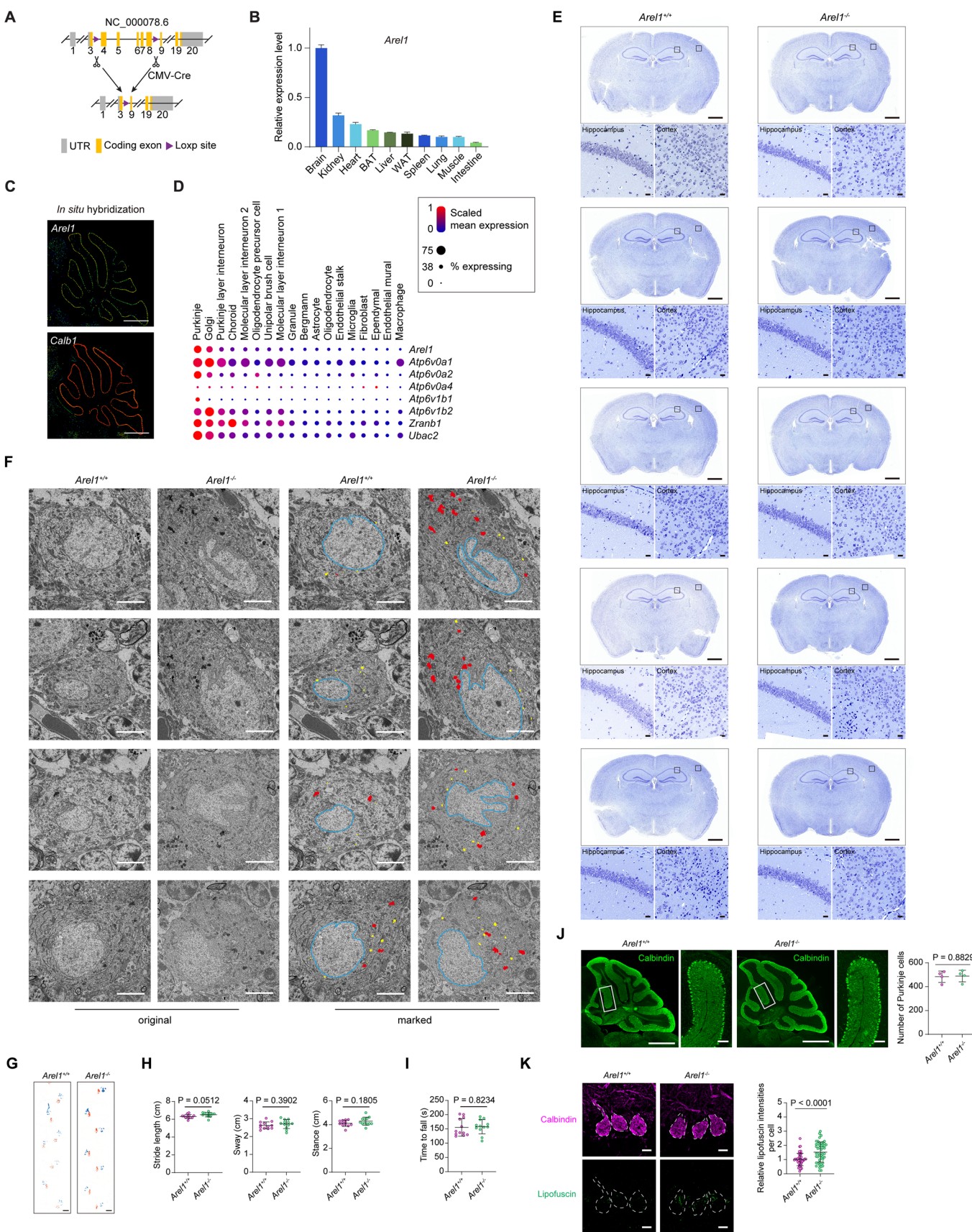

◄

**Figure EV9.  Characterization of AREL1 expression in mouse tissues and *Arel1⁻/⁻* mouse phenotypes, related to Fig. 6.**

(A) *Arel1* knockout strategy. *Arel1*^floxed/floxed^ mice were crossed with CMV-Cre transgenic mice to generate *Arel1⁻/⁻* mice. (B) Quantitative real-time PCR analysis showing *Arel1* mRNA levels in different mouse tissues. Data are presented as means ± SD ($n = 3$ biological replicates of the pooled samples from three 2-month-old C57/BL6J mice). (C) In situ hybridization image data retrieved from the Allen Mouse Brain Atlas (https://mouse.brain-map.org/) showing *Arel1* and *Calb1* expression in mouse cerebellum. Scale bars, 1 mm. (D) Dot plot showing expression of indicated genes in 18 cell types of adult mouse cerebellum. Data were retrieved from the work by Kozareva et al, 2021 and analyzed in Single Cell Portal. Dot size encodes percentage of cells expressing the gene, and color encodes the average per cell gene expression level. (E) Representative Nissl staining of coronal brain sections from five 12-month-old *Arel1⁺/⁺* and *Arel1⁻/⁻* male mice per genotype. The hippocampal CA1 region and cerebral cortex are enlarged and shown on the bottom. Scale bars, 1 mm (main), 100 µm (inset). (F) Transmission electron micrographs (original and color-labeled) showing Purkinje cells in 12-month-old *Arel1⁺/⁺* and *Arel1⁻/⁻* male mice. Nuclear contours are outlined by blue lines and lysosomes and lipofuscin granules marked in yellow and red, respectively. Scale bars, 5 µm. (G) Representative footprints of 6-month-old *Arel1⁺/⁺* and *Arel1⁻/⁻* male mice. Scale bars, 2 cm. (H) Stride length, sway and stance of 6-month-old *Arel1⁺/⁺* and *Arel1⁻/⁻* male mice. Data are presented as mean ± SD ($n = 12$ for both *Arel1⁺/⁺* and *Arel1⁻/⁻* mice). Unpaired two-tailed Student's *t* test. Stride length in *Arel1⁺/⁺* mice vs stride length in *Arel1⁻/⁻* mice, $P = 0.0512$; sway in *Arel1⁺/⁺* mice vs sway in *Arel1⁻/⁻* mice, $P = 0.3902$; stance in *Arel1⁺/⁺* mice vs stance in *Arel1⁻/⁻* mice, $P = 0.1805$. (I) Time to fall off the rods of 6-month-old *Arel1⁺/⁺* and *Arel1⁻/⁻* male mice. Data are presented as mean± SD ($n = 12$ for both *Arel1⁺/⁺* and *Arel1⁻/⁻* mice). Unpaired two-tailed Student's *t* test. *Arel1⁺/⁺* mice vs *Arel1⁻/⁻* mice, $P = 0.8234$. (J) Representative confocal images showing calbindin expression in the cerebellum of 6-month-old *Arel1⁺/⁺* and *Arel1⁻/⁻* male mice. Boxed areas are enlarged and shown on the right. Scale bars, 1 mm (main), and 50 µm (inset). Quantification of calbindin-positive Purkinje cells is presented as mean ±SD ($n = 4$ mice per genotype). Unpaired two-tailed Student's *t* test. *Arel1⁺/⁺* mice vs *Arel1⁻/⁻* mice, $P = 0.8829$. (K) Representative confocal images showing calbindin staining (magenta) and lipofuscin autofluorescence (green) in Purkinje cells of 6-month-old *Arel1⁺/⁺* and *Arel1⁻/⁻* male mice. Scale bars, 10 µm (main). Quantification of lipofuscin autofluorescence in calbindin-positive Purkinje cells is presented as mean ± SD ($n = 53$ and 50 cells for *Arel1⁺/⁺* and *Arel1⁻/⁻* male mice, respectively). Unpaired two-tailed Student's *t* test. *Arel1⁺/⁺* mice vs *Arel1⁻/⁻* mice, $P < 0.0001$ Source data are available online for this figure.

