## [Peer Review File · The EMBO Journal]

E3 ligase AREL1 controls perinuclear localization of lysosomes and supports Purkinje cell survival

Lu-Yi Jiang, Jiangfen Tang, Ya-Fen Zhang, Wen-Xuan Zou, Gang Deng, Na Tian, Xiaolu Zhao, Lei Han, Kai Liu, Bao-Liang Song, and Jie Luo

Corresponding author: Jie Luo (jjeluo@whu.edu.cn)

Review Timeline:

Submission Date:	3rd Sep 25
Editorial Decision:	17th Oct 25
Revision Received:	28th Oct 25
Accepted:	13th Nov 25

Editor: Ieva Gailite

Transaction Report:

Please note that the manuscript was previously reviewed at another journal. As EMBO Press has a transfer agreement with that journal, revision was invited based on the reports from that previous external submission.

Reviewers' comments:

Reviewer #1 (Remarks to the Author):

Jiang et al. have discovered a mechanism by which lysosomes are anchored to the endoplasmic reticulum. In their model, AREL1, an ER-localized E3 ligase, binds to the subunit of V-ATPase on the lysosome and catalyzes K33-linked ubiquitylation of ATP6V1B2, whose ubiquitin chain is then bound by UBAC2 on the ER, thus anchoring lysosomes to the ER via the interaction between ER-localized UBAC2 and K33-ubiquitylated V-ATPase on the lysosome. Based on their observation that AREL1 is highly expressed in the mouse brain, and ATP6V1B2, that is, the B2 isoform of the V1 domain of V-ATPase is enriched in the brain, the authors extend their study to an in vivo model. In *Arel1*^{-/-} mice, they find that upon reaching middle age, the mice exhibit cerebellar neurodegeneration potentially due to lysosomal malfunction in their Purkinje cells. Moreover, they find an association between hereditary cerebellar ataxia and AREL1 mutation in humans and can recapitulate their initial in vitro phenotype using the V50M mutation observed in some cases of human hereditary cerebellar ataxia.

The manuscript is well-written, and the data is of good quality for most panels. The mechanisms raised by the authors are interesting, and they will conceptually advance the field. However, a few points undermine the authors' current claims. The manuscript will significantly improve if the authors address these points.

Major comments

1. The authors found that AREL1, ZRANB1, UBAC2, and ATP6V1B2 are important for ER-lysosome contact and lysosome acidification. However, this was mainly shown by imaging analysis of cells over-expressing multiple components. An alternative method would be required to strengthen the authors' claim.

Response: We thank the reviewer for the insightful comments.

In the previous version of manuscript, *AREL1* knockout cells (original Fig. 2, 3 and S4) and *UBAC2* knockdown cells (original Fig. 4I-Q) were utilized to demonstrate the importance of AREL1 and UBAC2 in regulating perinuclear localization and degradative function of lysosomes.

As suggested by the reviewer, we have carried out additional experiments using *ATP6V1B2* knockdown cells (Extended Data Fig. 6b,c) and *ZRANB1* knockdown U2OS cells (Extended Data Fig. 6h). Silencing of *ATP6V1B2* reduced calnexin-LAMP1 PLA puncta (Extended Data Fig. 6d,e) and dispersed lysosomes to cell periphery (Fig. 4d,e), whereas that of *ZRANB1* caused the opposite phenotypes (Extended Data Fig. 6i,j and Fig. 4j,k).

Since V₁B2 and V₁A constitute the V₁ domain that is responsible for ATP binding and hydrolysis, the detrimental effects of *ATP6V1B2* deficiency on lysosomal acidity and degradative function were anticipated and therefore not examined. Silencing of *ZRANB1* markedly increased the fluorescence intensities of LysoSensor green and DQ-OVA (Fig. 4I-o).

These new results further support that the AREL1–ZRANB1–V-ATPase–UBAC2 axis orchestrates lysosomal positioning and degradative function.

a. For example, in Fig. 2A, the imaging analysis shows that lysosomes in the *AREL1*^{-/-} cells seem more peripheral. If the authors perform split-Turbo ID in these cell backgrounds with Tb(C) Sec61b

and Lamp1-mCherry-Tb(N), and enrich biotinylated proteins, would the authors see fewer biotinylated proteins as their contact is lost? Also, compared to WT cells after streptavidin enrichment, would a reduced level of proteins (such as VAPA, ATP6AP1,2, etc., shown in S1D) be found in AREL1-/- cells? This could be an example of unbiasedly confirming the authors' claim without using an over-expressive system. The result of this may convince the readers even further.

Response: We thank the reviewer for the helpful advice.

As suggested, we performed the split-Turbo ID-based in proximity labeling assay in WT and *AREL1* KO U2OS cells treated with increasing concentrations of biotin. The total levels of biotinylated proteins were reduced in *AREL1* KO cells at all the biotin concentrations used (Extended Data Fig. 1j). The protein levels of VAPA (vesicle-associated membrane protein-associated protein A) in the pellets were reduced as well. These results support the notion that AREL1 is responsible for establishing ER-lysosome MCSs. (Page 5)

b. Likewise, the effect of ZRANB1 and UBAC2 on ER-lysosome contact was shown by imaging analysis only. Proving this using a complementary approach will significantly strengthen the author's claim regarding the role of AREL1-ZRANB1-UBAC2-ATP6V1B2 on ER-lysosome contact.

Response: We thank the reviewer for the helpful advice.

As suggested, we performed additional experiments on various knockdown/knockout cells in combination with the proximity ligation assay (PLA) to corroborate the role of AREL1-ZRANB1-UBAC2-V1B2 on ER-lysosome contacts.

PLA allows the detection of two proteins at a distance of less than 40 nm and provides a quantitative measure of ER-lysosome MCSs (PMID: 17072308, 34315878, 31548609).

In *AREL1* KO U2OS cells, the number of calnexin-LAMP1 PLA puncta was significantly reduced (Extended Data Fig. 1k,l), suggestive of reduced ER-lysosome MCSs. We also generated *ATP6V1B2* knockdown cells (Extended Data Fig. 6b,c) and *ZRANB1* knockdown cells (Extended Data Fig. 6h). Depletion of *ATP6V1B2* reduced PLA puncta for LAMP1-calnexin interactions (Extended Data Fig. 6d,e), whereas silencing of *ZRANB1* increased PLA puncta numbers (Extended Data Fig. 6i,j). *UBAC2* knockdown cells showed reduced numbers of PLA puncta (Extended Data Fig. 7f,g).

These new results support that the AREL1-ZRANB1-UBAC2-V1B2 axis plays a critical role in regulating ER-lysosome MCSs.

2. Proteomics results/data were not shared, and the data analysis was not transparently reported.

Response: We have included the proteomics raw results of two split-Turbo ID-based in proximity labeling experiments as Supplementary Table 1 (related to Extended Data Fig. 1a) and Supplementary Table 2 (related to Extended Data Fig. 3a). The results from mass spectrometry were analyzed as below.

To identify proteins at ER-lysosome MCSs, we transfected HEK293T cells with plasmids expressing Tb(C)-EGFP-Sec61 β , LAMP1-mCherry-Tb(N), or both. As shown in Supplementary Table 1, of all the proteins identified (*the sheet named "Raw MS data"*), 85 harboring transmembrane domain(s) were profoundly enriched in cells transfected with both LAMP1-mCherry-Tb(N) and Tb(C)-EGFP-Sec61 β compared with cells transfected with either one (*the sheet named "Filtered results"*). Proteins reported to localize in the membranes other than the ER or lysosomes, or with well-established functions were

further excluded. Among 18 remaining candidates, most were well-known lysosomal or ER proteins (Extended Data Fig. 1d). Apoptosis-resistant E3 ubiquitin protein ligase 1 (AREL1) is of particular interest given its uncharacterized subcellular localization.

Eighteen candidates were listed in descending order of their abundance detected in cells transfected with both LAMP1-mCherry-Tb(N) and Tb(C)-EGFP-Sec61 β (Extended Data Fig. 1d). The effects of each of the 18 candidates plus RNF26 on ER-lysosome MCSs were evaluated in using PLA in shRNA knockdown cells (Extended Data Fig. 1e-g).

To identify the interacting partners of AREL1 on the lysosomal membrane, we employed the split-TurboID in proximity labeling strategy again, with the two enzyme halves fused with AREL1 and LAMP1, respectively (Extended Data Fig. 3a). The identified proteins (*the sheet named "Raw MS data"*) were first subjected to statistical analysis using Perseus. A total of 11 late endosome/lysosome proteins (LAMP1 as the bait was not counted) were significantly enriched following biotin treatment, and 6 among them were the subunits of V-ATPase (Extended Data Fig. 3b and Supplementary Table 2), with ATP6AP2/PRR, ATP6AP1/Ac45, V₁B2, V₀c, and V₁A being the top 5.

These descriptions have been included in the revised manuscript (Pages 4 and 6).

a. Fig. S1D lists the names of proteins, but there's no other information regarding the enrichment level or number of peptides identified.

Response: The revised Extended Data Fig. 1d shows 18 candidates in descending order of their abundance detected in cells transfected with both LAMP1-mCherry-Tb(N) and Tb(C)-EGFP-Sec61 β .

b. Fig. S3B also shows a volcano plot, but only ATPase subunits are highlighted in the plot. As ATPases are one of the most abundant proteins on the lysosomal membrane, detecting many subunits in their split-TurboID is not surprising. Sharing raw data would be crucial for readers to evaluate the analysis and approaches used and other factors enriched in their experiments (many hits are more enriched than the ATPase subunits).

Response: We have included the proteomics raw results for Extended Data Fig. 3b as Supplementary Table 2.

We agree with the reviewer that V-ATPases are abundant on the lysosomal membrane. However, the 6 V-ATPase subunits (ATP6AP2/PRR, ATP6AP1/Ac45, V₁B2, V₀c, V₁A and V₀d1) were specifically and significantly enriched following biotin treatment. In fact, ATP6AP2/PRR, ATP6AP1/Ac45, V₁B2, V₀c, and V₁A were the top 5 of 11 significantly enriched late endosome/lysosome proteins (LAMP1 as the bait was not counted) (Supplementary Table 2). Many of the hits that seemed to be more enriched than the V-ATPase subunits were ER proteins, as shown in Extended Data Fig. 3b in green circles.

Based on the enrichment of V-ATPase subunits in mass spectrometry, we further performed co-immunoprecipitation experiments to identify the exact subunit(s) that interacted with AREL1. All four isoforms of subunit a of the V₀ domain (V₀a1 to V₀a4) were co-immunoprecipitated with AREL1 (Extended Data Fig. 3d). In fact, it was the homologous amino-terminal domain (NTD) of V₀a and the hinge region (G₇₀₆TGDIS) of AREL1 that mediated the interaction between the two (Extended Data Fig. 3e-g). Using the V₀a3 isoform as a representative, we mapped out the critical amino acids mediating the interaction between the hinge region of AREL1 and the V₀a3 using co-IP experiments (Fig. 1o-q).

All these results support that V_oa is the bona fide binding partner of AREL1.

3. How the authors quantified the images is unclear, although imaging quant is the primary support for the authors' conclusions.

Response: The image quantification details are below and have also been included in the revised Methods section (Pages 27 and 34).

For vesicle distribution measurement, a total of 10 z-stacks were taken per cell from the top to the bottom and then merged. Images were exported with scale bars using Leica LAS X software. Images were then imported to Fiji and scale bars were used to set scale using the *Analyze>Set Scale* command. Straight lines were manually drawn from vesicles to the nuclear envelope, and the distances of straight lines were calculated using the *Analyze>Measure* command and recorded.

For fluorescence intensity quantification of LysoSensor green, SiR-lysosome, and DQ-OVA, cell borders were manually outlined using the *Freehand selections* tool in Fiji, and total fluorescence intensities within the borders were quantified using the *Analyze>Measure* command and recorded. Relative fluorescence intensities were calculated by comparing experimental groups with control groups.

For colocalization quantification, images were exported in separate color channels using Leica LAS X software. Images were then imported to Fiji. Cell borders were manually outlined using the *Freehand selections* tool in one channel, and the borders were applied to the other channel using the *Edit>Selection>Restore Selection* command. The Pearson's correlation coefficient of the two channels per cell was quantified by the Coloc 2 plugin and recorded.

PLA signal quantification was performed using Fiji as previously described (PMID: 34315878). The 561 nm-channel images were imported to Fiji and converted to binary images, and a threshold was applied to eliminate background using the *Image>Adjust>Threshold* command. The Watershed feature was applied to separate any signal dots apparently in touch using the *Process>Binary>Watershed* command. Dots that were at least 5 pixels in size were counted using the *Analyze>Analyze Particles* command, and the total number of dots per cell (counterstained by DAPI) was measured and recorded.

a. Did the authors take several z-steps and merge them for quantification? Or was one Z-section taken per cell? If latter, how did the authors decide the location of the z-plane? Clarifying this would be necessary as perinuclear lysosomes appear more on the top of the cells, whereas peripheral lysosomes appear more on the bottom.

Response: A total of 10 z-stacks were taken per cell from the top to the bottom and then merged to assess vesicle distribution. This information has been included in the revised Methods section (Page 34).

b. How were the cells selected? Were unbiased approaches used for cell selection or distance measurement? How varied were the over-expression levels for those for which Pearson correlation was measured?

Response: Cells were selected and analyzed as previously described (PMIDs: 34315878, 35819772). Those with flat and round shapes were selected, and the narrow ones were excluded. In the cases of

over-expression, cells expressing medium and similar levels of indicated proteins were chosen for analysis, and those showing high expression levels were excluded to avoid artifacts.

The above descriptions have been included in the Method (Pages 33 and 34).

c. There are many imaging experiments where the sample size is counted by the number of lysosomes quantified; however, adding the number of unique cells used for the quantification will be informative. There can be cell-to-cell variation, so quantifying a small number of cells can skew the data, even if there are many lysosomes in each cell. This is particularly important as the authors use transient overexpression systems where there will be varying degrees of protein expression in the transfected vector from cell to cell.

Response: We agree with the reviewer that quantification is vital for the study. We have taken the advice and analyzed many more cells and lysosomes in each cellular experiment as previously described (PMIDs: 27368102, 34315878, 35314674, 33472082, 37519262). The quantification results per condition were from three independent experiments. The exact numbers of lysosomes, cells and repeats have been clearly stated in the relevant figure legends. All these new quantification results support our conclusions that the AREL1–UBAC2–V₁B2–ZRANB1 axis regulates lysosomal positioning and function.

4. Use of an over-expression system for studying the selectivity of E3 and DUB.

a. AREL1 and ZRANB1 are extremely lowly expressed proteins based on label-free quantification (<https://opencell.czbiohub.org/search/arel1>). So, showing the overall phenotypes with endogenous protein levels will be critical throughout the study.

Response: We thank the reviewer for the helpful advice.

Both AREL1 and ZRANB1 express at low levels in HEK293T cells according to the OpenCell. Their expression levels in U2OS cells are about twice as high as that in HEK293T cells (Figure 1-for reviewer). The Ct values of AREL1 and ZRANB1 were 28.14 and 27.02, respectively, in U2OS cells. In the same qPCR analysis, the Ct value of GAPDH was 18.61.

Figure 1-for reviewer. Endogenous AREL1 and ZRANB1 levels in HEK293T and U2OS cells determined by quantitative real-time PCR. Relative expression levels were calculated by comparing the Ct values (U2OS/HEK293T) after normalizing with GAPDH. Data are presented as mean±SD (n = 3 independent experiments).

As shown in Figs. 2 and 3, we have utilized AREL1 KO U2OS cells to demonstrate the importance of endogenous AREL1 in regulating lysosomal positioning and degradative function.

We have taken the advice and generated ZRANB1 knockdown U2OS cells (Extended Data Fig. 6h). The lysosomal phenotypes in ZRANB1 knockdown cells, including ER–lysosome MCSs (Extended

Data Fig. 6i,j), lysosomal distribution (Fig. 4j,k), lysosomal pH (Fig. 4l,m), and lysosomal degradation (Fig. 4n,o), were all opposite to those in *AREL1* KO cells.

These results corroborate the crucial roles of endogenous *AREL1* and *ZRANB1* in controlling lysosomal positioning and degradative function.

b. *AREL1* forming a K33 chain on ATP6V1B2 was confirmed by over-expressing all three components (*AREL1*, Ub, and substrate). Accumulating evidence suggests that over-expression of an E3 and Ub can promiscuously ubiquitylate the proteins in proximity. So, it would be great to show this with endogenous proteins.

Response: We fully agree with the reviewer. We have taken the advice and immunoprecipitated endogenous *V₁B2* protein from WT and *AREL1* KO U2OS cells. As shown in Fig. 4b, the ubiquitylation of endogenous *V₁B2* protein was nearly eliminated in *AREL1* KO U2OS cells.

c. Related to *AREL1* over-expression ubiquitylation: The authors show the UB pattern only on over-expressed ATPase subunits. Did the authors look at other endogenous substrates unbiasedly (ex., total K33 blot in the lysate)? How does the K33 WB pattern change when the mutant *AREL1* is over-expressed? This may inform readers of a more unbiased phenotype of *AREL1* overexpression or KO.

Response: Thank you for the helpful suggestion.

We observed similar endogenous ubiquitin levels in the lysates of WT and *AREL1* KO cells (Fig. 4b). The levels of total ubiquitylated proteins also remained similar in the cells overexpressing WT and mutant *AREL1* (Fig. 4a and Extended Data Fig. 5).

These results suggest that *AREL1* selectively catalyzes *V₁B2* ubiquitylation.

d. Fig. 4B. ATP6V1B2-Flag showing a ubiquitin smear perhaps suggests that overexpressing a subunit of the multiprotein complex caused the excess ATP6V1B2 subunit to be constitutively targeted to the proteasome. Could the authors comment on this? How does the endogenous ATP6V1B2 blot look? It is unclear how much of the K33 blots are on the ATP6V1B2 stably forming complex with other components and how much of it is artifacts of monomeric components not correctly located to the lysosomal surface.

Response: Thank you for the insightful comments. We agree that overexpression of a subunit of the multiprotein complex might cause non-specific ubiquitylation of the overexpressed protein for proteasomal degradation.

Therefore, we evaluated the ubiquitylation of endogenous *V₁B2* in WT and *AREL1* KO U2OS cells and found nearly eliminated ubiquitylation of endogenous *V₁B2* protein in *AREL1* KO cells (Fig. 4b).

We further immunoprecipitated endogenous *V₁B2* protein from membrane and cytosolic fractions, respectively, and found only membrane-associated *V₁B2* to be ubiquitylated (Fig. 4c). These results suggest that the *V₁B2* subunit in intact V-ATPase complex, rather than the unassembled cytosolic *V₁B2*, is ubiquitylated by *AREL1*.

These multiple lines of evidence demonstrate that *V₁B2* is selectively ubiquitylated by *AREL1*.

e. Fig. 4B. The authors should cite the previous papers showing the K33 linkage specificity of *AREL1* (PMID: 25752577, 25723849).

Response: These two papers have been cited as Refs 41 and 42 in the revised manuscript.

f. ZRANB1 deubiquitylating the K33 chains: this was done with only over-expressing conditions without blots showing the over-expression level compared to the endogenous ZRANB1 level. DUBs, especially when over-expressed, can hydrolyze any ubiquitin chains in proximity. So, it is unclear with the current data whether the ZRANB1 is relevant to ATP6V1B2 K33 chain trimming in the endogenous context.

Response: We thank the reviewer for the helpful advice.

To demonstrate that ZRANB1 could trim ubiquitin chains from V₁B2 protein in the endogenous context, we immunoprecipitated endogenous V₁B2 protein from WT and *ZRANB1* knockdown U2OS cells (Extended Data Fig. 6h), and found markedly increased ubiquitylation of endogenous V₁B2 when ZRANB1 was depleted (Fig. 4i). Meanwhile, the ubiquitylation of total proteins was unaffected by ZRANB1 knockdown (Fig. 4i). These results indicate that ZRANB1 de-ubiquitylates endogenous V₁B2 protein.

5. The authors show that lysosomes in the *AREL1*^{-/-} cells are less acidic than those in WT cells. Accordingly, the DQ-BSA assay, artificial cargo turnover, shows reduced lysosomal hydrolase activity in the KO cells. This suggests that the defective lysosomal processing may be prolonged in the KO cells. Do the authors observe any defects in endogenous protein, lipids, nucleic acids, or sugar turnover in the KO cells in a steady state or during stress, such as starvation? This could potentially link the authors' cell biological findings to the mouse KO phenotype.

Response: We thank the reviewer for the helpful advice and have performed the following experiments to investigate whether *AREL1* deficiency induces the accumulation of proteins and lipids in cultured cells.

We first subjected WT and *AREL1* KO cells to amino acid starvation and examined the protein levels of lipidated microtubule-associated protein A light chain 3B (LC3B) as previously described (PMID: 31692446). As shown in Figure 2-for reviewer, LC3B protein levels appeared slightly higher in *AREL1* KO cells at later time points following amino acid starvation, suggestive of compromised protein degradation by lysosomes in the absence of *AREL1*.

Figure 2-for reviewer. Starvation-induced LC3B degradation was slightly delayed in the absence of *AREL1*. A, U2OS cells were starved by incubating with DMEM without amino acids and serum for indicated periods. Cells were harvested, lysed and subjected for immunoblotting analysis. B, Quantification of relative levels of LC3B normalized to Tubulin. Data are presented as mean \pm SD (n = 3 independent experiments). Student's t-test.

We also tried to stain misfolded and aggregated proteins using the Proteostat dye (PMIDs: 29590078, 39713930). WT cells treated with bafilomycin A1 were used as a positive control. No Proteostat staining was detected in lysosomes of both WT and *AREL1* KO cells grown under normal conditions (Figure 3-for reviewer).

Figure 3-for reviewer. Representative confocal images showing U2OS cells immunostained with mouse anti-LAMP1 antibody and then incubated with Proteostat detection reagent. Cell contour and nucleus are outlined using white dashed lines. For bafilomycin A1 treatment, WT U2OS cells were incubated with 0.1 μ M bafilomycin A1 for 12 h and then fixed with 4% paraformaldehyde. Scale bars, 10 μ m.

To investigate whether loss of *AREL1* has any effects on lysosomal degradation of lipids, we stained neutral lipids with BODIPY and lysosomes with LysoTracker as described in a recently published paper (PMID: 39847635). The BODIPY puncta in lysosomes remained similar in WT and *AREL1* KO cells (Figure 4-for reviewer). We also failed to detect cholesterol accumulation, which can be induced by U18666A that blocks lysosomal cholesterol export (PMID: 26646182), in *AREL1* KO cells (Figure 5-for reviewer).

Figure 4-for review. Representative confocal images showing U2OS cells stained with 5 μ M BODIPY and 100 nM lysotracker at 37 $^{\circ}$ C for 30 min and then fixed with 4% paraformaldehyde. Cell contour and nucleus are outlined using white dashed lines. Scale bars, 10 μ m. B, Pearson's correlation coefficient (R) for BODIPY and lysotracker in A. $n = 24$ cells for each condition. Data are presented as mean \pm SD ($n = 3$ independent experiments). Mann-Whitney U-test.

Figure 5-for review. Representative confocal images showing U2OS cells stained with 50 μ g/ml Filipin. For U18666A treatment, live cells were treated with 5 μ M U18666A for 12 h. Cell contour and nucleus are outlined using white dashed lines. Scale bars, 10 μ m.

In summary, we found slightly attenuated degradation in LC3 without accumulation of misfolded proteins, neutral lipids or cholesterol in *AREL1* KO cells. This can be attributed to rapid proliferation of U2OS cells and thus turnover of lysosomes. However, in postmitotic Purkinje cells of 12-month-old *Arel1*^{-/-} mice, lysosomes were significantly enlarged, and more and generally larger lipofuscin granules were frequently detected (Fig. 6k-r and Extended Data Fig. 9f). The accumulation of lipofuscin was less severe in 6-month-old *Arel1*^{-/-} mice than that in 12-month-old *Arel1*^{-/-} mice (compare Fig 6k,l with Extended Data Fig. 9k). Therefore, it seems to take a long time for proteins and/or lipids to accumulate

in lysosomes of *Arel1*^{-/-} Purkinje cells. These results support that late-onset Purkinje cell loss and ataxia in mid-aged *Arel1*^{-/-} mice.

Minor comments

1. Figure 1K: it would be good if the authors could include representative images. Additionally, the main body of the paper and panel K categorize the lysosomes that are +Lamp1-mCherry and +Dylight403-Streptavidin positive as being in contact with the ER, but there is no evidence that these lysosomes are currently in contact with the ER at the time of imaging, as the lysosome can have been labeled and then trafficked elsewhere over the 30 min biotin treatment.

Response: We thank the reviewer for the insightful comments.

To avoid biotin-related artifacts and explicitly label ER–lysosome MCSs, we turned to use the proximity ligation assay involving the anti-LAMP1 and anti-calnexin primary antibodies in the revised manuscript. The representative images and quantification results have been provided (Fig. 1k,l). These new results support our conclusion that AREL1 mediates the formation of ER–lysosome MCSs.

2. Figure 1M: where there is knockdown of a protein that is not being detected in the experiment itself, it would be good to show verification of protein knockdown.

Response: The knockdown efficiency of ATP6V0A3 was examined using real-time quantitative PCR (Extended Data Fig. 3j) and immunoblotting (Extended Data Fig. 3k).

3. Similarly, the manuscript lacks information/confirmation on AREL1 KO in U2OS and HeLa cells (method, blots, gRNA sequence, etc.).

Response: The knockout of *AREL1* in U2OS and HeLa cells was verified using genomic sequencing (Extended Data Fig. 1h and Extended Data Fig. 4e) and immunoblotting (Extended Data Fig. 1i and Extended Data Fig. 4f). The methods, including gRNA sequences, of generating *AREL1* KO cells have been included in the revised manuscript (Pages 23 and 24).

4. Figure 3A, the authors should include statistical analysis for the degree of difference between the two different populations.

Response: We have performed the multivariate analysis of variance (MANOVA) to examine the degree of difference between lysosomes from WT and *AREL1* U2OS KO cells. Fig. 3a has been updated and MANOVA analysis was included the revised method (Page 35).

5. Figure 4: Can the authors explain why there is still a faint ubiquitin smear in the HA immunoblot in the negative control conditions? This is particularly unusual in panel C, where we still see a faint smear in even the K29-only lanes.

Response: We have re-performed all over-expressive ubiquitylation experiment. Each experiment was repeated at least twice and representative data are shown.

As shown in Extended Data Fig. 5b and c, ubiquitin signals were barely detectable when AREL1 was not expressed, or when K29 ubiquitin was expressed regardless of the presence of AREL1. By contrast, K33R ubiquitin mutant was the only one out of seven K-to-R mutants that failed to confer

AREL1-mediated ubiquitylation of V1B2 (Extended Data Fig. 5b). The ubiquitin that only contains the lysine residue at position 33 (K33 only) was sufficient to support AREL1-mediated V1B2 ubiquitylation (Extended Data Fig. 5c). (Pages 9-10)

6. Figure 4 E, the zoom panel is further from the nucleus in the BFP-ZRANB1(WT) condition than in the other conditions, so the image appears somewhat misleading, as there is clearly less signal in the HA-Ubiquitin channel, but that may just be due to the positioning of the zoom panel.

Response: We have taken the advice and presented an area that is much closer to the nucleus of cells expressing BFP-ZRANB1(WT) (Extended Data Fig. 6f). The results show that the expression of WT but not catalytically inactive ZRANB1 markedly reduces ubiquitin-positive signals associated with lysosomes.

7. Figure 4F. The quantification in shown in Panel F is not the same as what is stated in the body of the paper: “expression of WT ZRANB1 markedly reduced ubiquitin-positive signals at the sites where AREL1 and lysosomes were opposed (Fig. 4E-F)”, however, the quantification is only for HA-Ubiquitin – LAMP1 correlation.

Response: We have taken the advice and rephrased the sentence to “Overexpression of WT ZRANB1 but not catalytically inactive C443S mutant markedly reduced ubiquitin-positive signals associated with lysosomes (Extended Data Fig. 6f,g)”. (Page 10)

Reviewer #2 (Remarks to the Author):

This manuscript suggests a new mechanism by which the ER regulates the perinuclear localization of lysosomes. The authors report experiments showing that an ER-resident E3 ligase AREL1 initiates contacts with lysosomes by interacting with the V0a subunit of V-ATPase and catalyses the K33-linked polyubiquitylation of V1B2, which then binds to UBAC2 present in the perinuclear ER. Depletion of AREL1 or UBAC2 resulted in more and less acidic lysosomes being distributed to the cell periphery. Overexpression of the K33 deubiquitylating enzyme ZRANB1 counteracted B1 ubiquitylation by AREL1 and also redistributed lysosomes to the cell periphery. The authors also describe experiments showing that AREL1 deficiency causes age-associated neurodegeneration in mice with enlarged lysosomes and increased lipofuscin granules in Purkinje cells.

Whilst there are definitely new findings in this manuscript, they are presented in a way that does not give full credit to what is already known about the role of ubiquitylation in enabling the ER to regulate the perinuclear localization of lysosomes and ignores some relevant aspects of the regulation of V-ATPase. Moreover, there are a number of concerns about the experimental data that the authors must address. These issues are addressed in the major and minor points below:

Response: We thank the reviewer for acknowledging the new findings of our work.

We fully respect previous studies on lysosomes positioning, particularly a series of work by Drs. Neefjes and Berlin revealing how ER-resident UBE2J1/RNF26 ubiquitylation complex acts cooperatively with USP15 (and probably USP17 according to PMID: 35080333) to define the organization of the entire endolysosomal system. That reporting RNF26 mediating the perinuclear anchoring of lysosomes was cited as original Ref.16, and the work reporting that RNF26 binds

perinuclear vimentin filaments was cited as original Ref.36. The previous study showing that perinuclear lysosomes are more acidic than the peripheral ones was cited as original Ref.17. We also mentioned in the Introduction that “The association of these two domains and thus V-ATPase activity are reversibly regulated by nutrient status and growth factor signaling (original Ref. 2)”.

We are very willing to take the reviewer’s advice and present our findings in the context of previous studies using a moderate tone as well as cite more relevant studies in the field. We have re-written the Abstract and Introduction as well as substantially revised the manuscript and given full credits to previous studies as we know.

Major Points

1. P2 L50. It is an over-statement to say that mechanisms for lysosomes to cluster in the perinuclear region and the physiological consequences are largely unknown. Although reference 16 is cited here and reference 36 later in the manuscript, the authors largely ignore the extensive work showing the role of the ER-embedded UBE2J1/RNF26 ubiquitylation complex in the perinuclear organization of the endolysosomal system (doi: 10.1016/j.celrep.2020.10865) or the complexity of motor-driven organelle transport to the perinuclear region and periphery (doi: 10.1016/j.celrep.2020.10865; doi: 10.1242/jcs.259689; doi: 10.15252/embj.2019102301). The authors do not discuss their findings in the context of this literature. It is difficult to be certain that the physiological consequences of perinuclear localisation are largely unknown. We do know that the decreased pH is important for the function of many lysosomal enzymes as discussed in reference 17, lysosomal positioning is coordinated with cellular nutrient responses (reference 9) and integration of ER and endolysosomal responses to proteotoxic stress is related to perinuclear localisation (reference 36).

Response: We have taken the reviewer’s advices and substantially revised the manuscript including re-writing the Abstract and Introduction.

We admire previous seminal studies on lysosomal positioning. As the reviewer points out, endolysosomal transport and positioning are modulated by a complex array of factors including motors and the associated proteins as well as interorganelle contacts (PMIDs: 28231489, 36382597, 32080880), particularly ER–endolysosome membrane contact sites (PMIDs: 19564404, 25855459, 33912962, 31868590, 23389631). The perinuclear ER-resident UBE2J1/RNF26 ubiquitylation complex can act cooperatively with USP15, probably USP17 as well (according to PMID: 35080333), to establish the perinuclear positioning of the entire endolysosomal system (PMIDs: 27368102, 33472082). The sorting nexin SNX19 is another ER protein reported to bind phosphatidylinositol 3-phosphate on the endolysosomal surface and tether them in the perinuclear region (PMID: 34315878).

In the context of these previous excellent studies, we hereby identify a dual-tethering mechanism that involves two ER-embedded proteins (AREL1 and UBAC2) and two V-ATPase subunits (V_{0a} and V_{1B2}). AREL1 interacts with V_{0a} and catalyzes K33-linked ubiquitylation of V_{1B2} . The ubiquitylated V_{1B2} then binds to perinuclear localized UBAC2 to anchor lysosomes in the perinuclear region. Depletion of AREL1 or UBAC2 disperses lysosomes to the cell periphery and impairs lysosomal acidification and degradative capacity. More interestingly, depletion of *AREL1* or *UBAC2* results in more lysosomes without V_{1D} , suggestive of more partially assembled V-ATPases, which can explain for elevated luminal pH of peripheral lysosomes. The deubiquitylating enzyme ZRANB1 by counteracting AREL1-mediated V_{1B2} ubiquitylation releases lysosomes to the cell periphery and impairs lysosomal acidification and degradative function. These findings provide new mechanistic explanations for the earlier observations that perinuclear lysosomes are more acidic and less mobile compared with the peripheral ones (PMID: 26975849).

Regarding the physiological consequences of proper lysosomal positioning, it has been shown, in cultured cells, that lysosomes can change their intracellular positioning in response to nutrient availability, with the perinuclear ones responsible for substrate degradation (PMIDs: 21394080, 38096602). The RNF26-mediated perinuclear localization of lysosomes has been shown to promote lysosomal trafficking of activated EGFR and termination of EGF-induced AKT signaling (PMID: 33472082), as well as ER reorganization in response to proteotoxic stress (PMID: 37519262).

In our study, we take a step further and evaluate the functional significance of AREL1-mediated lysosomal positioning at the animal level. *Are11^{-/-}* mice have more peripherally localized lysosomes and, strikingly, increased numbers and sizes of lipofuscin granules — suggestive of lysosomal dysfunction and senescence— in Purkinje cells (Fig. 6k-r and Extended Data Fig. 9f). *Are11^{-/-}* mice display age-dependent loss of Purkinje cells (Fig. 6i-l), where *ARE1L1* is highly expressed (Fig. 6h and Extended Data Fig. 9c,d), and progressive ataxic phenotype (Fig. 6a-f and Supplementary Movies 1-4). Given the important functions of AREL1 in cultured cells and mice, we conclude AREL1 is a new type of ER tether regulating lysosomal positioning and degradative capacity in addition to previous identified ones.

We have taken the advice and removed the overstated sentence. The above discussions have been included in the revised manuscript (Pages 1, 2,3,14,15 and 16), and the relevant papers have been cited (such as Refs 1,2,4,8,9,15 and 31 as mentioned by the reviewer).

2. P3 L69-71, FigS1E and P35 Fig legend S1E L995. The top hits from the LAMP1/Sec61 split-Turbo ID assay were assessed by silencing the hits to reduce the % of lysosomes in contact with the ER. The data are presented as mean±SEM with n= 3 cells per group. Were the 3 points shown for each depletion from 3 separate independent experiments (with 3 cells in each)? If not the statistical analysis is highly questionable see eg Nature. 2012 Dec 13;492(7428):180-1. doi: 10.1038/492180a. In view of previously published reports, it would have been interesting to see if depletion of RNF26 also reduced the % of lysosomes in contact with the ER in the U2OS cells. In Fig S1E protein depletion is achieved by shRNA treatment – where are the data showing the extent of depletion?

Response: We thank the reviewer for the helpful advice.

We have re-performed the entire experiment using PLA as a measure for ER–lysosome contacts. The effects of all 18 candidates in Extended Data Fig. 1d plus RNF26 on ER–lysosome MCSs were evaluated using shRNA-mediated knockdown (Extended Data Fig. 1e) followed by PLA in U2OS cells (Extended Data Fig. 1f,g). The experiments were repeated three times and quantified. The exact numbers of cells and repeats were all clearly stated in the relevant figure legends. Based on the suggestions, we depleted *RNF26* using shRNA and found a 24% reduction in PLA puncta in U2OS cells (Extended Data Fig. 1f,g). The knockdown efficiency of each candidate gene and RNF26 was determined using real-time quantitative PCR (Extended Data Fig. 1e).

3. P3 L109. The authors claim that it is the hinge region of AREL1 (G706TGDIS) that interacts with the amino terminal domain of the V0a subunits. The evidence is from co-immunoprecipitation of expressed tagged proteins (Fig 1L and S3 E-F) and, in the case of the hinge domain, colocalization of delta hinge with LAMP1 (Fig 1 E,F). Tagged AREL1 constructs that do not contain the hinge region also fail to colocalize with LAMP 1 (Fig S2 B,C). The delta hinge construct is not shown schematically in FigS2B but assuming it is just a full length AREL1 missing 6 amino acids G706TGDIS, the experiments shown raise the intriguing question of how this hinge binds to the N-terminal domain of

V0a subunits. Do the authors have any data either from mutagenesis of the hinge or biophysical experiments with expressed protein fragments that address the nature and affinity of the binding. Have they used alphafold simulations to predict any favoured binding sites for the hinge in the aminoterminal domain of V0a, which could inform point mutation experiments? Any of these approaches would greatly strengthen the evidence that it is this small hinge that accounts for the binding of AREL1 to V0a.

Response: We thank the reviewer for the helpful advice.

The scheme of construct lacking the hinge region has been included in Extended Data Fig. 2c.

To determine the interaction mode between the N-terminal domain of V₀a subunits and the hinge region of AREL1, we first used AlphaFold 3 to predict the interaction between the two. The N-terminal domains of all four V₀a isoforms were predicted to interact with the hinge region in a similar manner (Extended Data Fig. 3h). We further utilized alanine-scanning mutagenesis to determine the exact amino acid(s) mediating the interaction between the hinge region and the V₀a subunit. The V₀a3 isoform (encoded by *ATP6V0A3*) was chosen as a representative because it was highly expressed in U2OS cells (Extended Data Fig. 3i), and knockdown of *ATP6V0A3* (Extended Data Fig. 3j,k) almost completely disrupted AREL1 association with lysosomes (Fig. 1m,n). The co-immunoprecipitation of AREL1 by V₀a3 was nearly completely eliminated by the D709A mutation and profoundly reduced by the I710A mutation, with an even greater effect observed for the double-site mutant (Fig. 1o). The AlphaFold 3 modelling predicts that the D709 residue of the hinge region forms a hydrogen bond with the H297 residue of V₀a3, and that the I710 residue inserts into a hydrophobic groove (Fig. 1p). Consistently, single and double mutations of the H297 and A305 residues of V₀a3 markedly abrogated its interaction with AREL1 (Fig. 1q). These new results further support that the hinge region is responsible for the binding of AREL1 to V₀a3.

4. P3 L79 , P5 L123 and P6 L166. How do the authors account for the AREL1 C790A (i.e. ubiquitin ligase inactive) mutant failing to dissociate AREL1 from lysosomes with the same mutant not rescuing the scattering of lysosomes caused by AREL1 depletion and failing to anchor lysosomes perinuclearly, given that AREL1 is an integral membrane protein of the ER? It is not clear that the data showing the interaction of ubiquitylated V1B2 with UBAC2 are sufficient to resolve this conundrum if AREL1 ubiquitin ligase activity is necessary to tether lysosomes in the perinuclear region.

Response: We thank the reviewer for the insightful comments.

The findings that AREL1 C790A mutant despite interacting with lysosomes (Fig. 1e,f) fails to anchor lysosomes perinuclearly (Fig. 2d,e) suggest that AREL1–V₀a interaction alone does not, at least not sufficient to, mediate perinuclear localization of lysosomes. Rather, the interaction between the ubiquitylated V₁B2, which is conferred by AREL1, and UBAC2 is required for constraining lysosomes in the perinuclear region.

In support of this notion, catalytically inactive AREL1(C790A) was unable to revert UBAC2–V₁B2 interaction (Fig. 5c) and restrain lysosomes perinuclearly (Fig. 2d,e) in *AREL1* knockout cells. UBA domain-deleted UBAC2 that failed to interact with V₁B2 (Fig. 5b) was unable to confer lysosomal perinuclear localization as WT UBAC2 did in *UBAC2* knockdown cells (Fig. 5e,f).

Interestingly, deficiency of *AREL1* did not affect the subcellular distribution of UBAC2 (Extended Data Fig. 7h). However, in *UBAC2* knockdown cells where lysosomes were peripherally dispersed (Extended Data Fig. 7d), WT AREL1 and the C790A mutant were redistributed to the cell periphery,

whereas the hinge region-deleted AREL1 still stayed perinuclearly (Extended Data Fig. 7i). Since the hinge region-deleted AREL1 failed to interact with lysosomes (Fig. 1e,f), these results suggest that the peripheral distribution of AREL1 is actually conferred by that of lysosomes in *UBAC2* knockdown cells. The findings that AREL1 as an integral membrane protein can co-travel with lysosomes are not totally unexpected, since lysosomes can actively regulate ER structure and distribution (PMID: 33328230).

These new results have been included in the revised manuscript (Pages 11-12).

5. P4 L104-105. From the volcano plot shown in Fig S3B is not clear that 'the most obviously enriched hits were subunits of the V-ATPase'. Other hits are equally or more enriched but they are not identified and should be.

Response: By using split-TurboID in proximity labeling strategy, we identified a total of 11 late endosome/lysosome proteins, except for Tb-fused LAMP1 as the bait, that were significantly enriched following biotin treatment. Six among them were V-ATPase subunits, with ATP6AP2/PRR, ATP6AP1/Ac45, V₁B2, V₀c, and V₁A being the top 5 proteins. We therefore conclude that V-ATPase subunits are highly enriched.

Many of the hits that seemed to be more enriched than the ATPase subunits were ER proteins, as shown in Extended Data Fig. 3b in green circles. Two Tb-fused proteins are in blue, and 11 enriched late endosome/lysosome proteins are in orange. We have included the proteomics raw results for Extended Data Fig. 3b as Supplementary Table 2.

6. P6 L169-210 and P10 L262 & L269. The evidence presented that AREL1 ubiquitinates V1B2, which can be reversed by the deubiquitylating enzyme ZRANB1 is compelling as are the data showing redistribution of lysosomes to the periphery by ZRANB1 and *UBAC2* as well as the co-immunoprecipitation of tagged V1B2 with *UBAC*. However, it is not at all clear how the proposed dual-tethering mechanism and the AREL1-*UBAC1*-ZRANB1 axis are proposed to work mechanistically. A major issue is the well described reversible assembly and disassembly of V-ATPase that occurs in mammalian cells. Evidence from FRAP experiments has shown that the V1 subunits of the V-ATPase on pre-synaptic vesicles or endolysosomal compartments are rapidly exchanging and in dynamic equilibrium with a cytosolic pool with FRAP curves suggesting t_{1/2} of ~10-50s (doi: 10.1016/j.celrep.2017.07.040; doi: 10.1091/mbc.E23-08-0322). If this is occurring on the perinuclear lysosomes, it is difficult to see how the ubiquitylated V1B2 interaction with *UBAC2* can function to keep the lysosomes at the perinuclear location in the model as presented here.

Response: We thank the reviewer for the insightful comments.

The dynamics of V-ATPase assembly presented in these two papers (PMIDs: 28793259, 38446621) are interesting but not necessarily discordant with our model. We show that ER-resident AREL1 can interact with the V₀a subunit of V-ATPase and ubiquitylate the V₁B2 subunit. The ubiquitylated V₁B2 then interacts with *UBAC2* and anchors lysosomes in the perinuclear ER. ZRANB1 antagonizes AREL1-mediated ubiquitylation of V₁B2 and disperses lysosomes to the cell periphery. These results suggest that V₁B2 ubiquitylation/deubiquitylation is a key step in regulating lysosomal positioning.

To reconcile our model with the rapid assembly/disassembly kinetics of V-ATPase, we performed fluorescence recovery after photobleaching (FRAP) experiments as described in the above two papers. V₁E1 tagged with mCherry exhibited a robust punctate staining pattern that was colocalized with LAMP1-EGFP (Extended Data Fig. 8a), and was chosen as a measure for V₁ subunit recruitment to

the relatively immobile perinuclear lysosomes. The perinuclear regions of WT and *ZRANB1* knockdown cells were photobleached and the recovery of fluorescence was measured. The half time of fluorescence recovery ($\tau_{1/2}$) was 24.46 s in WT cells and 44.06 s in *ZRANB1* knockdown cells (Extended Data Fig. 8b,c), indicating a delayed exchange of V₁E1 between the cytosolic pool and that bound to the lysosomal surface when *ZRANB1* was depleted. These results suggest that increased ubiquitylation can help stabilize the V-ATPase holoenzyme and facilitate UBAC2-mediated perinuclear localization of lysosomes. It should also be emphasized that the recovery time in our FRAP experiments was similar to that reported by the abovementioned previous studies, suggesting that V-ATPase assembly is indeed rapid but still subjected to regulation by ubiquitylation/deubiquitylation.

The above descriptions have been included in the revised manuscript as well (Page 12).

7. P8 L211 onwards. The data on AREL1 deficient mice are interesting but rather preliminary although they do show that in Purkinje cells of the KO mice there are larger lysosomes and increased lipofuscin granules. The relationship of these data to the rest of the paper is not entirely clear.

Response: In the cell biology part of our work, we have demonstrated that AREL1 is a critical regulator of lysosomal perinuclear positioning and that AREL1 deficiency compromises degradative function of lysosomes. Given that AREL1 is expressed highly in the mouse brain including the cerebellum (Fig. 6g,h and Extended Data Fig. 9b-d), and that lysosomal dysfunction is closely implicated in neurodegenerative diseases, we therefore generated *Arel1* KO mice to evaluate whether AREL1 deficiency could cause neurological deficits, which, in our opinion, is an essential step to validate the physiological importance of AREL1.

In the current version, we have included many more mice, including 12-month-old females, for the gait and rotarod analysis as suggested by Reviewer#4. In summary, our results show that *Arel1*^{-/-} mice develop late-onset ataxia (Fig. 6a-f and Supplementary Movies 1-4), Purkinje cell loss (Fig. 6i,j) and lipofuscin accumulation (Fig. 6k-r). Notably, 6-month-old *Arel1*^{-/-} mice showing no Purkinje cell loss or ataxic phenotype, but still have lipofuscin accumulation in Purkinje cells (Extended Data Fig. 9g-k), suggesting that lysosomal dysfunction might drive late-onset Purkinje cell loss and ataxia.

These results underscore the importance of ER-regulated perinuclear lysosomal positioning in neuronal health and disease.

8. P9 L242 onwards. The section on the V50M mutant of AREL1 is very preliminary. The authors state the V50M mutant failed to rescue peripheral scattering and speculate that is due to reduced protein expression in a system in which the mutant is over-expressed in a cultured cell. Without analysis using cells with the same levels of expression of WT and mutant protein it is impossible to make any conclusions.

Response: We thank the reviewer for the helpful advice. As suggested by Reviewer#4, we have removed the results on V50M mutant, and performed many new analyses on cultured cells and mice to demonstrate that AREL1 is indeed critical for maintaining lysosomal function and Purkinje cell survival.

9. In many of the figure legends it is not clear that statistical analysis was performed on data from 3 or more independent experiments, because often n is given as number of cells or numbers of lysosomes. As mentioned in point 2 above, statistics should really be performed on data from several independent experiments.

Response: We have taken the advice and clearly stated the exact numbers of lysosomes, cells and repeats in the revised figure legends. We have also re-performed the experiments and quantified sufficient numbers of cells and lysosomes using previous studies (PMIDs: 27368102, 34315878, 35314674, 33472082, 37519262) as the guidance.

10. Title. Given that it is already known that localisation of lysosomes governs lysosomal acidity and that ER-lysosome interaction including a role for ubiquitylation plays a role in perinuclear localisation, the proposed title is over-stated and should be revised to relate more closely to the new findings in this manuscript.

Response: Based on the advice, we have modified the title to “E3 ubiquitin ligase AREL1 controls perinuclear localization of lysosomes and supports Purkinje cell survival”.

Minor Points

1. P2 L40. In addition to the review in reference 2, the authors should refer to more recent work on regulation of V-ATPase assembly/disassembly by nutrient status, in particular doi: 10.1038/s41467-022-32515-6.

Response: We have taken the advice and included more recent work on V-ATPase assembly/disassembly (Refs. 19-24). The indicated paper (doi: 10.1038/s41467-022-32515-6) has been cited as Ref. 23.

2. P3 L63-69. Was the split-TurboID screen to give the hits listed in Fig S1D carried out in U2OS cells as implied from the legends to other parts of Fig S1 or in HEK293T cells as stated in the Methods section L465?

Response: The split-TurboID screen coupled to mass spectrometry to identify proteins at ER-lysosome MCSs was performed in HEK293T cells owing to relatively high transfection efficiency.

The effects of 18 candidates plus RNF26 on ER-lysosome MCSs were evaluated using shRNA-mediated knockdown (Extended Data Fig. 1d,e) followed by PLA in U2OS cells (Extended Data Fig. 1f,g). Silencing of *AREL1* and several other genes (*ATP6AP1*, *ATP6AP2*, *VAPA*, *VMA21*, *UBE2J1*, *MOSPD2*, *ZFYVE27*, and *TMEM9*) significantly reduced the percentage of lysosomes in contacts with the ER in U2OS cells. In *AREL1* KO U2OS cells (Extended Data Fig. 1h,i) and *AREL1* knockdown HEK293T cells (Extended Data Fig. 1m), PLA puncta were also significantly reduced (Extended Data Fig. 1k,l,n,o). These results support that *AREL1* is required for ER-lysosome MCSs.

We have clearly stated cell lines for each experiment in Extended Data Fig. 1 and throughout the manuscript as well.

3. L75. The data from the centrifugation assay were consistent with ER localization of *AREL1* but cannot be said to confirm ER localization.

Response: We have rephrased the sentence to “It was mostly present in the ER-enriched membrane fractions in density gradient centrifugation (Extended Data Fig. 2a)”. (Page 5)

4. P3 L76 and Fig S2C. Pearson correlation coefficients should be given for colocalization of all the AREL1 mutants with LAMP1 as they were for others in Fig1F

Response: We have included Pearson correlation coefficients as a measure for the colocalization between AREL1 mutants and LAMP1 (Extended Data Fig. 2d,e).

5. P5 L142. The pH of lysosomes is indeed maintained by V-ATPase but as is explained later also affected by a pH leak that may be increased in peripheral lysosomes.

Response: We agree with the reviewer that lysosomal pH is determined by the dynamic balance between proton influx through V-ATPase and proton efflux through TMEM175 (PMID: 35750034), SLC7A11 (PMID: 40280132), as well as other transporters or channels.

The sentence has been modified accordingly (Page 8).

6. P6 L132-163. Whilst it was important that the authors showed the increased pH and reduced proteolytic capacity of peripheral lysosomes after AREL1 depletion, they should make it clearer that the data are entirely consistent with previous reports that peripheral lysosomes have increased pH, reduced proteolytic capacity and also reduced amounts of the V1 subunits of V-ATPase (see reference 17 which should also be cited here).

Response: We have taken the advice and included the following sentences in the revised manuscript (Page 9):

“These findings are consistent with the previous reports that peripheral lysosomes have increased pH (PMID: 26975849), reduced proteolytic capacity (PMIDs: 26975849, 21394080), and reduced amounts of the V₁ subunits of V-ATPase (PMID: 34435379), underscoring an important role of AREL1 in regulating lysosomal perinuclear positioning and degradative capacity.”

Reviewer #3 (Remarks to the Author):

The central claim of Jiang et al. is that the ER-resident E3 ligase AREL1 initiates membrane contacts with lysosomes by interacting with ATP6V0A, catalyzing K33 polyubiquitylation of ATP6V1B2. ATP6V1B2 then binds to UBAC2 in the perinuclear ER, thereby localizing lysosomes to the perinuclear ER. With mutation of AREL1 lysosomes localize to the cell periphery and V1 subunits are absent (and the lysosomes have increased pH). Mice lacking Arel1 exhibit age-dependent Purkinje cell loss and cerebellar ataxia, recapitulating the effects of a pathological loss of function polymorphism. Together, these findings constitute an important advance.

The split turboID system for biotinylated and detecting proteins at lysosome-ER contact sites is quite ingenious. The results of this experiment and subsequent confirmation show that AREL1 is responsible for localization of lysosomes to the perinuclear ER.

I do not understand why only ATP6AP1 and ATP6AP2 from the V-ATPase were identified in the LAMP1-mCherry and EGFP-Sec61 β split turboID experiment. Wouldn't one expect that the other subunits of V-ATPase would also be identified? Further, I would expect ATP6AP1 and 2 to be two of

least accessible V-ATPase subunits because they are both small and hydrophobic membrane proteins. These proteins should be almost inaccessible in the structure when V1 is assembled with Vo. Can the authors explain this result (or correct my misunderstanding)?

Response: We thank the reviewer for the insightful comments.

We do not know the exact reason why ATP6AP1 and ATP6AP2 were enriched in the LAMP1-mCherry and EGFP-Sec61 β split-Turbo ID assay. They might be in the right distance to the TurboID enzyme and therefore become biotinylated. In addition, ATP6AP1 and ATP6AP2 have been previously shown to be co-immunoprecipitated with TMEM106B and TMEM9, respectively (PMIDs: 28728022, 30374053). Given that TMEM106B and TMEM9 were highly enriched in the split-Turbo ID assay (Extended Data Fig. 1d and Supplementary Table 1), ATP6AP1 and ATP6AP2 as their binding partners may be detected.

The mass spectrometry analysis provides 18 strong candidates (Extended Data Fig. 1d). We then examined the effects of each of 18 candidates on ER-lysosome MCSs using shRNA-mediated knockdown (Extended Data Fig. 1e) followed by proximity ligation assay (PLA) (Extended Data Fig. 1f,g). AREL1 was chosen for further investigation given its uncharacterized function in maintaining ER-lysosome MCSs.

How can the authors interpret the finding that all four ATP6V0A subunits were identified as interacting with AREL1 when these subunits are expressed in a cell-, tissue-, and organelle-specific manner? For example, mass spectrometry analysis of purified brain, HEK293, and kidney V-ATPase identified almost entirely ATP6V0A1, presumably from the lysosomes of kidneys, not the specialized ATP6V0A4 from intercalated cells. If this IP is the only evidence of a physical association between V-ATPase and AREL1 this discrepancy is worrying and makes me nervous about the authors interpretation of the result. Do the authors expect the 0.5% NP40 to solubilize the membranes in cells fully, or could membranes pellet during IP regardless of interaction with AREL1?

Response: The co-IP experiments (Extended Data Fig. 3d,f,g and Fig. 1o,q) were performed by overexpressing AREL1 together with each indicated subunit of V-ATPase in HEK293T cells. The condition where no V-ATPase subunit but only AREL1 was expressed was included as the negative control (1st lane). The cells were lysed in 0.5% NP40 IP buffer and centrifugated at 12,000 g for 10 min to discard insoluble materials. The supernatant was subjected to co-IP experiments. After incubation with anti-FLAG beads, the sample was centrifugated at 1,000 g for 5 min and washed with 0.5% NP40 IP buffer for 3 times. AREL1 was not detected in the pellets unless V-ATPase subunits were co-expressed (1st lane). These results demonstrate that 0.5% NP40 is sufficient to fully solubilize the membranes.

The AlphaFold 3 predicts that the NTDs of all four Vo_a isoforms interact with the hinge region of AREL1 in a similar manner (Extended Data Fig. 3h). Using the Vo_a3 isoform (encoded by ATP6V0A3) as a representative due to its high expression in U2OS cells (Extended Data Fig. 3i) and its crucial role in AREL1 association with lysosomes (Fig. 1m,n), we mapped out the critical amino acids mediating the interaction between the hinge region of AREL1 and the Vo_a3 using co-IP experiments (Fig. 1o-q).

Therefore, our co-IP experiment results should be reliable.

The decrease in lysosome-associated V1 complex in AREL1 knockout cells appears convincing.

Response: We thank the reviewer for the positive comment.

I believe that AREL1 ubiquitination experiments were done with HEK293 cells. Therefore, it is not surprising that only ubiquitination of ATP6V1B2 was detected (as opposed to B1) because mass spectrometry of V-ATPase purified from HEK cells showed only the B2 isoform. B1 was found in V-ATPase isolated from synaptic vesicles, and B2 from V-ATPase isolated (presumably the lysosomes of) kidney.

Response: We thank the reviewer for advancing our understanding about cell-, tissue-, and organelle-specific composition of V-ATPases.

The ubiquitylation experiments in Fig. 4a,f and Extended Data Fig. 5 were performed using the over-expressive system in HEK293T cells. In these experiments, AREL1, ubiquitin and indicated subunit of V-ATPase were overexpressed, with the condition where AREL1 was not expressed as the negative control. In the over-expressive system, AREL1 was found to selectively ubiquitylate the B2 isoform of the V₁ domain, among the V-ATPase subunits examined, in a catalytic activity-dependent manner (Fig. 4a and Extended Data Fig. 5a).

We have also performed new experiments to examine the ubiquitylation of endogenous V₁B2 protein using U2OS cells, a human osteosarcoma cell line. *ATP6V1B2* was highly expressed in U2OS cells (Extended Data Fig. 6a) and endogenous V₁B2 protein was ubiquitylated by AREL1 (Fig. 4b). In addition, we generated *ZRANB1* knockdown U2OS cells (Extended Data Fig. 6h), and found substantially increased ubiquitylation of endogenous V₁B2 protein compared with control cells (Fig. 4i).

We have clearly stated cell lines for each experiment throughout the manuscript. The above descriptions have been included in the revised manuscript (Pages 9 and 10).

The authors state: “Without AREL1, lysosomes are redistributed to the cell periphery (due to loss of interaction between the B2 subunit of V-ATPase and UBAC2) and become less acidic (as a result of incomplete V-ATPase).” However, this statement seems a bit backward considering that V₁ must be attached to V_o by Rabconnectin-3 for lysosomes to acidify. Couldn’t an alternative interpretation be that in the absence of AREL1, V₁ is not successfully attached or retained on lysosomes and consequently they fail to become as acidic as mature perinuclear lysosomes?

Response: The reviewer is correct and we thank the reviewer for the helpful advice.

We found that, in both WT and *AREL1* KO cells, V₁D was perinuclearly distributed and colocalized with most, if not all, perinuclear lysosomes (Fig. 3d). Lysosomes negative for V₁D were more distant from the nucleus than V₁D-positive ones (Fig. 3e). Depletion of *AREL1* significantly increased the percentage of V₁D-negative lysosomes as well as their distances to the nuclear envelope (Fig. 3e,f). These results suggest that AREL1 is critical for maintaining lysosomes functionally competent in the perinuclear region.

As the reviewer points out, it is possible that AREL1 deficiency impairs the attachment or retention of V₁ domain on lysosomal surface, which can lead to peripherally localized lysosomes with less acidity and compromised degradative capacity. These findings provide new mechanistic explanations for perinuclear lysosomes being more acidic and less mobile compared with the peripheral ones (PMID: 26975849). Since Rabconnectin-3 is essential for the assembly and proper function of V-ATPases (PMIDs: 19758563, 22875945, 38984989, and 34249946), it will be interesting to examine whether Rabconnectin-3 is involved in AREL1-regulated V-ATPase assembly.

We have revised the sentence accordingly (Pages 8 and 15).

Minor

The 1 in V1 and the O should be subscripted. The use of the letter “O” is preferably to the number “0”, as VO is named in analogy with the FO region of the F1FO ATP synthase, where “FO” stands for the oligomycin-binding fraction.

Response: We thank the reviewer for the helpful advice. We have replaced the number “0” with the letter “O” in the revised manuscript. The 1 in V1 and the O have been subscripted as well.

Line 176: “K29/K33-sepecific”

Response: This error has been fixed. (Page 10)

Reviewer #4 (Remarks to the Author):

In this study, Jiang et al. show that an ER-resident E3 ligase AREL1 initiates membrane contacts with lysosomes through interacting with the subunit a of V-ATPase. Disruption of AREL1-mediated ubiquitylation redistributes lysosomes to the cell periphery. They also show that mice lacking Arel1 exhibit age-dependent Purkinje cell loss and cerebellar ataxia.

The study is interesting and novel, however I have some concerns especially regarding the last part of the work. The study on the mouse model appears quite preliminary, some experiments lack appropriate sample size and proper statistics to support the conclusions reported by the authors. Also, the genetic evidence is too weak to support the link of Arel1 to inherited cerebellar ataxia.

These major concerns need to be addressed:

1) In Fig.S7 the authors show that AREL1 is more expressed in the brain as compared to peripheral tissues by qPCR. However, publicly available RNA profiling data sets on normal human tissues (GEO and others e.g. GTEx, Illumina, BioGPS) show that AREL1 is expressed also in other districts, almost ubiquitously, and not specifically in the CNS. Please explain.

Response: We thank the reviewer for the comments on the expression profile of AREL1.

We agree that AREL1 is widely expressed across mouse tissues. However, it is at higher levels in the adult mouse brain according to BioGPS (<http://biogps.org/#goto=genereport&id=68497>), Expression Atlas (<https://www.ebi.ac.uk/gxa/experiments/E-MTAB-2801/Results?geneQuery=%5B%7B%22value%22%3A%22ensmusg00000042350%22%7D%5D&filterFactors=%7B%22STRAIN%22%3A%5B%22C57BL%2F6%22%5D%7D>), TissueEnrich (<https://tissueenrich.gdcb.iastate.edu/>), and JensenLab (https://tissues.jensenlab.org/Entity?figures=tissues_body_%&knowledge=10&experiments=10&textmining=10&homologs=10&type1=10090&type2=-25&id1=ENSMUSP00000048780). According to the Single Cell resource of the Human Protein Atlas (<https://www.proteinatlas.org/ENSG00000119682-AREL1/single+cell>), AREL1 is abundant in neurons and oligodendrocytes in the human brain.

The above databases and our own qPCR results (Extended Data Fig. 9b) support the high expression of AREL1 in adult mouse brain. We have modified the sentence to “AREL1 is widely expressed across

mouse tissues, with a relatively higher level in the adult brain of mice and humans” to avoid any confusion. (Page 13).

Also, it is not clear why the authors decide to concentrate on the cerebellum only. Why the in situ hybridization was performed on cerebellum only? Why not in other brain areas? Please address this issue and modify the text accordingly.

Response: We chose to examine cerebellum because *Arel1*^{-/-} mice at 12 months of age started to lose balance (Fig. 6a, and Supplementary Movies 1,2) and showed a clear hindlimb clasping phenotype when lifted up by the tail (Fig. 6b,c and Supplementary Movies 3,4). Further analysis revealed that 12-month-old *Arel1*^{-/-} male and female mice displayed aberrant gait patterns in footprint analysis (Fig. 6d,e) and spent significantly less time on the rotarod (Fig. 6f). These results suggest that *Arel1* deficiency causes late-onset ataxia in mice. This propelled us to focus on the cerebellum.

Using *Arel1*^{-/-} mice as negative controls for the antibody, we found the cerebellum was one of the brain regions expressing high levels of the AREL1 protein (Fig. 6g). Further examination of cerebellar sections by immunohistochemistry showed that AREL1 was concentrated in Purkinje cells (Fig. 6h). In support of our findings, the in situ hybridization data retrieved from Allen Mouse Brain Atlas (<https://mouse.brain-map.org/experiment/show/69352914>) also reveal the high expression of *Arel1* that corresponds to calbindin-positive Purkinje cells (Extended Data Fig. 9c). In fact, by analyzing the previously published single-nucleus transcriptomics of mouse cerebellar cortex (PMID: 34616064), we found that the *Arel1*, *Zranb1*, *V-ATPase*, *Ubc2* were highly expressed in Purkinje neurons (Extended Data Fig. 9d).

We found no apparent abnormalities in the cerebral cortex or hippocampus of 12-month-old *Arel1*^{-/-} male mice (Extended Data Fig. 9e). However, *Arel1*^{-/-} cerebellum had significantly less Purkinje cells compared with WT controls (Fig. 6i,j). Much more lipofuscin-positive puncta were detected in the residual Purkinje cells of *Arel1*^{-/-} mice (Fig. 6k-r).

These results all support the important role of AREL1 in the cerebellum, particularly Purkinje cells. The above descriptions have been included in the revised manuscript as well (Pages 13 and 14).

2) Related to the previous point: does AREL1 deficiency affect brain areas other than the cerebellum? Could the author provide histology studies made in other areas of the brain? Please address this issue.

Response: As explained above, we performed Nissl staining of brain sections from 12-month-old male mice and found no apparent abnormalities of the cerebral cortex or hippocampus (Extended Data Fig. 9e).

3) Some behavioural studies are performed on very few mice and lack appropriate statistics. Did the authors include both females and males in the study? This is not reported in the methods or in the legends. Please specify. Indeed, at 12 months C57/BL6 males and females generally show size differences that could influence the tests.

Response: We appreciate the helpful suggestions. We have taken the advice and re-performed the footprint analysis and rotarod analysis using many more mice including female mice as well.

Specifically, we used, for each genotype, 12 male mice and 10 female mice at 12 months of age for the footprint analysis (Fig. 6d,e) and the rotarod test (Fig. 6f). Both male and female *Arel1*^{-/-} mice

displayed aberrant gait patterns in footprint analysis (Fig. 6d,e) and spent significantly less time on the rotarod (Fig. 6f). These results suggest that *Arel1* deficiency causes late-onset ataxia in mice.

To investigate whether ataxia, Purkinje cell loss and lipofuscin accumulation seen in mid-aged *Arel1*^{-/-} mice occurred concomitantly or sequentially, we subjected 6-month-old male mice for the footprint analysis (Extended Data Fig. 9g,h) and the rotarod test (Extended Data Fig. 9i). For each genotype, 12 male mice were used.

The numbers of mice and statistics used for analyses have been clearly stated in the relevant figure legends. The details of behavioral studies have been included in the revised Methods (Pages 31 and 32).

Why was the rotarod performed on 6 wt and 4 ko only (while the other tests in a higher number of mice), which is an insufficient number of mice for this kind of test? Also, the authors performed a T test wt vs ko for each day of trial. From this I infer that the datasets follow a normal distribution, therefore a repeated measure ANOVA analysis followed by post-hoc comparisons should be performed. Anyway, the authors need to increase the number of animals per group.

Response: As answered above, we have re-performed the rotarod test using both 12-month-old male mice (n=12 for each genotype) and female mice (n=10 for each genotype), as well as 6-month-old male mice (n=12 for each genotype).

The rotarod test was performed as previously described: (PMIDs: 31255487, 20359499, 26363424, 26352471). In brief, the mice were trained 3 days before test, mice were trained at 8 rpm for three times per day with a 5 min resting period between each trial. On day 4, mice were placed on an accelerating rotarod cylinder, and the latency time of the animals was measured. The speed was slowly increased from 4 to 40 rpm within 5 min. A trial ended if the animal fell off the rods. (Page 32)

The results followed a normal distribution and were presented as mean±SD and analyzed using unpaired two-tailed Student's t-test.

It is not clear in the footprint analysis how many times the mice repeated the exercise.

Response: The footprint analysis was performed as previously described (PMID: 31380846). In brief, mice were acclimated to the behavior rooms for 1 h prior to testing. The fore and hind paws were painted with orange and blue non-toxic water-based paints, respectively. Mice were allowed to walk in a straight line in a narrow tunnel (60 cm × 10 cm × 10 cm) on white paper, with a darkened cage used as the bait at the end of the tunnel. Mice were repeatedly tested for three times. In each test, three hind paw steps from the middle portion were measured for stride length, stride width and stance. (Page 31)

The results followed a normal distribution and were presented as mean±SD and analyzed using unpaired two-tailed Student's t-test.

4) In the last paragraph of the work the authors again concentrate specifically on cerebellar ataxia. However, the authors do not provide sufficient data to support their conclusions. This paragraph should be removed or strongly attenuated, as well as the relative sentence in the Abstract. So far, AREL1 is not linked to any mendelian form of cerebellar ataxia (see OMIM: <https://www.omim.org/entry/615380?search=AREL1&highlight=arel1>), so the the

sentence“AREL1 loss of function mutation is associated with hereditary cerebellar ataxia” is formally not correct under a genetic point of view. A mutation is a pathogenetic variant causing a disease, which is not present or present at very low frequency in the population. The genetic variant rs 140239552 (Alleles:C>T, Chromosome:14:74684549 (GRCh38)) that results in V-to-M substitution is present in the healthy population (Allele count: 7209; Allele frequency : 4.47e-3, with 31 homozygotes found in the general healthy population).

https://gnomad.broadinstitute.org/gene/ENSG00000119682?dataset=gnomad_r4

According to genetic data, this is likely a benign polymorphism, despite the functional data that the author show. Moreover, they do not show any pedigree in which this variant co-segregates with cerebellar ataxia, supporting its inheritance in this kind of disease.

Response: We thank the reviewer for sharing the valuable opinions on human genetics analysis. We would like to take the advice and remove this part. Since we have substantially improved our mechanistic study and mouse work, we hope the reviewer can agree with us that AREL1 plays an important role in regulating perinuclear lysosomal positioning, function and Purkinje cell survival.

Dear Jie,

Thank you for submitting a revised version of your manuscript. We have now received input from three of the original reviewers, who indicate some remaining concerns that can be addressed via mainly textual changes, including toning down of the statements on the specificity to only Purkinje cells, as well as a refinement for the description and terminology of the reported behavioural defects.

There are also a few editorial points that need to be addressed before I can extend official acceptance of the manuscript:

1. Please submit up to five keywords for your manuscript.
2. Please correct the order and headings of the manuscript sections to: Abstract / Keywords / Introduction / Results / Discussion / Methods / Data Availability / Acknowledgements / Disclosure and Competing Interests Statement / References / Figure Legends / Tables / Expanded View Figure Legends.
3. Please submit a complete author checklist, which you can download from our author guidelines (<https://www.embopress.org/pb-assets/embo-site/EMBO%20Press%20Author%20Checklist-1642513524327.xlsx>). Please insert information in the checklist that is also reflected in the manuscript. The completed author checklist will also be part of the Review Process File.
4. At EMBO Press we ask authors to provide source data for the main manuscript figures. You will receive a separate email with instructions for providing source data with your revised manuscript, including how to upload and organize the files.
5. Please rename Extended Data Figures into Expanded View (EV) figures and upload them as individual, production quality figure files.
6. CRediT has replaced the traditional author contributions section because it offers a systematic, machine-readable author contributions format that allows for more effective research assessment. Please remove the Authors Contributions from the manuscript and use the free text boxes beneath each contributing author's name in our online submission system to add specific details on the author's contribution. More information is available in our guide to authors.
7. Please rename "Competing interest declaration" section into "Disclosure and competing interests statement" (further info: <https://www.embopress.org/page/journal/14602075/authorguide#conflictsofinterest>).
8. Since no externally deposited datasets are associated with this study, in the "Data Availability" section please state: "This study includes no data deposited in external repositories". Further information about the format of this section can be found at <https://www.embopress.org/page/journal/14602075/authorguide#dataavailability>
9. Please update references according to The EMBO Journal style - where there are more than 10 authors on a paper, the first 10 should be listed, followed by 'et al.' DOIs should only be used for preprints and datasets that have not been published yet. Please see further information here: <https://www.embopress.org/page/journal/14602075/authorguide#referencesformat>
10. Please rename Supplementary Tables 1-2 into Dataset EV1-2 and update the callouts and legends accordingly. Please remove the legends from the manuscript text file and add to each dataset file in a separate tab/sheet.
11. Please rename the movies into Movie EV1-EV4 and update the callouts accordingly. The legends should be removed from the manuscript text file and zipped with each movie file. Further information is available here: <https://www.embopress.org/page/journal/14602075/authorguide#expandedview>
12. Figure panel 3L is not mentioned in the manuscript text; please add the corresponding callout.
13. Please ensure that the figure are inconsecutive order, currently e.g. Fig 5A,B,C and Fig 6A are called out before Fig 4B and Fig 6B,C before Fig 4C,D.
14. All Materials and Methods need to be described in the main text using our 'Structured Methods' format. According to this format, the Methods section includes a Reagents and Tools Table (listing key reagents, experimental models, software and relevant equipment and including their sources and relevant identifiers) followed by a Methods and Protocols section describing the methods, ideally using a step-by-step protocol format. The aim is to facilitate adoption of the methodologies across labs. Please download and fill our Reagents and Tools Table template (.docx), which you can find in our author guidelines: <https://www.embopress.org/page/journal/14602075/authorguide#structuredmethods>
When submitting your revised manuscript, please do not include the Reagents and Tools Table in the Methods section of the manuscript but upload it as a separate file choosing the file type "Reagent Table".
An example of a Method paper with Structured Methods can be found here: <https://www.embopress.org/doi/10.15252/msb.20178071>.
15. Our data editors have flagged the following issues in figure legends that need correcting:
 - Please provide the exact p values in the legends of figures 1F, N; 2C, E, G; 3C, E, F, J; 4H, K, M, O; 5F, H, J, K, L, M; 6C, E, F, K, L, P, EV1 E, L, O; EDF 2 E, EDF 3 I, J; EDF 4 H, EDF 6A, E, G, J; EDF 7C, E, G.
 - Please indicate the statistical test used for data analysis in the legends of figures EDF 1 E, M; EDF 3 B, I, J; EDF 6A, B.
 - Please define the box plots in terms of minima, maxima, centre, bounds of box and whiskers, and percentile in the legends of figures 3E, 5J, EDF 7I.
 - Please provide information on the number and nature of replicates in the legends of figures EDF3 B, I, J; EDF 6A, B, H; EDF 8C.
 - Please define the error bars in the legends of figures EDF 3 I, J; EDF 6A, B, H; EDF 8C.
 - Please define the scale bar for figures EDF 6F, EDF 9C.

- Please define the dashed lines in the legend of figure 2D.
- Please define the white arrows in the legend of figure EDF 6F.

16. Papers published in The EMBO Journal are accompanied online by a 'Synopsis' to enhance discoverability of the manuscript. It consists of A) a short (1-2 sentences) summary of the findings and their significance, B) 3-4 bullet points highlighting key results (the highlights can be repurposed for this) and C) a synopsis image that is 550x300-600 pixels large (width x height, jpeg or png format). You can either show a model or key data in the synopsis image. Please note that the image size is rather small and that text needs to be readable at the final size.

With kind regards,

Ieva

We realize that it is difficult to revise to a specific deadline. In the interest of protecting the conceptual advance provided by the work, we recommend a revision within 3 months (15th Jan 2026). Please discuss the revision progress ahead of this time with the editor if you require more time to complete the revisions.

Referee #1:

Overall, the authors addressed nearly all concerns raised by this reviewer, and the manuscript's quality has significantly improved. Especially, a tremendous amount of work was invested to address concerns related to over-expression artifacts, which undermined the significance of the original discovery in the prior manuscript. The AREL1 and ZRANB1 knockout (KO) or knockdown (KD) experiments with endogenous protein immunoprecipitation followed by Western blotting (IP-WB) support the previous IP-WB experiments conducted with triple over-expression systems, thereby satisfactorily resolving the major concerns. Methods for quantifying images are clearly described, and overall sample numbers for statistical analysis have significantly increased, further improving the quality and credibility. I only have two minor suggestions.

1. Extended data figure 3b: details on the statistics are not described. Including information on how the p-value is calculated (which statistical test was used) would be helpful. Was the data normalized at all? Endogenously biotinylated proteins such as PC, PCCC, and MCCC could serve as good normalization parameters if needed.

2. AlphaFold3 - including a confidence score would be helpful, as currently there's no information on the probability of the prediction.

Referee #3:

The authors have addressed my concerns convincingly (Reviewer 3) and corrected my misunderstanding of some points about their experimental process.

The authors have added a substantial number of additional experiments in what appears to be a sincere effort to address concerns raised by other reviewers.

From my perspective, the manuscript is suitable for publication in the EMBO Journal.

Referee #4:

In this study, Jiang et al. show that an ER-resident E3 ligase AREL1 initiates membrane contacts with lysosomes by interacting with the subunit α of V-ATPase. Disruption of AREL1-mediated ubiquitylation redistributes lysosomes to the cell periphery. They also show that mice lacking *Arel1* exhibit age-dependent Purkinje cell loss and motor impairment.

The study is interesting, and the authors addressed most of my major concerns regarding the last part of the work. They have improved the evaluation of the motor capabilities of the *Arel1*^{-/-} mouse model and removed all sentences about a potential link between *Arel1* and inherited cerebellar ataxia in humans, for which they had no evidence.

However, I have some doubts that the authors need to address before the manuscript is suitable for publication.

Overall, I still find the logic behind focusing on the cerebellum and in particular on Purkinje cells, and also the statements that *Arel1* deficiency causes specifically cerebellar ataxia, a bit weak.

Here my concerns:

1) In males, the footprint parameters, which relate more to coordination and balance with respect to rotarod, are mildly affected at 12 months. Differences are a bit higher in established ataxic models. I'm also surprised that rotarod performance is worse as compared to footprint one in males. Rotarod is able to detect motor coordination and balance deficits, which are key features of ataxia, however it is not specific for cerebellar dysfunction. Other types of motor or neurological impairments can also lead to altered rotarod performance, such as fatigue, muscle weakness, peripheral neuropathy, alterations of the vestibular system. Footprint parameters are a bit more altered in females (with the exception of the sway, why is the sway not significantly affected in females?).

Also, looking at the movies, the circling behaviour and head tilt of *Arel1*^{-/-} mice may suggest central or peripheral vestibular problems in addition to cerebellar defects.

Could the authors comment on these points?

For the reasons above, I would also attenuate the sentence "These results suggest that *Arel1* deficiency causes late-onset ataxia in mice". I would rather say it causes late onset motor impairment. This throughout the manuscript.

2) According to the WB they show in Fig 6g, cerebellum is not the brain area in which AREL1 is mostly expressed. They apparently do not detect abnormalities in the cerebral cortex or hippocampus of 12-month-old *Arel1*^{-/-} mice. However, the resolution of images in Extended Data Fig. 9e is a bit low to evaluate any specific loss of neurons. For sure there are not gross alterations. Could the authors specifically quantify neuronal loss in cortex or in midbrain where AREL1 is mostly expressed? Also considering the circling behaviour and head tilt of *Arel1*^{-/-} mice, a closer look at the ear and vestibular systems is important.

In summary, I think the authors have sufficient data supporting the involvement of Purkinje cells but not enough to exclude the involvement of other brain areas, including the central and vestibular systems. I suggest refining this analysis to better support their conclusions.

Referee #1 (Report for Author)

Overall, the authors addressed nearly all concerns raised by this reviewer, and the manuscript's quality has significantly improved. Especially, a tremendous amount of work was invested to address concerns related to over-expression artifacts, which undermined the significance of the original discovery in the prior manuscript. The AREL1 and ZRANB1 knockout (KO) or knockdown (KD) experiments with endogenous protein immunoprecipitation followed by Western blotting (IP-WB) support the previous IP-WB experiments conducted with triple over-expression systems, thereby satisfactorily resolving the major concerns. Methods for quantifying images are clearly described, and overall sample numbers for statistical analysis have significantly increased, further improving the quality and credibility. I only have two minor suggestions.

1. Extended data figure 3b: details on the statistics are not described. Including information on how the p-value is calculated (which statistical test was used) would be helpful. Was the data normalized at all? Endogenously biotinylated proteins such as PC, PCCC, and MCCC could serve as good normalization parameters if needed.

Response: We thank the reviewer for the helpful suggestion.

We performed four independent biotinylation experiment repeats to identify interacting partners of AREL1 on the lysosomal membrane. LC-MS/MS was performed using EASY-nLC 1000 system interfaced to Q Exactive HF. The database search was performed using Maxquant software, and the quantified protein list was further analyzed using Perseus software for statistical analysis. A two-sample t-test was conducted to calculate p-values using the implemented function in Perseus.

Based on the advice, we performed normalization using endogenously biotinylated proteins, namely pyruvate carboxylase (PC), propionyl-CoA carboxylase subunit a (PCCA), propionyl-CoA carboxylase subunit b (PCCB), methylcrotonoyl-CoA carboxylase subunit a (MCCC1) and methylcrotonoyl-CoA carboxylase beta chain (MCCC2), as the internal references. The V-ATPase subunits were still robustly enriched and highly ranked in the experimental samples after normalization (Expanded View Figure 3B).

We have revised the figure (Expanded View Figure 3B), figure legend (Page 50) as well as the Results and Methods section (Page 20) accordingly.

2. Alphafold3 - including a confidence score would be helpful, as currently there's no information on the probability of the prediction.

Response: We have included the confidence score in the legends for Figure 1P. The AlphaFold3-predicted interaction sites were verified using the co-immunoprecipitation assay, where single and double mutations of the H297 and A305 residues of V_oa3 markedly abrogated its interaction with AREL1 (Fig. 1Q).

Referee #2 (Report for Author)

The authors have addressed my concerns convincingly (Reviewer 3) and corrected my misunderstanding of some points about their experimental process.

The authors have added a substantial number of additional experiments in what appears to be a sincerely effort to address concerns raised by other reviewers.

From my perspective, the manuscript is suitable for publication in the EMBO Journal.

Response: We thank the reviewer for the helpful advice during revision.

Referee #3 (Report for Author)

In this study, Jiang et al. show that an ER-resident E3 ligase AREL1 initiates membrane contacts with lysosomes by interacting with the subunit a of V-ATPase. Disruption of AREL1-mediated ubiquitylation redistributes lysosomes to the cell periphery. They also show that mice lacking Arel1 exhibit age-dependent Purkinje cell loss and motor impairment.

The study is interesting, and the authors addressed most of my major concerns regarding the last part of the work. They have improved the evaluation of the motor capabilities of the Arel1^{-/-} mouse model and removed all sentences about a potential link between Arel1 and inherited cerebellar ataxia in humans, for which they had no evidence.

However, I have some doubts that the authors need to address before the manuscript is suitable for publication.

Overall, I still find the logic behind focusing on the cerebellum and in particular on Purkinje cells, and also the statements that Arel1 deficiency causes specifically cerebellar ataxia, a bit weak.

Here my concerns:

1) In males, the footprint parameters, which relate more to coordination and balance with respect to rotarod, are mildly affected at 12 months. Differences are a bit higher in established ataxic models. I'm also surprised that rotarod performance is worse as compared to footprint one in males. Rotarod is able to detect motor coordination and balance deficits, which are key features of ataxia, however it is not specific for cerebellar dysfunction. Other types of motor or neurological impairments can also lead to altered rotarod performance, such as fatigue, muscle weakness, peripheral neuropathy, alterations of the vestibular system.

Footprint parameters are a bit more altered in females (with the exception of the sway, why is the sway not significantly affected in females?).

Also, looking at the movies, the circling behaviour and head tilt of *Arel1*^{-/-} mice may suggest central or peripheral vestibular problems in addition to cerebellar defects.

Could the authors comment on these points?

For the reasons above, I would also attenuate the sentence "These results suggest that *Arel1* deficiency causes late-onset ataxia in mice". I would rather say it causes late onset motor impairment. This throughout the manuscript.

Response: We thank the reviewer for helping us refine the interpretation of mouse phenotypes.

It is interesting that the rotarod performance was worse as compared to footprint one in males, whereas footprint parameters were a bit more altered in females. We do not know why the sway was not significantly affected in females either. These complicated phenotypes merit further investigation and suggest that *Arel1* deficiency is very likely to cause central or peripheral vestibular problems in addition to cerebellar defects, as the reviewer pointed out.

We have incorporated these comments in the relevant part (Pages 13 and 14) as well as toned down the statements on the specificity to only Purkinje cells throughout the manuscript.

2) According to the WB they show in Fig 6g, cerebellum is not the brain area in which AREL1 is mostly expressed. They apparently do not detect abnormalities in the cerebral cortex or hippocampus of 12-month-old *Arel1*^{-/-} mice. However, the resolution of images in Extended Data Fig. 9e is a bit low to evaluate any specific loss of neurons. For sure there are not gross alterations. Could the authors specifically quantify neuronal loss in cortex or in midbrain where AREL1 is mostly expressed? Also considering the

circling behaviour and head tilt of *Are11*^{-/-} mice, a closer look at the ear and vestibular systems is important.

Response: We have included images showing the cerebral cortex and hippocampus of 12-month-old *Are11*^{-/-} mice and controls at higher magnifications in the revised Expanded View Figure 9E. We also point out that “whether *Are11* deficiency may result in other neurological impairments such as central and peripheral vestibular problems merits further investigation” (Page 14).

In summary, I think the authors have sufficient data supporting the involvement of Purkinje cells but not enough to exclude the involvement of other brain areas, including the central and vestibular systems. I suggest refining this analysis to better support their conclusions.

Dear Jie,

Thank you for addressing the final editorial points. I am now pleased to inform you that your manuscript has been accepted for publication in the EMBO Journal. Congratulations with a nice study!

Please note that it is The EMBO Journal policy for the transcript of the editorial process (containing referee reports and your response letters) to be published as an online supplement to each paper. If you should prefer removal of any referee-only figures included in the point-by-point response(s), e.g. because they may still be used for future publication or because they have been reproduced from published work by others, please do let us know immediately via response email.

More information is available here: https://www.embopress.org/transparent-process#Review_Process

If you have any questions, please do not hesitate to contact the Editorial Office or me directly. Thank you for this interesting contribution to The EMBO Journal!

Best wishes,

Ieva
